# CoDA: From Text-to-Image Diffusion Models to Training-Free Dataset Distillation

**Letian Zhou**
National University of Singapore
e1546594@u.nus.edu

**Songhua Liu**[*]
Shanghai Jiao Tong University
liusonghua@sjtu.edu.cn

**Xinchao Wang**[*]
National University of Singapore
xinchao@nus.edu.sg

## Abstract

Prevailing Dataset Distillation (DD) methods leveraging generative models confront two fundamental limitations. First, despite pioneering the use of diffusion models in DD and delivering impressive performance, the vast majority of approaches paradoxically require a diffusion model pre-trained on the full target dataset, undermining the very purpose of DD and incurring prohibitive training costs. Second, although some methods turn to general text-to-image models without relying on such target-specific training, they suffer from a significant distributional mismatch, as the web-scale priors encapsulated in these foundation models fail to faithfully capture the target-specific semantics, leading to suboptimal performance. To tackle these challenges, we propose Core Distribution Alignment (CoDA), a framework that enables effective DD using only an off-the-shelf text-to-image model. Our key idea is to first identify the "intrinsic core distribution" of the target dataset using a robust density-based discovery mechanism. We then steer the generative process to align the generated samples with this core distribution. By doing so, CoDA effectively bridges the gap between general-purpose generative priors and target semantics, yielding highly representative distilled datasets. Extensive experiments suggest that, without relying on a generative model specifically trained on the target dataset, CoDA achieves performance on par with or even superior to previous methods with such reliance across all benchmarks, including ImageNet-1K and its subsets. Notably, it establishes a new state-of-the-art accuracy of 60.4% at the 50-images-per-class (IPC) setup on ImageNet-1K. Our code is available on the project webpage: https://github.com/zzzlt422/CoDA

## 1 Introduction

Dataset Distillation (DD) Yu et al. (2023b); Xiao et al. (2025) has gained significant attention for its core promise: to synthesize a compact dataset, allowing drastically more efficient model training with highly competitive performance. Addressing the limitations of early DD methods Yin et al. (2023b); Yu et al. (2024; 2025), which often faced prohibitive computational demands and struggled to scale to large, high-resolution datasets, a recent wave of research has turned to leveraging powerful generative models like GANs (Goodfellow et al. (2020)) and Diffusion Models (Ho et al. (2020); Song et al. (2020a)). However, this emerging paradigm confronts two fundamental limitations.

First, although pioneering and highly effective, the vast majority of these methods (Minimax Gu et al. (2024); IGD Chen et al. (2025); Santiago et al. (2025); Moser et al. (2024); Zhao et al. (2025)) employ a diffusion model pre-trained or fine-tuned on the target dataset itself, a conceptually flawed process that incurs prohibitive computational costs. This approach puts the cart before the horse: the core motivation of DD is to create an efficient dataset substitute to avoid the prohibitive cost of training on the full dataset, yet this approach inherently requires first training a diffusion model on the full dataset, only to then use it to create the distilled dataset. For instance, the diffusion transformer (DiT) model (Peebles & Xie (2023)) used by Minimax Gu et al. (2024) to distill ImageNet first requires a pre-training phase on the full ImageNet dataset, consuming approximately 3,000 TPU-days.

Second, although some methods ($D^4M$ Su et al. (2024)) turn to general text-to-image models without relying on such target-specific training, they suffer from a significant distributional mismatch, as the web-scale priors encapsulated in these foundation models fail to faithfully capture

---

[*]Corresponding authors.

the target-specific semantics, leading to suboptimal performance. This mismatch is visually evident in Figure 1a. The general LDM's samples are confined to a limited region, failing to capture the data distribution and achieving a poor 52%. In contrast, the target-pretrained LDM achieves a strong accuracy of 60% precisely because it is inherently capable of generating a dataset that effectively covers the data distribution. This suggests that the remarkable success of these target-trained methods stems largely from this pre-existing alignment. A more detailed analysis is in Appendix A.12.

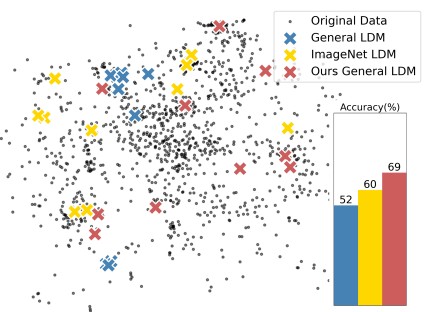

(a) Distribution and performance of samples by text-to-image LDM, LDM pre-trained on ImageNet, and text-to-image LDM with our method.

In this work, we tackle these challenges by proposing **Co**re **D**istribution **A**lignment (**CoDA**), a novel framework that enables effective DD using only a single, off-the-shelf text-to-image diffusion model. The foundation of **CoDA** is our hypothesis that a dataset's "intrinsic core distribution" is constituted by its most representative samples ($S_r$), which typically reside in high-density regions of the data manifold. Therefore, our approach consists of two stages: first, in **Distribution Discovery**, we introduce a novel density-based clustering mechanism to automatically and effectively identify these $S_r$; second, in **Distribution Alignment**, we subsequently utilize $S_r$ to directly guide the generative process, aligning the generated samples with core distribution.

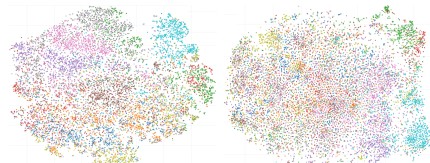

(b) ResNet18 space     (c) VAE space

Figure 1: (a): Performance comparison of different LDMs on ImageNette (IPC=10). (b-c): ImageNette original features in ResNet18 (b) and VAE (c) latent spaces. Each color indicates a different class.

By doing so, CoDA effectively bridges the gap between general-purpose generative priors and target-specific semantics, yielding distilled datasets that are both compact and highly representative, as demonstrated by extensive visualization and quantitative evaluations. In summary, our contributions are as follows:

- We are the first to confront the conceptual flaws of existing generative DD methods, enabling a off-the-shelf text-to-image model to achieve state-of-the-art performance.
- We introduce the concept of an "intrinsic core distribution" and a novel, density-based strategy to identify its representative samples, which are then used to guide the generative process and effectively bridge the model-dataset distributional mismatch.
- Our method establishes new state-of-the-art performance across all seven major ImageNet subsets. Notably, it establishes a new state-of-the-art accuracy of 60.4% at the 50-images-per-class (IPC) setup on ImageNet-1K.

## 2 PRELIMINARIES

### 2.1 BACKGROUND ON DATASET DISTILLATION

We consider a target dataset $T = \{(x_i, y_i)\}_{i=1}^{|T|}$, where each sample $x_i \in \mathbb{R}^d$ is drawn i.i.d. from a distribution $q(x)$ with a ground-truth label $y_i \in \{1, \ldots, C\}$. The goal of DD is to learn a much smaller synthetic dataset $S = \{(u_i, y_i)\}_{i=1}^{|S|}$, where $|S| \ll |T|$. This process can be formulated as an optimization problem, where the synthetic set $S$ is learned to serve as an effective proxy for $T$. Specifically, the goal is to find an optimal set $S^*$ that minimizes the expected loss over the true data distribution when a model is trained on $S$:

$$S^* = \underset{S}{\operatorname{argmin}} \, \mathbb{E}_{(x,y) \sim q(x)}[\ell(f_{\theta_S}, x, y)] \qquad (1)$$

where $\theta_S$ represents the model parameters after training to convergence on the synthetic set $S$, $f_{\theta_S}$ is the corresponding trained model, and $\ell$ is the loss function.

The degree of compression is measured by Images Per Class (IPC), while the quality of the synthetic set $S$ is measured by the accuracy of a model trained on it against the original test dataset.

## 2.2 LATENT DIFFUSION AND SCORE-BASED GUIDANCE

In this work, we employ a Latent Diffusion Model (LDM) Rombach et al. (2022) that operates in a compressed latent space $z = E(x)$, encoded by a VAE encoder $E$ from images $x$. Diffusion Models (DMs) (Ho et al. (2020); Song et al. (2020a)) train a network to predict the noise $\epsilon$ added during the forward noising process:

$$z_t = \sqrt{\bar{\alpha}_t} z_0 + \sqrt{1 - \bar{\alpha}_t} \epsilon \tag{2}$$

The core principle of DMs is the direct proportionality between the optimal noise prediction and the model's score function (Ho et al. (2020); Song et al. (2020b)):

$$\nabla_{z_t} \log p_t(z_t) = -\frac{\epsilon}{\sqrt{1 - \bar{\alpha}_t}} \tag{3}$$

This equivalence implies that modifying the noise prediction $\epsilon_\theta$ is functionally equivalent to modifying the gradient field that the sampling process follows. This enables Guided Diffusion (Dhariwal & Nichol (2021); Bansal et al. (2023); Yu et al. (2023a)). By introducing an external energy function (EBM) $f_C(z_t)$, the gradient of this function during sampling $\nabla_{z_t} f_C(z_t)$ is superimposed onto the original $\epsilon_\theta$ prediction. This creates a new, guided gradient field, forcing the generative process towards states that satisfy condition $C$. Our methodology builds upon this principle, adding an additional energy guidance field derived from the target dataset's core distribution $S_r$.

## 3 METHOD

Our approach comprises two main stages: **Distribution Discovery** and **Distribution Alignment**. In the first stage, we identify the representative samples ($S_r$) using a rigorous density-based clustering mechanism, primarily based on the HDBSCAN algorithm. Aside from the initial VAE encoding, the remainder of **Distribution Discovery** is executed on CPUs. We then employ a novel **SplitCluster** mechanism to ensure the number of samples in $S_r$ meets our target, Images Per Class (IPC). Subsequently, in the alignment stage, we introduce a new guidance term derived from $S_r$ to modify the predicted noise, thereby steering the sampling process of the generative model.

### 3.1 DISTRIBUTION DISCOVERY

The objective of the Distribution Discovery stage is to locate high-density regions within the target dataset's manifold and, from these regions, to extract a set of representative samples $S_r$. This process begins by mapping the images into a suitable feature space where the manifold's structure can be analyzed. To adhere to our principle of relying on an off-the-shelf generative model, we eschew any components trained on the target dataset, which includes avoiding the external feature extractors common in prior work (Zhou et al. (2022); Yin et al. (2023a); Shao et al. (2024)), such as a ResNet without its classifier. Instead, we leverage the VAE encoder native to the LDM to map the target dataset into its latent space for the subsequent search for $S_r$.

**The VAE Latent Space Challenge.** However, a significant challenge arises from the intrinsic properties of the VAE encoder. The encoder's latent space is regularized by a standard Gaussian prior $\mathcal{N}(0, \mathbf{I})$, which compels all encoded data points into a unified hyperspherical latent distribution (Kingma & Welling (2013)). While this regularization is essential for maintaining a smooth and continuous latent manifold suitable for the diffusion model's generative process, it inherently mixes features from different classes, thereby severely degrading inter-class separability and intra-class compactness. As shown in Figure 1b and Figure 1c, these factors collectively result in a latent space that is not well-structured for direct clustering, exhibiting poor separation between classes compared to features from a target-trained classifier.

**Limitations of Prior K-Means-Based Approaches.** Previous works, such as D$^4$M (Su et al. (2024)) and MGD$^3$ (Santiago et al. (2025)), have also attempted to identify representative prototypes in VAE latent space. However, unaware of or failing to account for the aforementioned issues, they typically resort to a direct application of K-Means clustering for each class, forcing the number of clusters, $k$, to be equal to the IPC. This one-step approach is brittle in the face of the latent space challenges for two primary reasons. First, K-Means implicitly assumes clusters are convex and isotropic, requirements that are fundamentally violated by the VAE latent space, which contains

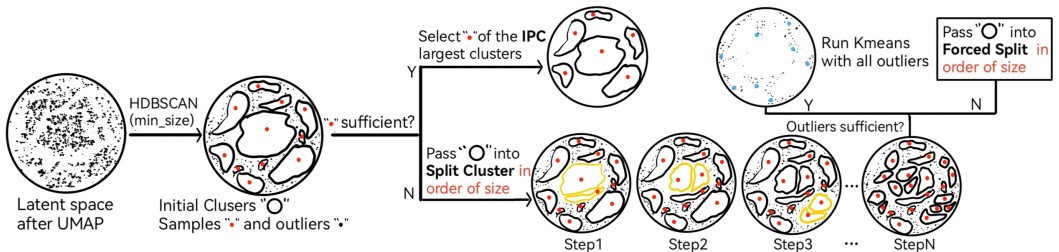

Figure 2: The workflow of our density-based pipeline. **Red** dots are samples found by HDBSCAN, while **Blue** dots are found by K-Means on outliers (**Black** dots). Yellow regions highlight two new clusters split from a "mother-cluster".

arbitrarily shaped class structures within a single Gaussian hypersphere. Second, K-Means is highly sensitive to outliers, as it assigns every point to a cluster. When the IPC is large, the algorithm's objective of minimizing Within-Cluster Sum of Squares (WCSS) forces it to establish spurious clusters for outliers, thereby failing to identify the true high-density regions. However, we aim to create a compressed dataset utilizing information in all original data points. Therefore, we deliberately avoid data purification techniques like LOF (Breunig et al. (2000)) that would discard outliers.

All these limitations and constraints motivate the need for a more advanced mechanism capable of finding meaningful cluster centers without strong distributional assumptions, and one that can even make effective use of the outliers themselves.

**Our density-based pipeline.** Our Distribution Discovery stage employs a more robust, multi-step pipeline, which is applied independently to each class of the target dataset. We select HDBSCAN (Hierarchical Density-Based Spatial Clustering of Applications with Noise) (McInnes et al. (2017)) as our core algorithm to automatically identify the high-density regions, as it makes no assumptions about cluster shape, robustly handles outliers, and allows us to control the granularity of the discovered clusters via its $min\_cluster\_size$ parameter. However, before applying HDBSCAN, we first leverage UMAP (Uniform Manifold Approximation and Projection) (McInnes et al. (2018)) to preprocess the latent embeddings from the VAE. This step not only reduces dimensionality, enabling our density-based HDBSCAN to operate reliably, but also allows us to tune the local compactness of the VAE latent space to yield a more distinct clustering structure.

Following the initial HDBSCAN, we identify $M$ valid high-density clusters from the data, which denoted as the collection $C_0 = \{c_1, c_2, \ldots, c_M\}$. From each cluster $c_i \in C_0$, we then select the single point $s_i$ with the highest membership probability to form our initial set of representative samples, $S_0 = \{s_1, s_2, \ldots, s_M\}$. A key challenge, however, is that the size $M$ is data-driven and therefore highly unlikely to equal the target IPC. We devise a **post-processing** scheme to resolve this discrepancy. The overall pipeline is shown in Figure 2, and the details of the **SplitCluster** and **ForcedSplit** functions are provided in Algorithm 1. The full pseudocode is provided in Appendix A.4.

If $M \geq$ IPC, we select the representative samples corresponding to the IPC largest clusters to form the final set $S_r$. Conversely, if $M <$ IPC, we bridge the gap by iteratively splitting clusters from $C_0$ to identify new representative samples.

1. **Strategy 1: SplitCluster.** We iteratively apply HDBSCAN to the largest viable "mother-cluster" to identify its internal sub-clusters. We use the representative points ($s_i$) from each sub-cluster as candidate initial centroids for K-Means, selecting the optimal pair that yields the lowest inertia for a 2-way partition. This process successively replaces the single original sample with two new, well-placed ones. If the number of samples is still insufficient and **Strategy 1** fails to split any remaining clusters, we turn to **Strategy 2**.

2. **Strategy 2: Clustering Outliers.** In this stage, we utilize the unassigned outliers as a secondary source. Applying K-Means exclusively to this population serves a dual purpose: it is complementary, as it finds samples in low-density regions that HDBSCAN ignored, and it is non-disruptive, as it prevents K-Means from altering the high-quality clusters already identified. If sufficient outliers exist, we apply K-Means directly to this population to identify the exact number of remaining samples needed, efficiently closing the gap in a single step. If there are insufficient outliers, we turn to **Strategy 3**.

---

**Algorithm 1** Post-processing Scheme for IPC Matching

---

1: **Global variables:** $C$, current clusters; $S$, corresponding representative samples.
2: **procedure** SPLITCLUSTER($c_{\text{mother}}$, $min\_cluster\_size$)
3:  Run HDBSCAN on $c_{\text{mother}}$ with $min\_cluster\_size$ to find $k$ sub-clusters as $S_{\text{cand}}$
4:  If $k < 2$ **return False**
5:  $S \leftarrow S \setminus \{s_{\text{mother}}\}$ and $C \leftarrow C \setminus \{c_{\text{mother}}\}$
6:  Iterate all pairs $(s_i, s_j)$ from $S_{\text{cand}}$ as initial seeds for K-Means ($k = 2$) on $c_{\text{mother}}$, select the pair yielding the minimum resulting inertia, denoted as $\{c_{\text{new1}}, c_{\text{new2}}\}$ and $\{s_{\text{new1}}, s_{\text{new2}}\}$
7:  $C \leftarrow C \cup \{c_{\text{new1}}, c_{\text{new2}}\}$ and $S \leftarrow S \cup \{s_{\text{new1}}, s_{\text{new2}}\}$
8:  **return True**
9: **end procedure**

10: **procedure** FORCEDSPLIT($c_{\text{mother}}$, $min\_size$)
11:  **while** SPLITCLUSTER($c_{\text{mother}}$, $min\_size$) = **False** & $min\_size > 2$ **do**
12:    $min\_size \leftarrow \max(2, \lfloor min\_size \times 0.75 \rfloor)$         ▷ Relax the constraint
13:  **end while**
14: **end procedure**

---

3. **Strategy 3: ForcedSplit.** This stage works by gradually lowering the $min\_cluster\_size$ threshold until the cluster can be split into two. This entire process is repeated until the number of samples finally aligns with the IPC.

## 3.2 DISTRIBUTION ALIGNMENT

In the **Distribution Alignment** stage, we leverage the obtained set of representative samples, $S_r = \{s_1, \ldots, s_{\text{IPC}}\}$, to steer the generation process. We achieve this by introducing a dynamic, data-driven guidance term derived from $S_r$. This term works by modifying the predicted noise, thereby augmenting the standard Classifier-Free Guidance (CFG) framework (Ho & Salimans (2022)).

At each timestep $t$ of the reverse process, we start with the noisy latent $z_t$. This latent is processed by the U-Net $\epsilon_\theta$ to produce two noise estimates for CFG: the unconditional noise $\epsilon_\theta(z_t, t, \emptyset)$, which is intentionally left unmodified to preserve the backbone model's inherent stability, and the original text-conditional noise $\epsilon_\theta(z_t, t, c)$, which serves as the basis for our guidance.

First, we can estimate the predicted clean latent $\hat{z}_0$ from $\epsilon_\theta(z_t, t, c)$, using the following closed-form solution derived from Equation (2):

$$\hat{z}_0(z_t) = \frac{1}{\sqrt{\bar{\alpha}_t}}(z_t - \sqrt{1 - \bar{\alpha}_t}\epsilon_\theta(z_t, t, c)) \tag{4}$$

Next, to synthesize the final IPC images (per class), we employ a sequential assignment strategy: the generation of the $j$-th synthetic image (where $j \in \{1, \ldots, \text{IPC}\}$) is exclusively and consistently guided by the $j$-th representative sample ($s_j \in S_r$). Thus, for the entire generation process of the $j$-th image, the guidance vector at every timestep is defined as:

$$g(z_t) = s_j - \hat{z}_0(z_t) \tag{5}$$

This guidance is then scaled by $\gamma$, a hyperparameter controlling the alignment strength, yielding our required correction vector, $\Delta\hat{z}_0 = \gamma \cdot g(z_t)$. Applying this correction to the original prediction is equivalent to a linear interpolation: $\hat{z}_{0,\text{new}} = \hat{z}_0(z_t) + \Delta\hat{z}_0 = (1-\gamma)\hat{z}_0(z_t) + \gamma s_j$. To apply this $\Delta\hat{z}_0$ during the denoising process, we must translate it into an equivalent correction in the noise space, $\Delta\epsilon_\theta$. From Equation (4), we can derive the relationship between $\Delta\hat{z}_0$ and $\Delta\epsilon_\theta$ as follows:

$$\hat{z}_{0,\text{new}} = \frac{1}{\sqrt{\bar{\alpha}_t}}(z_t - \sqrt{1 - \bar{\alpha}_t}(\epsilon_\theta + \Delta\epsilon_\theta)) = \hat{z}_0(z_t) - \left(\frac{\sqrt{1 - \bar{\alpha}_t}}{\sqrt{\bar{\alpha}_t}}\right)\Delta\epsilon_\theta \tag{6}$$

This leads to our final formulation for the correction term:

$$\Delta\epsilon_\theta = \gamma \cdot g(z_t) \cdot \left(-\frac{\sqrt{\bar{\alpha}_t}}{\sqrt{1 - \bar{\alpha}_t}}\right) \tag{7}$$

Finally, this correction term $\Delta\epsilon_\theta$ is added to the original $\epsilon_\theta(z_t, t, c)$ to obtain $\epsilon'_\theta = \epsilon_\theta(z_t, t, c) + \Delta\epsilon_\theta$. This $\epsilon'_\theta$ is then processed through the standard CFG framework, and the resulting final noise estimate is used by the scheduler for the next denoising step.

## 4 EXPERIMENTS

### 4.1 EXPERIMENT SETUP

**Datasets and Evaluation.** Our method is extensively evaluated on the challenging ImageNet-1K benchmark ($\mathbf{224{\times}224}$) (Deng et al. (2009)) and its eight 10-class subsets ($\mathbf{256{\times}256}$), including the commonly used ImageNette (Howard (2019)), ImageIDC (Kim et al. (2022)), ImageWoof and the ImageNet-A through ImageNet-E sets from Cazenavette et al. (2023). The scope of our evaluation extends further in two key aspects. First, we include additional benchmarks like Places365 ($\mathbf{256{\times}256}$) (Zhou et al. (2017)) to demonstrate the versatility of the proposed method. Second, we perform multi-resolution experiments, testing at $\mathbf{128{\times}128}$ and establishing the first high-resolution baseline for dataset distillation at $\mathbf{1024{\times}1024}$.

We evaluate our distilled datasets using two standard protocols. For the **hard-label** protocol, we follow the established setup of Gu et al. (2024). For our ImageNet-1K evaluation, we employ a **soft-label** knowledge distillation framework from Sun et al. (2024), where both the teacher and student models are ResNet-18 architectures.

**Baselines.** Our diffusion-based baselines are categorized into two groups: **(1)** methods that utilize generative models pre-trained or fine-tuned on the target dataset, including Minimax (Gu et al. (2024)), LD3M (Moser et al. (2024)), IGD (Chen et al. (2025)), MGD³ (Santiago et al. (2025)), as well as the standard LDM (Rombach et al. (2022)) and DiT (Peebles & Xie (2023)); and **(2)** methods that employ truly off-the-shelf models, D⁴M (Su et al. (2024)) and standard SDXL (Podell et al. (2023)). Furthermore, for a more extensive verification, we also include several key non-diffusion paradigms: Methods based on Generative Priors (GLaD Cazenavette et al. (2023); H-PD Zhong et al. (2025)); Statistical Matching (SRe2L Yin et al. (2023a); G-VBSM Shao et al. (2024)); and Patch Assembly (RDED Sun et al. (2024)).

**Implementation details.** For the **Distribution Discovery** stage, we configure UMAP with an output dimension of 50 and $min\_dist = 0.0$ to maximize cluster density. For HDBSCAN, we fix $min\_samples = 3$ to preserve the data's hierarchical structure. The key hyperparameters for this stage are UMAP's $n\_neighbors$ and HDBSCAN's $min\_cluster\_size$. For the **Distribution Alignment** stage, we employ SDXL as our base generative model with a DPM++ Karras sampler over 50 inference steps (Lu et al. (2025); Karras et al. (2022)). The text prompt for each class is the first part of its official label, while the negative prompt is left empty and the CFG scale is fixed at 5.0. The primary hyperparameters are our alignment guidance strength ($\gamma$ in Equation (7)) and Prior Injection Steps (PIS). We define PIS as the final number of steps that revert to the standard CFG, allowing the model's prior to naturally refine image details and enhance photorealism.

### 4.2 COMPARISON WITH STATE-OF-THE-ART METHODS

**ImageIDC and ImageNette.** The primary results of our method against state-of-the-art (SOTA) approaches on ImageIDC and ImageNette are presented in Table 1. For a fair comparison, the results for D⁴M are based on the hard-label protocol.

Our pipeline establishes a new and decisive SOTA across all IPC configurations on both datasets. Specifically, it outperforms the prior SOTA by **2.6%**, **2.7%**, and **5.5%** on ImageIDC for IPC=10, 20, and 50 respectively, and by **2.4%**, **0.9%**, and **3.6%** on ImageNette.

These results also reveal the impact of the distributional mismatch. The text-to-image SDXL shows a significant performance drop compared to the LDM and DiT pre-trained on ImageNet. Meanwhile, MGD³'s performance also collapses when transplanted to the SDXL, suggesting its effectiveness is not from its algorithm alone, but largely dependent on the pre-existing distributional alignment.

**ImageWoof.** We also provide a comprehensive comparison on ImageWoof, a challenging 10-class subset characterized by extremely high inter-class similarity. As shown in Table 2, while previous SOTA methods maintain a marginal advantage in low storage budget settings (IPC $\leq$ 20), our method establishes a new SOTA across all configurations with IPC $\geq$ 50, demonstrating its robustness on challenging, fine-grained datasets. For example, CoDA achieves 71.4% on ResNet-18 at IPC=100, significantly outperforming the prior SOTA of 68.8%.

Table 1: Comparison with state-of-the-art methods on ImageIDC and ImageNette across various IPC settings. All methods are evaluated by training a ResNet10-AP at a 256x256 resolution. The best results are in **bold**, and the second best are in **bold**.

| Type | IPC | IDC | | | Nette | | |
|---|---|---|---|---|---|---|---|
| | | 10 | 20 | 50 | 10 | 20 | 50 |
| - | Random | 48.1±0.8 | 52.5±0.9 | 68.1±0.7 | 54.2±1.6 | 63.5±0.5 | 76.1±1.1 |
| Based on Model Pretrained With ImageNet | Minimax | 53.1±0.2 | 59.0±0.4 | 69.6±0.2 | 62.0±0.2 | 66.8±0.4 | 76.6±0.2 |
| | LDM | 50.8±1.2 | 55.1±2.0 | 63.8±0.4 | 60.3±3.6 | 62.0±2.6 | 71.0±1.4 |
| | LDM+MGD³ | 53.2±0.2 | 58.3±1.7 | 67.2±1.3 | 61.9±4.1 | 65.3±1.3 | 74.2±0.9 |
| | DiT | 54.1±0.4 | 58.9±0.2 | 64.3±0.6 | 59.1±0.7 | 64.8±1.2 | 73.3±0.9 |
| | DiT+MGD³ | **55.9±2.1** | **61.9±0.9** | **72.1±0.8** | **66.4±2.4** | **71.2±0.5** | **79.5±1.3** |
| Based on General text-to-image Model | SDXL | 40.1±0.8 | 41.8±0.2 | 45.1±1.8 | 51.7±0.2 | 56.1±0.4 | 67.3±0.2 |
| | SDXL+D⁴M | 41.1±1.0 | 45.1±0.7 | 56.6±0.2 | 58.4±1.4 | 66.6±1.9 | 73.4±0.7 |
| | SDXL+MGD³ | 47.2±0.6 | 52.0±1.0 | 61.4±1.4 | 59.1±0.5 | 63.9±0.8 | 75.5±0.9 |
| | SDXL+Ours | **58.5±0.9** | **64.6±0.5** | **77.6±0.6** | **68.8±0.1** | **72.1±0.2** | **83.1±0.3** |

Table 2: Performance comparison with pre-trained diffusion models and other SOTA methods on ImageWoof. All the results are reproduced by us for the $256{\times}256$ resolution. The missing results are due to out-of-memory. Results shown for the previous works are from MGD³ (Santiago et al. (2025)). Best result **bold** and second best **bold**. The arrow annotations indicate the performance difference between CoDA and the previous SOTA.

| IPC | Test Model | Random | Herding | DiT | DM | IDC-1 | GLaD | Minimax | MGD³ | CoDA(Ours) | Full |
|---|---|---|---|---|---|---|---|---|---|---|---|
| 10 (0.8%) | ConvNet-6 | 24.3±1.1 | 26.7±0.5 | 34.2±1.1 | 26.9±1.2 | 33.3±1.1 | 33.8±0.9 | **37.0±1.0** | 34.7±1.1 | **35.7±1.4** (↓1.3) | 86.4±0.2 |
| | ResNetAP-10 | 29.4±0.8 | 32.0±0.3 | 34.7±0.5 | 30.3±1.2 | 39.1±0.5 | 32.9±0.9 | **39.2±1.3** | **40.4±1.9** | 39.2±0.7 (↓1.2) | 87.5±0.5 |
| | ResNet-18 | 27.7±0.9 | 30.2±1.2 | 34.7±0.4 | 33.4±0.7 | 37.3±0.2 | 31.7±0.8 | 37.6±0.9 | **38.5±2.5** | **38.8±1.3** (↑0.3) | 89.3±1.2 |
| 20 (1.6%) | ConvNet-6 | 29.1±0.7 | 29.5±0.3 | 36.1±0.8 | 29.9±1.0 | 35.5±0.8 | - | 37.6±0.9 | **39.0±3.5** | **38.1±1.3** (↓0.9) | 86.4±0.2 |
| | ResNetAP-10 | 32.7±0.4 | 34.9±0.1 | 41.1±0.8 | 35.2±0.6 | **43.4±0.3** | - | **45.8±0.5** | 43.0±1.6 | 42.5±0.6 (↓3.3) | 87.5±0.5 |
| | ResNet-18 | 29.7±0.5 | 32.2±0.6 | 40.5±0.5 | 29.8±1.7 | 38.6±0.2 | - | **42.5±0.6** | 41.9±2.1 | **42.0±0.8** (↓0.5) | 89.3±1.2 |
| 50 (3.8%) | ConvNet-6 | 41.3±0.6 | 40.3±0.7 | 46.5±0.8 | 44.4±1.0 | 43.9±1.2 | - | 53.9±0.6 | **54.5±1.6** | **56.8±0.9** (↑2.3) | 86.4±0.2 |
| | ResNetAP-10 | 47.2±1.3 | 49.3±0.7 | 49.3±0.2 | 47.1±1.1 | 48.3±1.0 | - | 56.3±1.0 | **56.5±1.9** | **59.4±1.0** (↑2.9) | 87.5±0.5 |
| | ResNet-18 | 47.9±1.8 | 48.3±1.2 | 50.1±0.5 | 46.2±0.6 | 48.3±0.8 | - | 57.1±0.6 | **58.3±1.4** | **61.2±0.9** (↑2.9) | 89.3±1.2 |
| 70 (5.4%) | ConvNet-6 | 46.3±0.6 | 46.2±0.6 | 50.1±1.2 | 47.5±0.8 | 48.9±0.7 | - | **55.7±0.9** | 55.1±2.5 | **56.4±1.2** (↑1.3) | 86.4±0.2 |
| | ResNetAP-10 | 50.8±0.6 | 53.4±1.4 | 54.3±0.9 | 51.7±0.8 | 52.8±1.8 | - | 58.3±0.2 | **60.2±2.4** | **61.2±0.7** (↑1.0) | 87.5±0.5 |
| | ResNet-18 | 52.1±1.0 | 49.7±0.8 | 51.5±1.0 | 51.9±0.8 | 51.1±1.7 | - | 58.8±0.7 | **59.7±2.7** | **62.4±1.4** (↑2.7) | 89.3±1.2 |
| 100 (7.7%) | ConvNet-6 | 52.2±0.4 | 54.4±1.1 | 53.4±0.3 | 55.0±1.3 | 53.2±0.9 | - | **61.1±0.7** | 60.1±1.2 | **62.4±0.7** (↑1.3) | 86.4±0.2 |
| | ResNetAP-10 | 59.4±1.0 | 61.7±0.9 | 58.3±0.8 | 56.4±0.8 | 56.1±0.9 | - | 64.5±0.2 | **66.5±1.0** | **67.6±0.8** (↑1.1) | 87.5±0.5 |
| | ResNet-18 | 61.5±1.3 | 59.3±0.7 | 58.9±1.3 | 60.2±1.0 | 58.3±1.2 | - | 65.7±0.4 | **68.8±0.7** | **71.4±1.4** (↑2.6) | 89.3±1.2 |

**ImageNet-A to ImageNet-E.** We evaluate the cross-architecture performance of our method against several generative baselines with four distinct downstream architectures: AlexNet (Krizhevsky et al. (2012)), VGG11 (Simonyan & Zisserman (2014)), ResNet18 (He et al. (2016)), and ViT (Dosovitskiy et al. (2020)). Specific data are presented in Table 3. Our method demonstrates superior cross-architecture generalization, outperforming the prior SOTA by **9.6%**, **8.4%**, **7.5%**, **4.8%**, **6.6%** on ImageNet-A to ImageNet-E, respectively. We provide the results at other resolutions and a more detailed breakdown of the cross-architecture performance in Appendix A.8.

**ImageNet-1K.** We compare our method against various categories of SOTA baselines on ImageNet-1K using the soft-label protocol, which is detailed in Table 4.

At IPC=10, while IGD, which relies on an ImageNet-trained diffusion models, holds the top position, our method achieves a massive lead over our truly off-the-shelf competitor D⁴M (**+16.4%**) and performs on par with the excellent ImageNet-trained method Minimax. At IPC=50, we not only decisively outperform D⁴M (**+5.2%**), but also surpass the strongest baseline IGD, establishing a new SOTA accuracy of **60.4%**.

**Baselines on 1024×1024 Resolution Images** As shown in Table 5, we provide baseline results on seven common subsets at a resolution of 1024×1024 at IPC=10. We present a broad cross-architecture performance comparison, testing ResNet18, ResNet50, ResNet101, ResNet10 with average pooling and ViT architectures for ImageIDC, ImageNette, and ImageNet-A through ImageNet-E. All experiments use hard labels and adhere to the setup in Gu et al. (2024), and all results are the mean and standard deviation of three independent runs.

Table 3: Comparison with SOTA methods on ImageNet-A to ImageNet-E with IPC=10. All results are the average accuracy across all four architectures and five independent runs.

| | | ImNet-A | ImNet-B | ImNet-C | ImNet-D | ImNet-E |
|---|---|---|---|---|---|---|
| DC | Pixel | 52.3±0.7 | 45.1±8.3 | 40.1±7.6 | 36.1±0.4 | 38.1±0.4 |
| | GLaD | 53.1±1.4 | 50.1±0.6 | 48.9±1.1 | 38.9±1.0 | 38.4±0.7 |
| | H-PD | 54.1±1.2 | 52.0±1.1 | 49.5±0.8 | 39.8±0.7 | 40.1±0.7 |
| | LD3M | 55.2±1.0 | 51.8±1.4 | 49.9±1.3 | 39.5±1.0 | 39.0±1.3 |
| DM | Pixel | 44.4±0.5 | 52.6±0.4 | 50.6±0.5 | 47.5±0.7 | 35.4±0.4 |
| | GLaD | 52.8±1.0 | 51.3±0.6 | 49.7±0.4 | 36.4±0.4 | 38.6±0.7 |
| | H-PD | 55.1±0.5 | 54.2±0.5 | 50.8±0.4 | 37.6±0.6 | 39.9±0.7 |
| | LD3M | 57.0±1.3 | 52.3±1.1 | 48.2±4.9 | 39.5±1.5 | 39.4±1.8 |
| - | MGD$^3$ | 63.4±0.8 | 66.3±1.1 | 58.6±1.2 | 46.8±0.8 | 51.1±1.0 |
| | Ours | **73.0±0.7** | **74.7±1.2** | **66.1±1.1** | **51.6±0.8** | **57.7±0.7** |

Table 4: Comparison with SOTA on ImageNet-1K using the soft-label protocol.

| Type | Method | IPC=10 | IPC=50 |
|---|---|---|---|
| Non-diffusion | SRe2L | 21.3±0.6 | 46.8±0.2 |
| | G-VBSM | 31.4±0.5 | 51.8±0.4 |
| | RDED | 42.0±0.1 | 56.5±0.1 |
| Model Trained On ImageNet | Minimax | 44.3±0.5 | 58.6±0.3 |
| | DiT | 39.6±0.4 | 52.9±0.3 |
| | IGD | **46.2±0.6** | **60.3±0.4** |
| General Model | D$^4$M | 27.9±0.2 | 55.2±0.1 |
| | Ours | **44.3±0.1** | **60.4±0.2** |

Table 5: Cross-architecture comparison on the ImageIDC, ImageNette and ImageNetA-E datasets under the IPC10 setting at **1024×1024** resolution. The evaluated architectures include ResNet18, ResNet50, ResNet101, ResNet10-AP and ViT. Note that each result represents the mean and standard deviation obtained from three independent runs of the architecture in the current row.

| | ImageIDC | ImageNette | ImageA | ImageB | ImageC | ImageD | ImageE |
|---|---|---|---|---|---|---|---|
| ResNet10-AP | 54.9 ± 1.7 | 57.1 ± 1.1 | 79.2 ± 0.2 | 81.1 ± 0.3 | 72.2 ± 0.4 | 52.3 ± 0.1 | 62.7 ± 0.5 |
| ResNet18 | 50.5 ± 1.0 | 56.9 ± 2.0 | 79.7 ± 1.0 | 79.5 ± 0.5 | 71.4 ± 1.1 | 52.7 ± 0.6 | 64.9 ± 1.5 |
| ResNet50 | 46.4 ± 1.3 | 52.9 ± 1.3 | 77.1 ± 0.1 | 73.2 ± 1.4 | 70.6 ± 0.9 | 49.7 ± 2.2 | 61.7 ± 0.7 |
| ResNet101 | 45.8 ± 0.5 | 52.0 ± 0.5 | 72.0 ± 0.1 | 71.8 ± 0.8 | 66.6 ± 1.1 | 50.6 ± 1.3 | 61.0 ± 0.9 |
| ViT | 47.5 ± 1.0 | 49.0 ± 1.2 | 72.1 ± 0.8 | 70.5 ± 1.2 | 61.8 ± 0.7 | 47.6 ± 1.2 | 56.6 ± 1.6 |

Table 6: Comparing Ours($\mathcal{R}$) and Ours($\mathcal{G}$) against the SOTA baseline on ImageIDC and ImageNette across different IPC. The best results are in **bold**, and the second best are in **bold**.

| Method | ImageIDC | | | ImageNette | | |
|---|---|---|---|---|---|---|
| | 10 | 20 | 50 | 10 | 20 | 50 |
| DiT+MGD$^3$ | 55.9±2.1 | 61.9±0.9 | 72.1±0.8 | **66.4±2.4** | **71.2±0.5** | 79.5±1.3 |
| Ours ($\mathcal{R}$) | **56.3±0.9** | **62.8±0.6** | **75.7±1.9** | 66.1±1.1 | 70.7±0.6 | **81.9±0.1** |
| Ours ($\mathcal{G}$) | **58.5±0.9** | **64.6±0.5** | **77.6±0.6** | **68.8±0.1** | **72.1±0.2** | **83.1±0.3** |

Table 7: ResNet18 accuracy on the first 10 classes of Places365 at IPC=10.

| Method | Accuracy (%) |
|---|---|
| Random | 41.7 ± 1.6 |
| SDXL | 33.9 ± 2.1 |
| SDXL + KMeans | 34.4 ± 1.6 |
| SDXL + Ours | **47.0 ± 1.6** |

# 5 ABLATION STUDY

**Ablation of Method Components.** Each stage of our method yields a distinct dataset. The first stage, Distribution Discovery, produces $\mathcal{R}$: a set of selected and resized real images from the original dataset. This set represents the output of our efficient, CPU-based process alone. The subsequent Distribution Alignment stage produces $\mathcal{G}$: the final set of generated images.

The results in Table 6 are striking. Our CPU-derived set $\mathcal{R}$ is sufficient to outperform the SOTA baseline across nearly all settings. Specifically, it surpasses the prior SOTA by **0.4%**, **0.9%**, and **3.6%** on ImageIDC for IPC=10, 20, and 50, and by **2.4%** on ImageNette for IPC=50. Leveraging this strong foundation, our generated set $\mathcal{G}$ improves performance even more significantly.

**Generalizability and Computational Efficiency.** Our framework's generalizability and efficiency provide a key advantage. Crucially, our framework achieves genuine zero-shot generalization, performing well on new domains like Places365 without any modification, as validated in Table 7. As the results show, our method significantly improves on the base SDXL model's performance and is substantially more effective than a direct K-Means. This overcomes the fundamental limitation of target-trained methods, which are locked into their original training domain and require another multi-thousand TPU-day re-training effort to adapt to new datasets. A further case study on domain change is provided in Appendix A.9.

Moreover, the efficiency is evident across both stages. For example, on ImageNet-1K, the **Distribution Discovery** stage completes in 2.2 hours for all 1,000 classes. The subsequent **Distribution Alignment** stage only adds a few arithmetic operations per step, which introduces negligible computational overhead compared to the U-Net forward pass, generating a 224×224 image within 2.7s. A more detailed computational analysis is provided in Appendix A.10.

**Efficacy of Distribution Discovery Components.** To demonstrate the importance of the "intrinsic core distribution" and our discovery stage's components, we conduct a step-by-step ablation study.

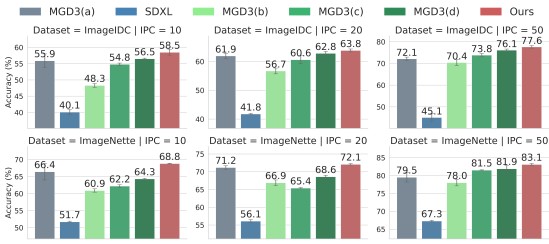 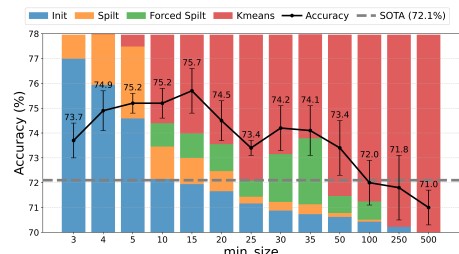

Figure 3: Step-by-step ablation on the efficacy of our Distribution Discovery stage components. **MGD$^3$(a)** is the ImageNet-trained SOTA baseline; **MGD$^3$(b)** is the direct application of MGD$^3$ to SDXL; **MGD$^3$(c)** adds UMAP before K-Means; **MGD$^3$(d)** uses our full discovery pipeline.

Figure 4: Analysis of $min\_size$ sensitivity on ImageIDC at IPC=50. The solid line tracks the accuracy of a model trained on the **real-image set** ($\mathcal{R}$). The stacked area chart illustrates the proportional source of the final IPC samples.

Table 8: Performance across different $\gamma$ and PIS values on ImageIDC (IPC=10). PIS=50 is equivalent to $\gamma = 0$. Cells highlighted in yellow indicate performance greater than the SOTA of 55.9.

Table 9: Performance with only 25 inference steps. Cells highlighted in yellow indicate performance greater than the SOTA of 55.9.

|  | $\gamma = 0.01$ | $\gamma = 0.03$ | $\gamma = 0.05$ | $\gamma = 0.08$ | $\gamma = 0.10$ | $\gamma = 0.15$ | $\gamma = 0.20$ |
|---|---|---|---|---|---|---|---|
| PIS=0 | 45.7±1.1 | 54.7±1.2 | 57.7±0.6 | 56.1±1.0 | 55.9±1.5 | 57.1±1.5 | 55.9±1.3 |
| PIS=5 | 45.5±1.0 | 54.6±0.4 | 55.8±0.2 | 56.0±0.8 | 58.5±0.9 | 57.0±1.1 | 56.9±1.4 |
| PIS=10 | 44.9±1.0 | 54.3±1.0 | 58.0±1.0 | 57.1±0.3 | 57.3±1.3 | 56.7±0.6 | 55.5±0.4 |
| PIS=15 | 45.1±0.5 | 54.0±0.7 | 57.1±0.3 | 56.7±1.3 | 57.1±1.0 | 56.6±1.0 | 56.4±0.9 |
| PIS=20 | 45.3±1.0 | 54.2±0.7 | 57.2±1.1 | 56.7±0.9 | 57.1±0.8 | 58.1±0.8 | 55.9±1.6 |
| PIS=25 | 45.0±0.4 | 53.3±0.5 | 56.5±1.5 | 56.9±0.6 | 56.5±0.7 | 56.5±1.5 | 56.3±1.8 |
| PIS=30 | 43.9±0.8 | 52.1±0.8 | 56.0±0.5 | 56.0±0.7 | 55.5±1.0 | 54.7±1.6 | 52.5±1.2 |
| PIS=40 | 42.3±0.8 | 50.3±1.0 | 52.3±1.3 | 58.2±1.5 | 51.5±1.0 | 47.7±1.2 | 49.7±1.3 |
| PIS=50 | 40.1±0.8 | 40.1±0.8 | 40.1±0.8 | 40.1±0.8 | 40.1±0.8 | 40.1±0.8 | 40.1±0.8 |

|  | $\gamma$=0.03 | $\gamma$=0.05 | $\gamma$=0.08 | $\gamma$=0.10 |
|---|---|---|---|---|
| PIS=0 | 56.9±1.7 | 58.5±0.6 | 55.4±0.7 | 55.4±1.8 |
| PIS=2 | 56.1±0.5 | 59.3±0.6 | 56.0±1.1 | 55.9±0.6 |
| PIS=5 | 56.5±0.9 | 59.1±1.2 | 55.8±0.6 | 55.3±0.3 |
| PIS=8 | 56.5±1.5 | 59.5±1.4 | 55.5±1.2 | 56.5±0.6 |
| PIS=10 | 55.9±0.4 | 59.1±0.8 | 56.5±0.7 | 55.6±1.3 |
| PIS=15 | 55.5±0.7 | 60.1±1.0 | 56.3±1.0 | 55.4±1.4 |
| PIS=20 | 50.1±0.4 | 56.3±0.6 | 59.1±0.4 | 55.0±1.4 |
| PIS=25 | 39.7±1.0 | 39.7±1.0 | 39.7±1.0 | 39.7±1.0 |

We validate our approach by progressively enhancing the MGD$^3$ (Santiago et al. (2025)) baseline on SDXL, with full results in Figure 3. For a fair comparison, and because UMAP embeddings cannot be inverted, the virtual cluster centers in all MGD$^3$ variants (b, c, d) are replaced with the nearest real data points, consistent with our main method's protocol.

The results show a clear, cumulative benefit. While MGD$^3$(b) improves on the base SDXL, it still lags far behind the ImageNet-trained SOTA (MGD$^3$(a)). By introducing UMAP for manifold pre-processing (MGD$^3$(c)), we observe a substantial performance boost. Crucially, substituting the prototype selection process with our complete Distribution Discovery pipeline (MGD$^3$(d)) dramatically increases accuracy, surpassing the SOTA (MGD$^3$(a)) in most scenarios. This demonstrates the superior ability of our method to identify the "intrinsic core distribution".

**Hyperparameter Analysis for Distribution Discovery.** The performance of the Distribution Discovery stage is influenced by several hyperparameters, with the HDBSCAN $min\_size$ being the most critical. This parameter dominantly controls the behavior of our multi-strategy post-processing design (detailed in Section 3.1), as it directly governs the source of the final samples: whether they originate from initial HDBSCAN clusters, subsequent splits, or K-Means applied to outliers. As illustrated in Figure 4, this mechanism reveals an optimal setting ($min\_size \in [10, 20]$ for IPC=50) where these strategies act in synergy. While the accuracy gracefully degrades outside this range, our method remains robust, consistently outperforming the SOTA baseline. A detailed region-by-region analysis is provided in Appendix A.5. The analysis of other hyperparameters for this stage, such as the $n\_neighbors$ of UMAP, can be found in Appendix A.6.

**Hyperparameter Analysis for Distribution Alignment.** The Distribution Alignment stage is governed by two hyperparameters: the alignment guidance strength $\gamma$ (from Equation (7)), and Prior Injection Steps (PIS). Jointly, they regulate the critical trade-off between the explicit guidance from $S_r$ and the model's inherent generative prior.

The results in Table 8, yield several key insights. First, even minimal guidance ($\gamma = 0.03$) dramatically outperforms the base SDXL by 14.6%. Second, our method exhibits remarkable robustness, consistently surpassing the SOTA baseline across a wide $\gamma$ range of $[0.05, 0.20]$, provided an appro-

Table 10: Ablation study on feature spaces and clustering methods. All results are for ImageIDC (IPC=10) and are the mean of three runs on ResNetAP10. Best result is in **bold**, second best is in **bold**.

| Dim | Pre | Space | Kmeans | CoDA |
|---|---|---|---|---|
| 3136 | None | VAE | 46.0±1.1 | 45.8±0.9 |
| 768 | None | CLIP | 51.8±1.2 | 52.4±0.7 |
| | PCA | VAE | 46.6±0.6 | 53.0±1.0 |
| 50 | PCA | CLIP | 49.8±0.5 | 53.6±0.9 |
| | | VAE | 51.7±0.8 | 54.6±0.6 |
| | UMAP | CLIP | 54.4±0.7 | **56.0±1.2** |
| | | VAE | 54.8±0.7 | **58.5±0.9** |

Table 11: Ablation study on transferring CoDA to an ImageNet-trained DiT model. All results are the mean of three runs on ResNetAP10. Best results are in **bold**, second best are in **bold**.

| Dataset | IPC | DiT | DiT+MGD3 | DiT+CoDA | SDXL+CoDA |
|---|---|---|---|---|---|
| ImageIDC | 10 | 54.1±0.4 | 55.9±2.1 | **59.4±1.5** | 58.5±0.9 |
| | 20 | 58.9±0.2 | 61.9±0.9 | **65.8±1.2** | 64.6±0.5 |
| | 50 | 64.3±0.6 | 72.1±0.8 | **74.6±1.6** | **77.6±0.6** |
| ImageNette | 10 | 59.1±0.7 | **66.4±2.4** | 65.2±1.5 | **68.8±0.1** |
| | 20 | 64.8±1.2 | 71.2±0.5 | **73.0±1.2** | 72.1±0.2 |
| | 50 | 73.3±0.9 | 79.5±1.3 | 81.4±1.3 | **83.1±0.3** |
| ImageWoof | 10 | 34.7±0.5 | **40.4±1.9** | 39.2±1.3 | **39.2±0.7** |
| | 20 | 41.1±0.8 | **43.6±1.6** | **44.2±0.9** | 42.5±0.6 |
| | 50 | 49.3±0.2 | 56.5±1.9 | 59.2±0.7 | **59.4±1.0** |

priate PIS (PIS $\leq$ 25). Third, allowing the model to freely inject its own prior knowledge during the final generation steps is crucial, as peak accuracy is consistently achieved when PIS $> 0$.

We also find that reducing the inference steps to 25 can even yield higher peak accuracy than 50 steps (as shown in Table 9). This increased efficiency, cutting the 1024×1024 generation time to just 3 seconds (1.3 seconds for 256×256), comes with a reasonable trade-off in hyperparameter robustness. Further analysis is provided in Appendix A.7.

**Transferring CoDA to Different Encoders.** To validate CoDA's versatility, we transferred the Discovery stage to the CLIP (ViT-L/14) feature space. As shown in Table 10, the CoDA framework effectively adapts to diverse feature spaces. Notably, its optimal result in the CLIP space (56.0%) is on par with the previous SOTA MGD[3] (55.9% from Table 1). However, this peak performance is still surpassed by our VAE-space result (58.5%). We attribute this to a "Domain Gap": the CLIP feature space ($Z_{clip}$) is optimized for text-visual alignment, whereas our downstream models, being purely visual, prefer prototypes selected in a space that prioritizes the visual manifold, as further analyzed in Appendix A.11. Furthermore, to guide SDXL generation, it is necessary to retrieve the source images corresponding to $Z_{clip}$, and re-encode them with the VAE to get $Z_{vae}$ ($s_j$). This discrepancy arising from the indirect encoding process degrades peak performance.

**Transferring CoDA to Target-Trained Models.** As shown in Table 11, transferring CoDA to DiT significantly improves DiT's baseline capability. At the IPC=50 setting, it surpasses the original DiT by **10.3%**, **8.1%**, and **9.9%** on ImageIDC, ImageNette, and ImageWoof, respectively. This combination outperforms the previous SOTA (DiT+MGD[3]) in most configurations, and even exceeds our "SDXL+CoDA" pipeline in several cases. This demonstrates that CoDA can be seamlessly adapted to various generative architectures, provided a mechanism for external guidance exists.

We further analyze the optimal hyperparameters and observe a shift between low and high IPC settings. For low IPCs ($\leq$ 20), the optimal parameters ($\gamma = 1.0, \text{PIS} = 15$) allow greater generative freedom. However, at IPC=50, peak performance demands a tighter alignment with the real dataset $\mathcal{R}$ ($\gamma = 0.8, \text{PIS} = 8$). This suggests that while DiT's inherent knowledge is beneficial, it is diversity-limited: as the demand increases to IPC=50, the model saturates and necessitates stronger guidance from the $\mathcal{R}$ to achieve peak performance. Additional visualizations are provided in Section A.14. To further demonstrate CoDA's cross-architecture applicability, we also apply its alignment stage to the Stable Diffusion 3 (SD3) framework, as detailed in Appendix A.16.

# 6 CONCLUSION

In this work, we identify the conceptual flaws in the prevailing dataset distillation paradigm that relies on diffusion models pre-trained on the full target dataset. We introduce **CoDA**, a training-free framework that enables effective DD using only an off-the-shelf text-to-image model to address these drawbacks. Our core strategy involves two stages: first, discovering the dataset's "intrinsic core distribution" using a novel, density-based mechanism, and second, aligning the generated samples with this discovered distribution. Extensive experiments demonstrate that by directly addressing the distributional mismatch, the proposed CoDA achieves performance on par with or even superior to previous methods relying on target-specific pre-training.

ETHICS STATEMENT

We have adhered to the ICLR Code of Ethics throughout this research. Our work is built upon publicly accessible research assets, including standard academic datasets (ImageNet, Places365) and the open-source model SDXL, all used in accordance with their respective licenses.

A significant ethical advantage of our dataset distillation approach lies in the generation of entirely synthetic data. By creating a distilled set that contains no real-world images, our method inherently provides strong protections for the privacy of individuals and the copyright of original creators.

While our method offers these robust privacy and copyright advantages, we recognize the broader challenges within the field of generative AI. Foundational models like SDXL, due to their training on web-scale data, may inadvertently reflect societal biases. Our current framework focuses on faithfully reproducing the source dataset's distribution and does not include an explicit de-biasing mechanism. Therefore, addressing the potential for bias propagation from the source data to the distilled set remains an important consideration and a valuable direction for future research.

In summary, we believe our method represents a positive step towards creating more efficient, secure, and privacy-conscious machine learning workflows.

REPRODUCIBILITY STATEMENT

To support the reproducibility of our research, we have provided key materials in the supplementary file attached to this submission. This includes a representative subset of the datasets distilled by our CoDA framework for several key benchmarks, along with the complete evaluation code required to validate their performance.

ACKNOWLEDGEMENTS

This project is supported by the Ministry of Education, Singapore, under its Academic Research Fund Tier 2 (Award Number: MOE-T2EP20122-0006).

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

# A APPENDIX

## A.1 THE USE OF LARGE LANGUAGE MODELS (LLMS)

In line with ICLR 2026 guidelines, we report the use of a large language model in preparing this paper. Its function was strictly as a writing aid, assisting in the refinement of language and enhancement of textual clarity. We confirm that the LLM did not contribute to the conceptualization of the research, the design of the methodology, or the analysis of the results. The authors are solely responsible for all scientific contributions presented.

## A.2 RELATED WORK

**Traditional Methods.** Early paradigms in Dataset Distillation (DD) were often hampered by prohibitive computational demands. This limitation largely stemmed from their reliance on complex bi-level optimization frameworks Finn et al. (2017). Mainstream approaches, including Performance Matching (Wang et al. (2018); Nguyen et al. (2020); Zhou et al. (2022)) and Parameter Matching (Zhao et al. (2020); Cazenavette et al. (2022); Cui et al. (2023)), are emblematic of this dependency. Such frameworks necessitate backpropagation through long, unrolled optimization trajectories, imposing substantial computational and memory overheads that severely restrict their scalability. Consequently, the application of these methods has been predominantly confined to small-scale, low-resolution datasets such as CIFAR-10/100 ($32 \times 32$ pixels) and MNIST ($28 \times 28$ pixels).

**Methods Based on Target-Trained Diffusion Models.** To overcome the scalability issues of traditional methods, a recent wave of research has shifted focus towards leveraging powerful generative models for dataset distillation. However, many prominent methods within this paradigm, such as Minimax Gu et al. (2024), LD3M Moser et al. (2024), IGD Chen et al. (2025) and MGD[3] Santiago et al. (2025), paradoxically require a diffusion model pre-trained on the full target dataset, undermining the very purpose of DD and incurring prohibitive costs.

MGD[3] serves as a case in point. It first encodes the target dataset into a VAE latent space and then applies K-Means clustering to identify virtual 'mode' prototypes. Guidance is then applied during denoising by directly adding a heuristic correction to the predicted clean sample ($\hat{z}_0$), steering it towards these prototypes. This approach fundamentally differs from our CoDA framework, which instead modifies the predicted noise ($\epsilon_\theta$). CoDA achieves this through a theoretically-grounded mapping that translates a desired interpolation in the data space into a self-consistent correction in the noise space.

**Methods Based on General Text-to-Image Diffusion Models.** A few "logically correct" methods, such as the notable D[4]M Su et al. (2024), turn to general text-to-image models like Stable Diffusion. The process begins by encoding the target dataset into a VAE latent space and then applying Mini-batch K-Means to identify representative prototypes from the real data points closest to the cluster centers. These selected prototypes are then noised to serve as the initial latents for the reverse diffusion process.

Crucially, no further guidance is applied after this initialization. This hands-off approach gives the model's powerful, general-purpose priors free rein, which in turn can cause the generation to drift from the target distribution, overriding the valuable information in the initial prototypes and ultimately leading to suboptimal performance. This stands in stark contrast to our CoDA framework, which begins from pure Gaussian noise but applies continuous, targeted guidance throughout the entire generation process to ensure alignment with the core distribution.

### A.3    LIMITATIONS AND FUTURE WORK

While the CoDA framework exhibits strong robustness across a wide range of parameters, in pursuit of optimal performance, we observed a subtle drift in its key Distribution Discovery hyperparameters: UMAP's $n\_neighbors$ and HDBSCAN's $min\_cluster\_size$. The root cause is that each dataset has its unique "intrinsic core distribution", which inevitably influences our density-based discovery mechanism. For instance, under the IPC=10 setting, the optimal configuration for the ImageNet-1K and ImageNet-A through E subsets was $min\_cluster\_size$=55 and $n\_neighbors$=85. In contrast, the optimal configuration for ImageIDC, ImageNette, and Places365 shifted to $min\_cluster\_size$=55 and $n\_neighbors$=90. Although this difference is minimal and the Distribution Discovery stage can be fully optimized via grid search on a CPU, this approach is ultimately not the most elegant or efficient solution and represents a key area we aim to improve. A promising future direction is to develop a meta-learning model that automatically selects optimal hyperparameters by analyzing the intrinsic variance and density of the target dataset's distribution.

### A.4    FULL PSEUDOCODE FOR THE POST-PROCESSING SCHEME

This section provides the full pseudocode for our post-processing scheme, detailed in Algorithm 2. This algorithm is designed to address the gap between the initial number of representative samples (M) and the target number IPC. It resolves this discrepancy by iteratively identifying new samples, either by splitting existing "mother-clusters" or by applying K-Means to the population of outliers until the sample count matches the IPC.

### A.5    ANALYSIS OF POST-PROCESSING COMPONENTS

This section provides the detailed, region-by-region analysis of the post-processing components, as summarized in the main paper. We examine how the core HDBSCAN hyperparameter, $min\_size$, governs the source distribution of the final distilled samples, which in turn dictates the final model performance. The analysis is supported by the data presented in Figure 4.

**Region 1** ($min\_size < 5$)**.** When the threshold is too low, HDBSCAN produces a multitude of trivial, fine-grained clusters, causing the initial count $M$ to vastly exceed the $IPC$ budget. This forces the post-processing scheme to only perform simple truncation. This approach effectively discards the contributions of all original data points belonging to the remaining $M - IPC$ clusters, leading to significant information loss. Consequently, performance in this region gradually rises as $min\_size$ increases, as this higher threshold prevents the fragmentation of the data, effectively

---

**Algorithm 2** Post-processing Scheme for IPC Matching

---

**Input:** Initial representative samples $S_0 = \{s_1, \ldots, s_M\}$; Clusters $C_0 = \{c_1, \ldots, c_M\}$.
**Parameters:** IPC, $min\_cluster\_size$.
**Ensure:** Final representative set $S$ with $|S| = $ IPC.

1: $S \leftarrow S_0, C \leftarrow C_0$
2: **if** $|S| \geq$ IPC **then**
3:      $S \leftarrow$ Samples corresponding to the IPC largest clusters in $C$.
4: **else**
5:      $W \leftarrow \{c \in C \mid |c| > 2 \times min\_cluster\_size\}$
6:      **while** $|S| <$ IPC & $W \neq \emptyset$ **do**              ▷ Strategy 1: Normally Split the valid clusters
7:          $c_{\text{mother}} \leftarrow$ Select and remove the largest cluster from $W$
8:          SPLITCLUSTER($c_{\text{mother}}$, $min\_cluster\_size$)
9:      **end while**
10:      **if** $|S| <$ IPC **then**
11:          $\mathcal{N} \leftarrow$ the set of all outliers
12:          $k_{\text{needed}} \leftarrow$ IPC $- |S|$
13:          **if** $|\mathcal{N}| \geq k_{\text{needed}}$ **then**             ▷ Strategy 2: Cluster the outliers if available
14:              Run K-Means on $\mathcal{N}$ to get $k_{\text{needed}}$ new clusters $C_{\text{outliers}}$ and samples $S_{\text{outliers}}$.
15:              $C \leftarrow C \cup C_{\text{outliers}}$ and $S \leftarrow S \cup S_{\text{outliers}}$.
16:          **else**              ▷ Strategy 3: Force split the viable clusters
17:              $W \leftarrow \{c \in C \mid |c| > 2 \times min\_cluster\_size\}$
18:              **while** $|S| <$ IPC **do**
19:                  $c_{\text{mother}} \leftarrow$ Select and remove the largest cluster from $W$
20:                  FORCEDSPLIT($c_{\text{mother}}$, $min\_cluster\_size$)
21:              **end while**
22:          **end if**
23:      **end if**
24: **end if**
25: **return** $S$

26: **procedure** SPLITCLUSTER($c_{\text{mother}}$, $min\_cluster\_size$)
27:      Run HDBSCAN on $c_{\text{mother}}$ with $min\_cluster\_size$ to find $k$ sub-clusters as $S_{\text{cand}}$
28:      If $k < 2$ **return False**
29:      $S \leftarrow S \setminus \{s_{\text{mother}}\}$ and $C \leftarrow C \setminus \{c_{\text{mother}}\}$
30:      Iterate all pairs $(s_i, s_j)$ from $S_{\text{cand}}$ as initial seeds for K-Means ($k = 2$) on $c_{\text{mother}}$, select the pair yielding the minimum resulting inertia, denoted as $\{c_{\text{new1}}, c_{\text{new2}}\}$ and $\{s_{\text{new1}}, s_{\text{new2}}\}$
31:      $C \leftarrow C \cup \{c_{\text{new1}}, c_{\text{new2}}\}$ and $S \leftarrow S \cup \{s_{\text{new1}}, s_{\text{new2}}\}$
32:      $W \leftarrow W \cup \{c \in \{c_{\text{new1}}, c_{\text{new2}}\} \mid |c| > 2 \times min\_cluster\_size\}$
33:      **return True**
34: **end procedure**

35: **procedure** FORCEDSPLIT($c_{\text{mother}}$, $min\_size$)
36:      **while** SPLITCLUSTER($c_{\text{mother}}$, $min\_size$) = **False** & $min\_size > 2$ **do**
37:          $min\_size \leftarrow \max(2, \lfloor min\_size \times 0.75 \rfloor)$             ▷ Relax the constraint
38:      **end while**
39: **end procedure**

---

consolidating the data points from trivial (and previously discarded) clusters into the larger, stable clusters that are ultimately retained.

**Region 2** ($min\_size \in [10, 20]$)**.** This is the optimal operational range for our method. In this region, $M < IPC$, which activates our complete post-processing pipeline. Here, each strategy fulfills its designated role: The first HDBSCAN (Init Clusters) is responsible for capturing the primary, high-density features; Strategies 1 and 3 (Splits) are responsible for refining these "large features" into more granular details; and Strategy 2 (K-Means) serves as the perfect complement, efficiently extracting the remaining feature details from the outliers that HDBSCAN intentionally ignores. The synergy of all components achieves peak performance in this specific region.

**Region 3** ($min\_size > 25$). As the threshold continues to increase, HDBSCAN becomes overly conservative and the number of initial clusters, $M$, drops sharply. In the [30,50] range, this often causes Strategy 1 to fail, forcing the pipeline to rely heavily on Strategy 3, which partially salvages performance. However, as $M$ approaches zero, Strategy 3 also fails due to the lack of any "mother-clusters" to operate on. This causes the pipeline to degenerate entirely into solely activating Strategy 2—running a single-step K-Means on all data points—leading to a sharp performance collapse.

It is critical to note, however, that while performance degrades relative to its peak at $min\_size = 15$, our method's accuracy across this vast operational range, $min\_size$ up to 100, consistently and significantly outperforms the SOTA (Santiago et al. (2025)).

### A.6 Impact of Key Hyperparameters in Distribution Discovery

The quality of $\mathcal{R}$ is primarily governed by two key hyperparameters: $n\_neighbors$ in UMAP and $min\_cluster\_size$ in HDBSCAN. The $n\_neighbors$ controls the balance between preserving local versus global data structures while the $min\_cluster\_size$ dictates the granularity of the clustering. As illustrated by the performance heatmaps for ImageIDC in Figure 5, the optimal space for these two parameters shifts with IPC. For IPC=10, peak performance is achieved with a large $min\_cluster\_size$ (>50) and a large $n\_neighbors$ (>85). In contrast, for IPC=50, it shifts to a small $min\_cluster\_size$ ($< 20$) and a small $n\_neighbors$ ($< 50$).

This behavior is intuitive. At a low IPC, a large $min\_cluster\_size$ prevents the fragmentation of major clusters, ensuring that the selected representative samples encapsulate features common across a broad spectrum of the data. This, in turn, requires a large $n\_neighbors$ to accurately model the global structure. Conversely, at a high IPC, the goal is to obtain more fine-grained clusters so that each representative can embody more specialized, local features. This necessitates a small $n\_neighbors$, which effectively narrows the "field of view" for each point and compels UMAP to preserve distinct local structures.

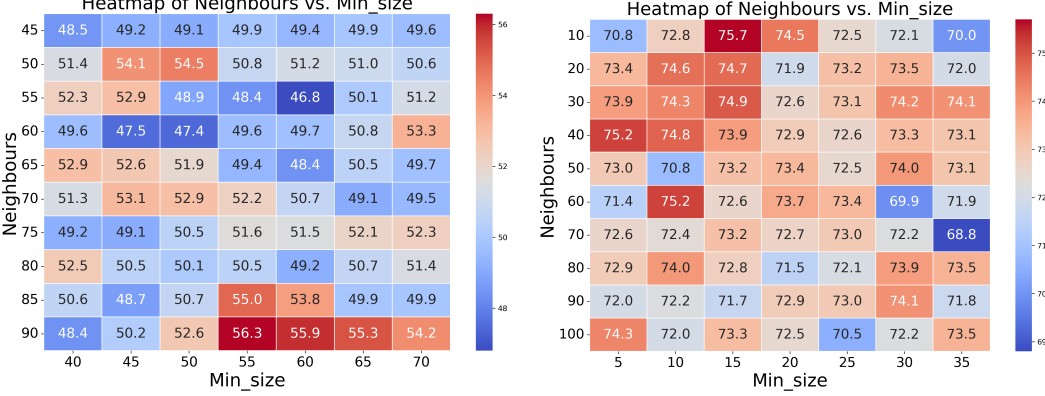

Figure 5: Performance heatmaps for hyperparameter sensitivity on ImageIDC. The optimal region shifts from large $n\_neighbors$ values and $min\_cluster\_size$ for IPC=10 (left) to low values for IPC=50 (right).

### A.7 Impact of Key Hyperparameters in Distribution Alignment

**Full Results for $\gamma$ and PIS Sensitivity Analysis** In this section, we provide the complete data table used for the sensitivity analysis of the guidance strength $\gamma$ and Prior Injection Steps (PIS). The results are presented in Table 12. Furthermore, we test whether the optimal hyperparameter space identified at IPC=10 is transferable to a high-IPC setting (IPC=50). We select a core range of effective parameters, with guidance strength $\gamma \in 0.05, 0.08, 0.10, 0.13, 0.15$ and Prior Injection Steps $PIS \in 0, 5, 10, 15, 20, 25$. As presented in Table 13 and Figure 6. The performance in all tested cases decisively surpasses the SOTA baseline of 72.1%, with the optimal parameter setting achieving a peak performance of 77.6% (+5.5% over SOTA).

Table 12: Performance across different $\gamma$ and PIS values. Cells highlighted in yellow indicate performance greater than the SOTA of 55.9.

|  | $\gamma = 0.01$ | $\gamma = 0.03$ | $\gamma = 0.05$ | $\gamma = 0.08$ | $\gamma = 0.10$ | $\gamma = 0.13$ | $\gamma = 0.15$ | $\gamma = 0.18$ | $\gamma = 0.20$ |
|---|---|---|---|---|---|---|---|---|---|
| PIS=0 | 45.7±1.1 | 54.7±1.2 | 57.7±0.6 | 56.1±1.0 | 55.9±1.5 | 57.1±1.0 | 57.1±1.5 | 56.1±0.5 | 55.9±1.3 |
| PIS=5 | 45.5±1.0 | 54.6±0.4 | 55.8±0.2 | 56.0±0.8 | 58.5±0.9 | 57.3±0.4 | 57.0±1.1 | 56.3±1.2 | 56.9±1.4 |
| PIS=10 | 44.9±1.0 | 54.3±1.0 | 58.0±1.0 | 57.1±0.3 | 57.3±1.3 | 57.1±0.8 | 56.7±0.6 | 56.0±0.4 | 55.5±0.4 |
| PIS=15 | 45.1±0.5 | 54.0±0.7 | 57.1±0.3 | 56.7±1.3 | 57.1±1.0 | 56.5±1.3 | 56.6±1.0 | 56.4±0.9 | 56.4±0.9 |
| PIS=20 | 45.3±1.0 | 54.2±0.7 | 57.2±1.1 | 56.7±0.9 | 57.1±0.8 | 56.6±0.4 | 58.1±0.8 | 56.1±0.6 | 55.9±1.6 |
| PIS=25 | 45.0±0.4 | 53.3±0.5 | 56.5±1.5 | 56.9±0.6 | 56.5±0.7 | 56.2±0.7 | 56.5±1.5 | 55.7±2.0 | 56.3±1.8 |
| PIS=30 | 43.9±0.8 | 52.1±0.8 | 56.0±0.5 | 56.0±0.7 | 55.5±1.0 | 55.4±0.9 | 54.7±1.6 | 52.1±1.2 | 52.5±1.2 |
| PIS=35 | 43.1±0.6 | 50.9±1.3 | 54.1±0.3 | 56.6±1.2 | 52.9±1.4 | 53.9±2.1 | 53.5±1.0 | 52.5±1.5 | 52.8±2.0 |
| PIS=40 | 42.3±0.8 | 50.3±1.0 | 52.3±1.3 | 58.2±1.5 | 51.5±1.0 | 50.4±0.6 | 47.7±1.2 | 49.7±1.1 | 49.7±1.3 |
| PIS=45 | 38.9±1.3 | 44.8±0.9 | 48.1±1.3 | 54.1±1.5 | 51.4±1.3 | 49.4±0.9 | 50.3±0.8 | 47.1±0.6 | 47.7±1.6 |
| PIS=50 | 40.1±0.8 | 40.1±0.8 | 40.1±0.8 | 40.1±0.8 | 40.1±0.8 | 40.1±0.8 | 40.1±0.8 | 40.1±0.8 | 40.1±0.8 |

Table 13: Robustness of key hyperparameters ($\gamma$ and PIS) on ImageIDC at IPC=50. All results are reported as the mean $\pm$ standard deviation over three independent runs. Notably, the performance in **all tested cases decisively surpasses the SOTA baseline** of 72.1%, with the optimal parameter setting achieving a peak performance of 77.6% (**+5.5% over SOTA**).

|  | $\gamma$=0.05 | $\gamma$=0.08 | $\gamma$=0.10 | $\gamma$=0.13 | $\gamma$=0.15 |
|---|---|---|---|---|---|
| **PIS=0** | 75.1 ± 0.8 | 75.5 ± 1.0 | 74.6 ± 0.6 | 75.9 ± 0.6 | 75.5 ± 1.0 |
| **PIS=5** | 74.5 ± 0.9 | 75.7 ± 0.9 | 75.0 ± 0.8 | 75.9 ± 1.1 | 75.5 ± 0.6 |
| **PIS=10** | 74.9 ± 0.2 | 76.5 ± 0.5 | 75.5 ± 1.4 | 76.4 ± 0.2 | 75.1 ± 1.0 |
| **PIS=15** | 75.1 ± 0.2 | 75.8 ± 0.2 | 76.5 ± 1.2 | 76.0 ± 1.4 | 75.7 ± 1.0 |
| **PIS=20** | 75.1 ± 0.4 | 77.6 ± 0.6 | 76.5 ± 1.2 | 75.5 ± 0.8 | 75.3 ± 0.8 |
| **PIS=25** | 74.5 ± 1.0 | 74.5 ± 1.2 | 76.6 ± 0.6 | 74.2 ± 0.9 | 73.9 ± 1.1 |

**Sampler Configuration.** In this section, we analyze the impact of different sampler configurations. We use the DPM++ Karras sampler and investigate its performance at a low number of inference steps. Specifically, we set the total number of steps to 25 and evaluate performance on the ImageIDC (IPC=10) dataset across a range of $\gamma$ and PIS values. The results are presented in Table 14.

The results demonstrate that our method remains highly effective even with a significantly reduced number of inference steps. At just 25 steps, our method maintains a decisive advantage over the SOTA baseline. Notably, a peak performance of 60.1% is achieved at $\gamma$=0.05 and PIS=15, which not only surpasses the SOTA by **4.2%** but also exceeds the optimal result we obtained with 50 inference steps.

However, this efficiency comes with a trade-off in robustness. As a comparison between Table 14 (25 steps) and Table 8 (50 steps) reveals, the range of hyperparameter settings that outperform the SOTA is considerably narrower at 25 steps than the broad robust region observed at 50 steps. Given this trade-off, we prioritized robustness and therefore chose to use 50 inference steps for our main experiments.

Table 14: Performance across different $\gamma$ and PIS values when using the DPM++ Karras sampler with only 25 inference steps. Cells highlighted in yellow indicate performance greater than the SOTA of 55.9.

|  | $\gamma$=0.01 | $\gamma$=0.03 | $\gamma$=0.05 | $\gamma$=0.08 | $\gamma$=0.10 | $\gamma$=0.13 | $\gamma$=0.15 | $\gamma$=0.18 | $\gamma$=0.20 |
|---|---|---|---|---|---|---|---|---|---|
| **PIS=0** | 41.8±0.9 | 56.9±1.7 | 58.5±0.6 | 55.4±0.7 | 55.4±1.8 | 55.4±0.2 | 55.1±1.0 | 54.7±1.5 | 53.6±0.4 |
| **PIS=2** | 42.1±0.3 | 56.1±0.5 | 59.3±0.6 | 56.0±1.1 | 55.9±0.6 | 55.7±1.2 | 54.9±1.4 | 55.0±0.5 | 54.0±1.3 |
| **PIS=5** | 41.8±0.4 | 56.5±0.9 | 59.1±1.2 | 55.8±0.6 | 55.3±0.3 | 55.6±2.2 | 56.2±1.6 | 54.3±2.1 | 55.3±0.6 |
| **PIS=8** | 41.8±0.8 | 56.5±1.5 | 59.5±1.4 | 55.5±1.2 | 56.5±0.6 | 55.5±1.2 | 54.5±0.9 | 54.9±0.5 | 56.1±0.6 |
| **PIS=10** | 42.2±1.9 | 55.9±0.4 | 59.1±0.8 | 56.5±0.7 | 55.6±1.3 | 56.1±0.3 | 54.7±0.8 | 55.3±1.0 | 55.2±1.1 |
| **PIS=15** | 40.5±1.3 | 55.5±0.7 | 60.1±1.0 | 56.3±1.0 | 55.4±1.4 | 54.5±1.3 | 53.4±0.5 | 51.5±1.0 | 50.9±1.0 |
| **PIS=20** | 39.8±0.2 | 50.1±0.4 | 56.3±0.6 | 59.1±0.4 | 55.0±1.4 | 54.2±0.7 | 52.4±2.1 | 49.7±0.6 | 45.7±0.8 |
| **PIS=25** | 39.7±1.0 | 39.7±1.0 | 39.7±1.0 | 39.7±1.0 | 39.7±1.0 | 39.7±1.0 | 39.7±1.0 | 39.7±1.0 | 39.7±1.0 |

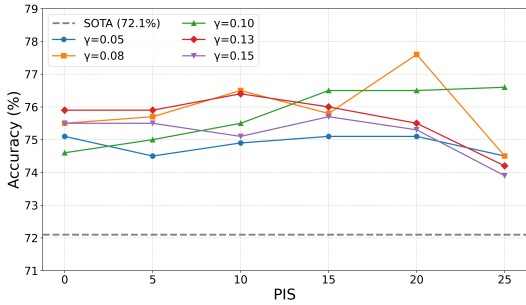

Figure 6: Performance on ImageIDC at IPC=50 across varying $\gamma$ and PIS settings.

## A.8 PERFORMANCE EVALUATION ACROSS DIFFERENT RESOLUTIONS

This section provides a detailed cross-architecture performance evaluation of our method on the ImageNet-A through ImageNet-E subsets across three different resolutions: **128x128**, **256x256**, and **1024x1024**. The cross-architecture performance is evaluated with four distinct downstream architectures: AlexNet (Krizhevsky et al. (2012)), VGG11 (Simonyan & Zisserman (2014)), ResNet18 (He et al. (2016)) and ViT (Dosovitskiy et al. (2020)). These results demonstrate the scalability and consistent superiority of our approach.

**Results at 128x128 Resolution.** As shown in Table 15, our method's performance at **128x128** resolution decisively surpasses previous SOTA methods. Specifically, on the ImageNet-A through E subsets, our method achieves an average improvement of **12.7%**, **17.5%**, **14.6%**, **9.9%**, and **14.5%**, respectively. The detailed performance breakdown for each downstream architecture is provided in Table 16.

Table 15: Comparing our method against SOTA baselines on ImageNet-A to ImageNet-E at **128x128** resolution with IPC=10. The best results are in **bold red** and the second best are in **bold**.

| Method | Alg. | ImNetA | ImNetB | ImNetC | ImNetD | ImNetE |
|--------|------|--------|--------|--------|--------|--------|
| Pixel | DC | 52.3±0.7 | 45.1±8.3 | 40.1±7.6 | 36.1±0.4 | 38.1±0.4 |
|        | DM | 52.6±0.4 | 50.6±0.5 | 47.5±0.7 | 35.4±0.4 | 36.0±0.5 |
| GLaD | DC | 53.1±1.4 | 50.1±0.6 | 48.9±1.1 | 38.9±1.0 | 38.4±0.7 |
|      | DM | 52.8±1.0 | 51.3±0.6 | 49.7±0.4 | 36.4±0.4 | 38.6±0.7 |
| LD3M | DC | 55.2±1.0 | 51.8±1.4 | **49.9±1.3** | **39.5±1.0** | 39.0±1.3 |
|      | DM | **57.0±1.3** | **52.3±1.1** | 48.2±4.9 | **39.5±1.5** | **39.4±1.8** |
| Ours | - | **69.7±1.1** | **69.8±0.7** | **64.5±1.5** | **49.4±1.1** | **53.9±1.3** |

Table 16: Detailed performance of our method across different architectures. Results are reported for IPC=10 at **128x128** resolution.

| Architecture | ImNet-A | ImNet-B | ImNet-C | ImNet-D | ImNet-E |
|--------------|---------|---------|---------|---------|---------|
| ResNet18 | 74.3±1.0 | 78.7±0.8 | 73.9±2.4 | 54.0±0.6 | 58.5±1.8 |
| AlexNet | 70.3±1.6 | 70.5±0.5 | 65.9±1.4 | 50.0±0.9 | 58.9±0.7 |
| VGG11 | 68.6±0.5 | 69.6±0.6 | 61.3±1.4 | 49.6±1.6 | 53.8±1.1 |
| ViT | 65.7±1.2 | 60.3±0.7 | 56.7±0.6 | 44.1±1.1 | 44.2±1.7 |
| **Mean** | **69.7±1.1** | **69.8±0.7** | **64.5±1.5** | **49.4±1.1** | **53.9±1.3** |

**Results at 256x256 Resolution.** The comparison of our method's mean performance against SOTA baselines at this resolution is presented in the main paper (Table 3). This section provides the detailed breakdown of our method's performance on each of the four downstream architectures, with the full results available in Table 17.

Table 17: Detailed cross-architecture test accuracies (%). The results for each architecture are reported as the mean $\pm$ standard deviation over five independent runs, using each distilled dataset at **256x256** resolution.

| Architecture | ImNet-A | ImNet-B | ImNet-C | ImNet-D | ImNet-E |
|---|---|---|---|---|---|
| ResNet18 | 78.1 ± 0.5 | 80.6 ± 0.7 | 72.0 ± 1.3 | 55.0 ± 1.8 | 62.2 ± 0.6 |
| AlexNet | 74.4 ± 0.7 | 74.5 ± 1.5 | 66.9 ± 0.6 | 52.1 ± 0.2 | 61.9 ± 0.6 |
| VGG11 | 69.5 ± 0.5 | 75.4 ± 2.0 | 65.1 ± 0.8 | 51.0 ± 0.4 | 55.9 ± 0.5 |
| ViT | 70.0 ± 1.1 | 68.1 ± 0.5 | 60.3 ± 1.7 | 48.1 ± 0.7 | 50.6 ± 0.9 |
| **Mean** | **73.0 ± 0.7** | **74.7 ± 1.2** | **66.1 ± 1.1** | **51.6 ± 0.8** | **57.7 ± 0.7** |

**Results at 1024x1024 Resolution.** We are the first to establish a dataset distillation baseline at a high resolution of **1024x1024**, opening a new avenue for research in this area. The detailed results for this track are presented in the main paper (Table 5).

## A.9 CASE STUDY ON ZERO-COST DOMAIN CHANGE: DRIVE

To further demonstrate the zero-cost domain change capability of CoDA, we present a case study transferring the framework to the DRIVE (Digital Retinal Images for Vessel Extraction) dataset (Staal et al. (2004)). Qualitative results are presented in Figure 7, where the top row displays the real representative samples ($\mathcal{R}$) identified by our "Discovery" stage, and the bottom row showcases the final synthesized images ($\mathcal{G}$) from the "Alignment" stage.

This visually confirms that CoDA can seamlessly adapt to a new and highly specialized domain without any additional manual intervention. This example powerfully illustrates the core limitation of target-trained methods. To our knowledge, no publicly available diffusion model pre-trained on DRIVE exists. Therefore, a target-trained approach would be forced to first train or fine-tune a generative model on DRIVE before the distillation could even begin.



Figure 7: Qualitative results of CoDA's zero-cost domain transfer to the DRIVE dataset. **Top row:** The set of real representative samples ($\mathcal{R}$) identified by the Distribution Discovery stage. **Bottom row:** The final set of guided images ($\mathcal{G}$) generated by the Distribution Alignment stage.

## A.10 FURTHER COMPUTATIONAL EFFICIENCY ANALYSIS

We provide a comprehensive analysis of CoDA's efficiency, including end-to-end runtime, resource utilization, and a comparison against training-free ($D^4M$) and target-trained ($MGD^3$) SOTA methods, as shown in Table 18.

The "Encode" stage utilizes two different VAE models. The Target-Trained DiT $MGD^3$ employs "sd-vae-ft-mse", while others utilize "sdxl-vae". However, our practical tests, using a single RTX 4090, show no significant difference in runtime efficiency for this one-time Lanczos resizing and VAE encoding or VRAM usage for this stage.

For the "Discovery" stage, we implement all algorithms exclusively on a 16-vCPU Intel Xeon Gold 6430. The specific performance breakdown for CoDA is detailed in Table 19. As observed, the computational cost of CoDA's "Discovery" stage is dominated by UMAP preprocessing, which is proven to be indispensable for performance in Appendix A.11. As shown in Table 18, if CoDA's UMAP is replaced with simple PCA, there is no significant difference in the discovery stage speed compared to SOTA methods. This indicates that CoDA's entire clustering logic, while complex, is highly efficient in low-dimensional space and does not consume significantly more resources than a

Table 18: End-to-end efficiency and resource comparison (ImageNet-1K, IPC=50). "Change Domain" is the cost to adapt to a new dataset (e.g., Places365).

| Type | Step | Method | Encode (VRAM) | Discovery (RAM) | Result ($\mathcal{R}$) | Generate (VRAM) | End-to-End Time | Result ($\mathcal{G}$) | Change Domain |
|---|---|---|---|---|---|---|---|---|---|
| General-model SDXL | 50 step | D4M | 1.5h (1553MiB) | 0.42h (1,996 MiB) | - | 2.7s/img (12323MiB) | 39.4h | 55.2±0.1 | 0 |
| | | MGD3 | 1.5h (1553MiB) | 0.42h (1,996 MiB) | - | 2.7s/img (12323MiB) | 39.4h | 57.0±0.2 | 0 |
| | | CoDA(PCA) | 1.5h (1553MiB) | 0.42h (2,019 MiB) | 55.4±0.2 | 2.7s/img (12323MiB) | 39.4h | 57.8±0.2 | 0 |
| | | CoDA(UMAP) | 1.5h (1553MiB) | 2.17h (2,144 MiB) | 58.2±0.2 | 2.7s/img (12323MiB) | 41.2h | 60.4±0.2 | 0 |
| | 25 step | CoDA(UMAP) | 1.5h (1553MiB) | 2.17h (2,144 MiB) | 58.2±0.2 | 1.3s/img (12323MiB) | 21.7h | 60.1±0.2 | 0 |
| Target-Trained DiT | 50 step | MGD3 | 1.5h (1553MiB) | 0.42h (1,996 MiB) | - | 1.2s/img (2796MiB) | 18.6h | 60.2±0.2 | multi-thousand TPU-day |

Table 19: Breakdown of the Distribution Discovery stage (224x224, on ImageIDC IPC=50).

| Type | Preprocess-time | Initial HDBSCAN | Strategy 1: SplitCluster | Strategy 2: Clustering Outliers | Strategy 3: ForcedSplit |
|---|---|---|---|---|---|
| UMAP | 72.273 s | 0.969 s | 1.156 s | 0.083 s | 0.200 s |
| PCA | 0.740 s | 1.592 s | 0.002 s | 0.157 s | 0.000 s |

direct K-Means. Notably, only CoDA produces a usable set of representative real images, $\mathcal{R}$, at this stage, which already possesses high performance. As shown in the main paper (Table 6), in some cases, CoDA($\mathcal{R}$) already surpasses the final generated datasets of some SOTA methods.

In the "Generate" stage, the variance in speed stems from the generative model architectures. While DiT benefits from faster generation, we demonstrate in the main paper (Table 9) and Appendix A.7 that configuring our sampler to 25 steps maintains or even improves optimal performance. With this setting, CoDA's efficiency approaches that of the fastest MGD[3] without performance degradation. Crucially, it must be emphasized that DiT's speed advantage merely amortizes the massive pre-training cost required for each domain. In contrast, as demonstrated in the main paper (Table 7) and Appendix A.9, CoDA is completely general and its cost to change domains is zero.

## A.11 VISUALIZATION AND ANALYSIS OF FEATURE SPACES AND THE ROLE OF UMAP

To further investigate the distinctions between the CLIP and VAE spaces, and to underscore the indispensability of UMAP preprocessing in the CoDA pipeline, we present comprehensive visualizations alongside quantitative metrics. We employ t-SNE (configured with "n_components=2", "perplexity=40", "random_state=42") to project high-dimensional feature spaces into 2D for visual analysis.

To quantify the "compactness" of the resulting manifolds visualized in Figure 8, we report the Mean $\sigma$ (the average kernel bandwidth from t-SNE) in Table 20. As shown in Figure 8(a), the original VAE space is highly diffuse, exhibiting a large Mean $\sigma$ of 12.57. This confirms our hypothesis (see Figure 1c) that raw VAE spaces are ill-suited for density-based clustering. Comparing the 768-dimensional VAE (c) and CLIP (b) spaces, both the visualization and the Mean $\sigma$ (8.06) are remarkably similar. Although inter-class separation is marginally improved, data points remain heavily intermixed, and intra-class compactness is insufficient. This observation challenges the assumption that CLIP is inherently superior solely due to its semantic alignment. Furthermore, as illustrated in (d), simply applying linear dimensionality reduction (PCA) to CLIP (50d) fails to yield a structured manifold; the Mean $\sigma$ drops to 6.33 primarily due to dimensionality reduction, yet the visualization reveals even severe class overlap.

In sharp contrast, applying UMAP (with "min_dist=0.0") in (e) dramatically transforms the manifold. The Mean $\sigma$ plummets to 0.29. This quantitatively validates that our choice of UMAP is both necessary and effective. UMAP successfully disentangles the mixed feature space into distinct islands, significantly maximizing inter-class distance while enhancing intra-class compactness. This transformation establishes a robust foundation for the subsequent clustering task, explaining why the UMAP+CoDA combination achieves SOTA performance (see Table 10).

Furthermore, we demonstrate UMAP's robustness on ImageWoof, a dataset characterized by extremely high inter-class similarity (containing breeds such as Australian terrier, Samoyed, Beagle, etc.). As shown in Figure 9, the original VAE space (a) is highly diffuse (Mean $\sigma$: 10.84), and standard PCA (b) fails to induce meaningful separation (Mean $\sigma$: 7.92). In stark contrast, the UMAP

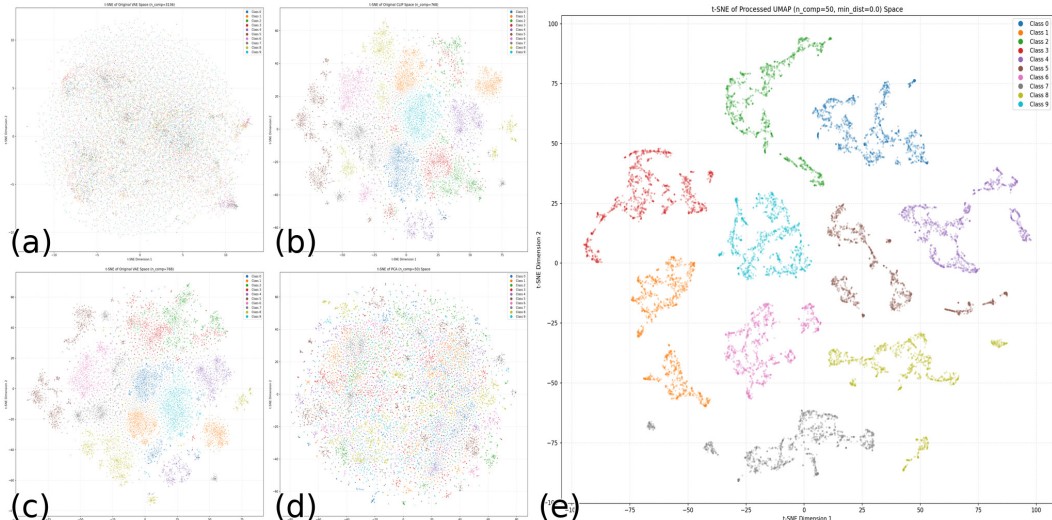

Figure 8: t-SNE visualization of different feature spaces for ImageIDC. (a) Original VAE (3136d). (b) Original CLIP (768d). (c) VAE (PCA to 768d). (d) CLIP (PCA to 50d). (e) CLIP (UMAP to 50d, using our pipeline's "min_dist=0.0" setting).

Table 20: Manifold compactness metrics for the spaces visualized in Figure 8.

| Space | Dimension | Mean $\sigma$ |
|---|---|---|
| Original VAE | 3136 | 12.57 |
| Original CLIP | 768 | 8.06 |
| VAE (PCA) | 768 | 8.06 |
| CLIP (PCA) | 50 | 6.33 |
| CLIP (UMAP, "min_dist=0.0") | 50 | **0.29** |

used by CoDA (c) successfully reshapes this challenging manifold, creating dense, compact, and well-separated clusters (Mean $\sigma$: 0.30). This confirms that our UMAP-based preprocessing is a critical component that remains effective even for fine-grained datasets.

### A.12 LIMITATIONS OF GENERAL-PURPOSE MODELS IN DD TASKS

As illustrated in Figure 10, using only the class name as a prompt for a general-purpose model like SDXL leads to highly inconsistent results. For classes with unambiguous names, such as 'garden spider' or 'gyromitra', the generated images are often acceptable. However, for polysemous words like 'bonnet', the model's generic, web-scale priors cause it to completely deviate from the target dataset's context. For instance, in the figure, we can see that out of ten generated images for the 'bonnet' class, only eight depict the required 'hat'. The other two, showing a bird and a car (referring to other meanings of the word), would introduce significant noise and confusion during subsequent model training. This highlights the necessity of our additional guidance to correct these semantic misunderstandings and align the model with the target dataset's specific distribution.

To provide a more direct comparison, we visualize the "tench" class from ImageNet in Figure 11. As shown, the images generated by the General_LDM (b) using the prompt "tench" are visually inconsistent with the original data (a), the target-trained ImageNet_LDM (c), and our CoDA (d). Although they are all technically images of "tench", the General_LDM's prior generates a style that is completely misaligned with the target dataset's distribution.

We also compute the FID score using these four sets of images, with the results presented in Table 21. Although the absolute values are high due to the small sample size, the relative relationship is clear:

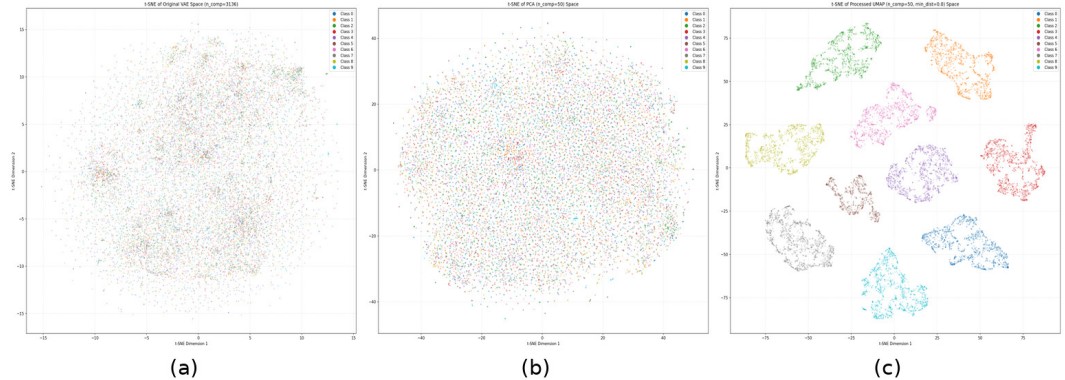

Figure 9: t-SNE visualization of the VAE feature space for ImageWoof. (a) Original VAE space (Mean $\sigma$: 10.84). (b) VAE (PCA to 50d) (Mean $\sigma$: 7.92). (c) VAE (UMAP to 50d) (Mean $\sigma$: 0.30).

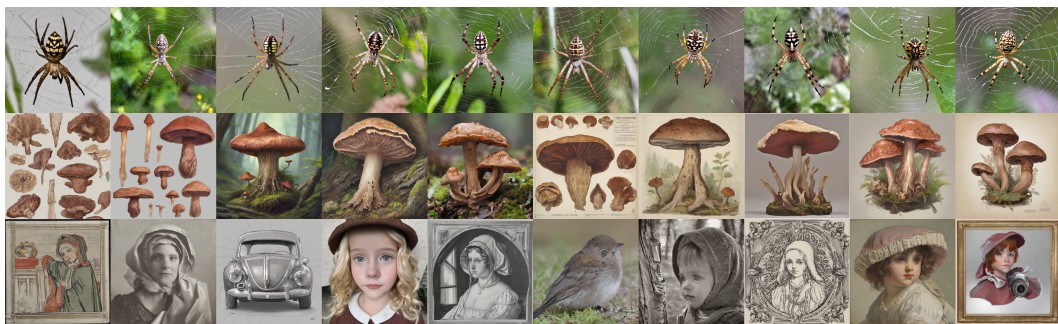

Figure 10: Qualitative comparison of SDXL's generation consistency on ImageIDC (IPC=10). The top two rows show successful generations for unambiguous classes ('garden spider' and 'gyromitra'). In contrast, the bottom row demonstrates a failure case for the polysemous class 'bonnet'. Despite the prompt intending a type of hat, the model's general prior also generates incorrect images related to other meanings of the word (a car and a bird), highlighting the distributional mismatch.

Table 21: FID-10 comparison for the "tench" class. Lower is better.

| Method | FID Score (Lower is Better) |
|---|---|
| Original Data | 81.42 |
| General_LDM | 104.14 |
| ImageNet_LDM | 71.05 |
| CoDA (Ours) | **60.09** |

- The **Original Data** (randomly sampled) serves as a baseline FID of 81.42. An effective DD method must perform better (lower FID) than this random selection.

- The **ImageNet_LDM** (71.05) achieves a better score, showing its inherent alignment to the data it was trained on, which naturally facilitates a simple DD process.

- The **General_LDM** (104.14) fails this task completely. Its severe distribution mismatch results in an FID far worse than random sampling.

- **CoDA** achieves the lowest FID (60.09) by a significant margin, demonstrating that our alignment strategy successfully bridges the gap and matches the target dataset's core distribution features.

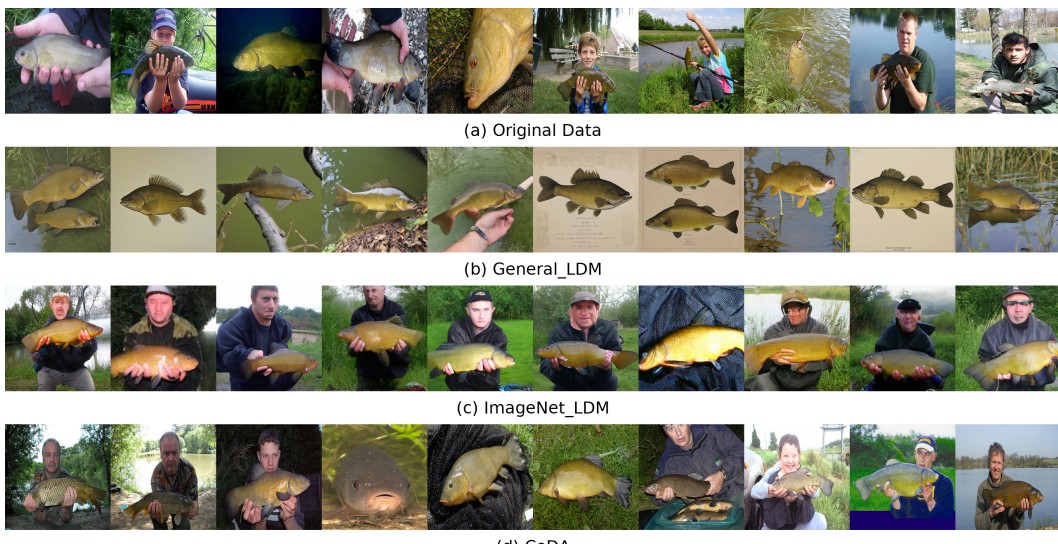

Figure 11: Qualitative comparison of the "tench" class. (a) 10 random samples from the original dataset. (b) 10 samples from General_LDM (SDXL) using the prompt "tench". (c) 10 samples from the target-trained ImageNet_LDM. (d) Our final CoDA (IPC=10) dataset. The General_LDM (b) produces a completely different style from the other three groups.

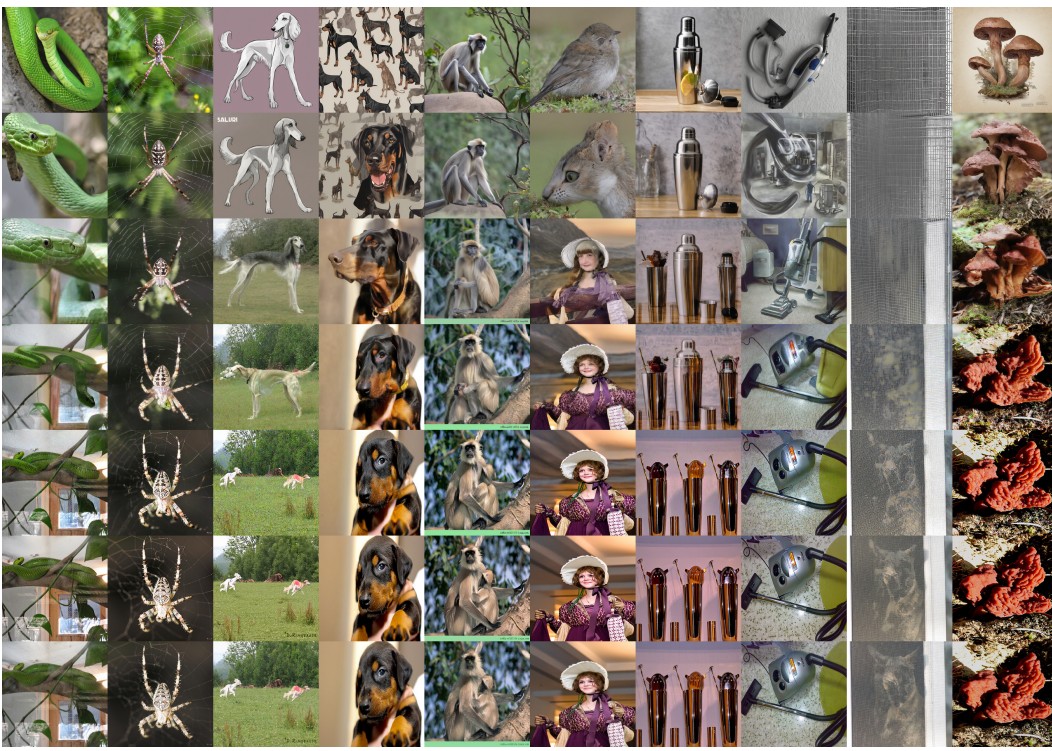

Figure 12: Visualizing the impact of guidance strength $\gamma$ on generated images from ImageIDC (IPC=10). **Rows (top to bottom):** The generated images with $\gamma = 0$ (Original SDXL), $\gamma = 0.01$, $\gamma = 0.03$, $\gamma = 0.05$, $\gamma = 0.1$, $\gamma = 0.2$ and the ground-truth real image. **Columns (left to right):** The 10 classes of the dataset: 'green mamba', 'garden spider', 'saluki', 'doberman', 'langur', 'bonnet', 'cocktail shaker', 'vacuum', 'window screen', and 'gyromitra'.

Table 22: Performance comparison of distilled datasets under different guidance strengths $\gamma$, with PIS fixed at 5.

| $\gamma$=0.00 | $\gamma$=0.01 | $\gamma$=0.03 | $\gamma$=0.05 | $\gamma$=0.10 | $\gamma$=0.20 | Real Image Set ($\mathcal{R}$) |
|---|---|---|---|---|---|---|
| $40.1 \pm 0.8$ | $45.5 \pm 1.0$ | $54.6 \pm 0.4$ | $55.8 \pm 0.2$ | $58.5 \pm 0.9$ | $56.9 \pm 1.4$ | $56.3 \pm 0.9$ |

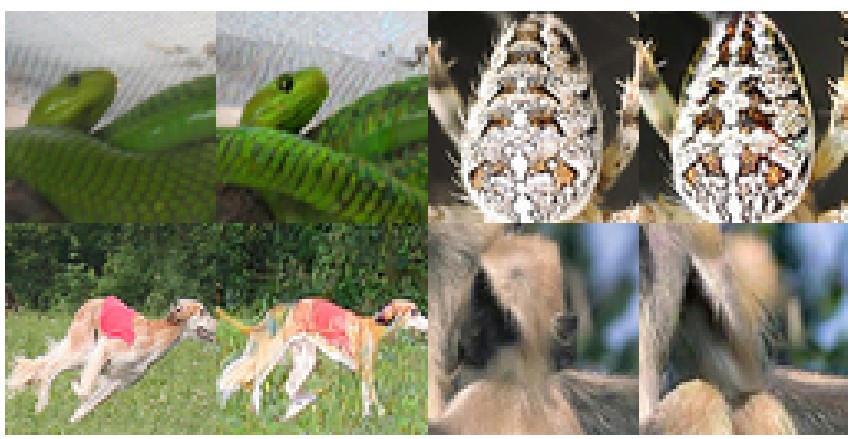

Figure 13: A detailed comparison between original real images (left in each pair) and their corresponding generated images (right in each pair) under our optimal guidance strength ($\gamma = 0.1$). The classes shown are 'green mamba' (top-left), 'garden spider' (top-right), 'saluki' (bottom-left), and 'langur' (bottom-right).

### A.13 VISUALIZING THE IMPACT OF GUIDANCE STRENGTH

This section visualizes the impact of the guidance strength $\gamma$ on the final image quality. As shown in Figure 12, we fix PIS=5 and select an image for each of the 10 classes from the ImageIDC dataset, comparing the results for $\gamma \in \{0, 0.01, 0.03, 0.05, 0.1, 0.2\}$ and the ground-truth real images. The corresponding test accuracy for each setting is presented in Table 22. As can be observed, a guidance strength of $\gamma = 0.03$ is already sufficient to substantially correct most generated images. As the strength increases, the resulting dataset's performance peaks at $\gamma = 0.1$. As the guidance is further increased to $\gamma = 0.2$, the generated images become nearly indistinguishable from the real images, and their corresponding test accuracy converges to that of the real-image set.

A more detailed qualitative comparison, presented in Figure 13, reveals the source of this performance gain. The 2.1% accuracy improvement at $\gamma = 0.1$ over the real-image set stems from the generated images' ability to enhance details while preserving the core structure of the real images. This core structure, identified by our Distribution Discovery stage, provides a solid foundation for the generation process, ensuring that the synthesized images align with the dataset's intrinsic core distribution. The richer details are injected by the model's prior, facilitated by two key mechanisms: (1) preserving the unmodified unconditional noise $\epsilon_\theta(z_t, t, \emptyset)$, and (2) allowing the model to freely refine the output during the final PIS steps.

Specifically, the generated images for 'green mamba' and 'garden spider' exhibit significantly improved brightness, contrast, and overall clarity compared to the real images. For 'saluki', the generated image avoids a potentially confusing pose in the real image—where the dog's legs are folded during a sprint—and instead depicts the limbs in a more canonical and recognizable state, which is less likely to introduce ambiguous features for a downstream classifier. For 'langur', the generation process intelligently removes a distracting element from the real image (the face of a baby monkey held in its arms) and seamlessly replaces it with the arm of the parent monkey. This refinement results in a cleaner composition where the langur's key features are more distinct and less occluded. These seemingly intelligent refinements serve as compelling evidence for the crucial role of the diffusion model's prior knowledge in creating a superior distilled dataset.

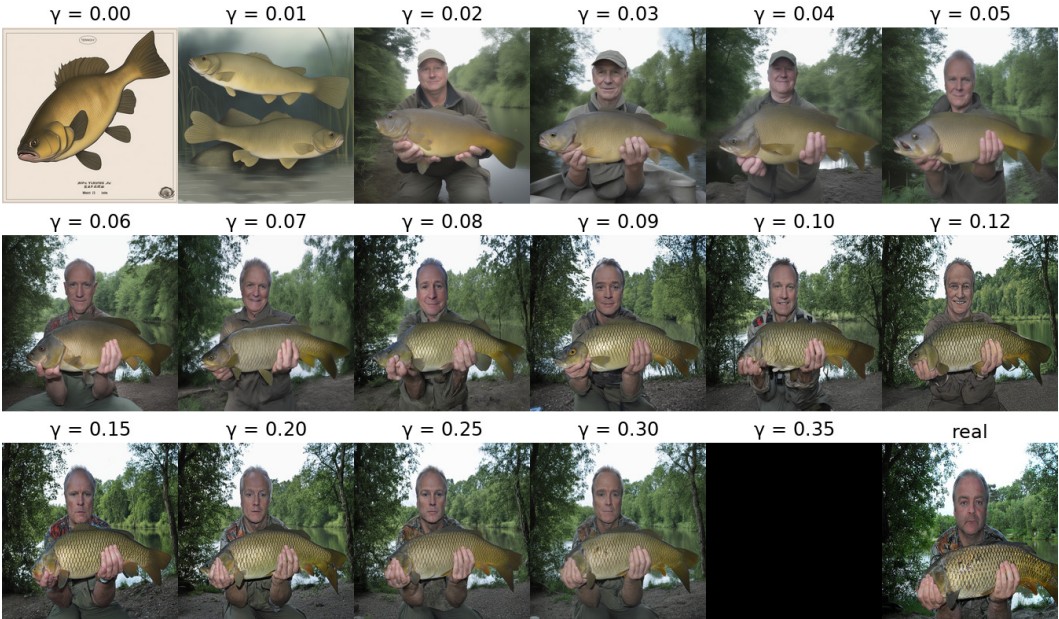

Figure 14: Visualization of the generation process collapsing as $\gamma$ increases. The guidance (PIS=5) is applied to a "tench" class sample from ImageNette. At $\gamma = 0.35$, the generation fails completely, resulting in a black image.

However, the guidance strength $\gamma$ cannot be set arbitrarily high. As visualized for a "tench" class sample in Figure 14, the generation process collapses when $\gamma$ becomes too large. Once the value reaches 0.35 or higher, the conditional guidance becomes overpowering, pulling the latent vector into a sparse region of the manifold. This causes the subsequent denoising steps to fail, resulting in a pure black image. This highlights the necessity of setting a reasonable parameter in the Distribution Alignment stage to balance fidelity and generative stability.

### A.14 Visualization for CoDA Transfer to DiT

To support our analysis in Table 11, we provide a qualitative comparison for the "doberman" class (IPC=10) in Figure 15. This visualization supports our hypothesis regarding the optimal hyperparameter settings for DiT.

As shown in (c), the optimal low-IPC configuration ($\gamma = 1.0, \text{PIS} = 15$) allows DiT more generative freedom, leveraging its target-trained prior to produce diverse images. In contrast, the optimal high-IPC configuration (b) ($\gamma = 0.8, \text{PIS} = 8$) applies stronger guidance, pulling the generated images closer in style to the real images (e). This reinforces our conclusion: DiT's inherent knowledge is beneficial but limited. At low IPCs ($\leq 20$), allowing more freedom is effective. However, when a larger, more diverse set is required (IPC=50), DiT's prior is saturated, and it requires stronger guidance from the real dataset $\mathcal{R}$ to achieve peak performance.

### A.15 Robustness to Noisy and Incomplete Labels

In this section, we address the robustness of CoDA's Distribution Discovery stage when facing noisy or incomplete labels.

**Robustness to Noisy Labels.** CoDA is inherently designed to handle noisy labels. This was a primary motivation for building our pipeline based on HDBSCAN rather than directly using K-Means. K-Means lacks intrinsic outlier detection capabilities and forces every data point into a cluster, often requiring additional, external outlier removal steps such as LOF. In contrast, HDBSCAN is fundamentally density-based, making it inherently robust to noise. For instance, a 'dog' image erroneously labeled as 'cat' will typically reside in an extremely low-density region within the 'cat'

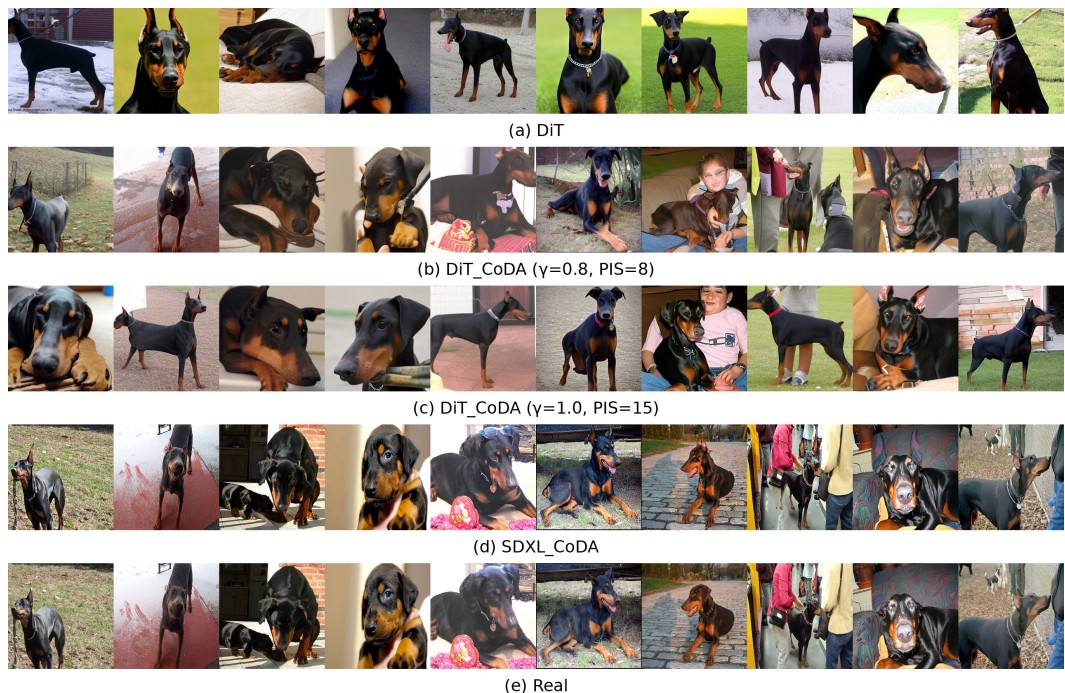

Figure 15: Qualitative comparison of CoDA transferred to DiT for the "doberman" class (IPC=10). (a) Original ImageNet-Trained DiT. (b) DiT+CoDA with high-IPC optimal parameters ($\gamma = 0.8, \mathrm{PIS} = 8$), forcing closer alignment. (c) DiT+CoDA with low-IPC optimal parameters ($\gamma = 1.0, \mathrm{PIS} = 15$), allowing more generative freedom. (d) Our SDXL+CoDA method. (e) Real images from the dataset.

VAE feature space, distinct from the dense core of true 'cat' samples. HDBSCAN automatically identifies such points as noise/outliers and excludes them from the representative sample set ($S_r$).

To validate this, we conducted a visualization experiment on the class "n02102040 English springer", where we introduced noise by mixing in 10 randomly sampled images from the class "n02123045 Tabby cat". We applied our standard pipeline: VAE encoding, followed by UMAP dimensionality reduction (with $min\_dist = 0.0$). Figure 16 visualizes the resulting manifold using t-SNE. Crucially, as observed in the figure, none of the noisy samples (Tabby cats, marked in red) are located in absolute high-density regions. Since our selection mechanism strictly targets the highest-density cores to form $S_r$, these noisy data points are naturally bypassed without requiring manual intervention.

**Robustness to Incomplete Labels (Semi-Supervised Learning).** The robustness of CoDA also enables it to handle incomplete labels efficiently, extending its applicability to semi-supervised dataset distillation. To illustrate this, we simulate a semi-supervised setting on the ImageIDC dataset and construct three configurations: (1) **Full Supervision**: The standard setting using 100% of the labeled data. (2) **Limited Labels (10%)**: We randomly select only 10% of images (130 images per class) as the labeled set, treating the remaining 90% as unseen. (3) **Semi-Supervised (10% + Pseudo)**: We utilize the 10% labeled set to guide the utilization of the 90% unlabeled data via pseudo-labeling. Specifically, we calculate the class centers of the 10% labeled data in the CLIP feature space. We then perform a global similarity search for the unlabeled samples against these centers. An unlabeled sample is assigned a pseudo-label and added to the pool if its cosine similarity to a class center exceeds a confidence threshold of 0.85.

The results are presented in Table 23. While using only 10% of the labels leads to a performance drop (70.6%), employing our pseudo-labeling strategy significantly recovers performance to 76.0%, closely approaching the fully supervised upper bound of 77.6% and significantly surpassing the previous SOTA MGD[3] (72.1%) by **3.9%**. This indicates that CoDA can effectively discover the

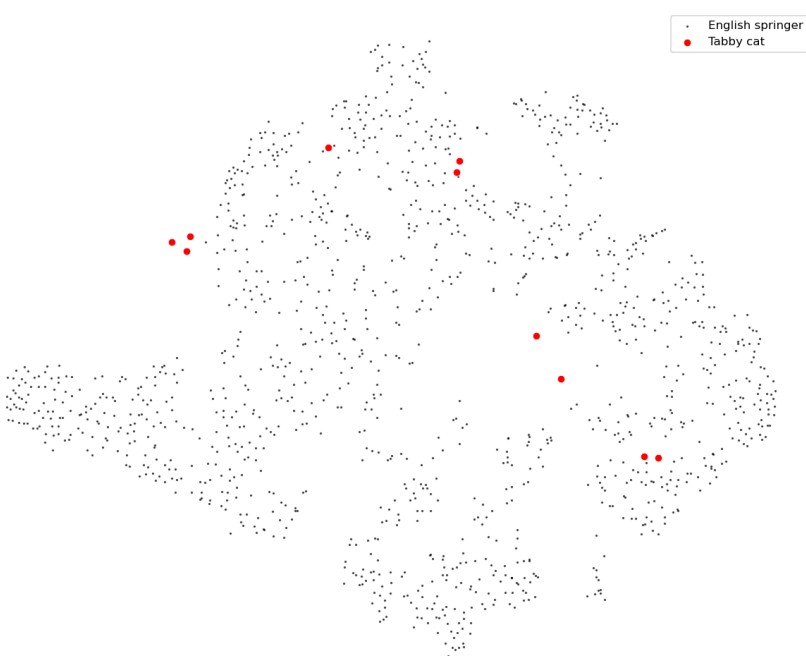

Figure 16: t-SNE visualization of the Distribution Discovery stage under noisy labels. The dataset consists of all images from the 'English springer' class (**Black dots**) mixed with 10 noisy images from the 'Tabby cat' class (**Red dots**). The noisy samples are naturally isolated in low-density regions, ensuring they are not selected by our density-based discovery algorithm.

core distribution even when the majority of the data lacks ground-truth labels, provided a small seed set of labeled data is available.

Table 23: Performance comparison on ImageIDC (IPC=50) under different label availability settings. The test model is ResNetAP-10. Results are the mean and standard deviation of three independent runs. The best results are in **bold**, and the second best are in **bold**. CoDA's discovery stage operates efficiently under both fully supervised and semi-supervised settings, consistently surpassing the SOTA.

| Method | Label Availability | Results |
|---|---|---|
| Random | | 68.1±0.7 |
| D$^4$M | Full Supervision | 63.8±0.4 |
| MGD$^3$ | | 72.1±0.8 |
| CoDA | | **77.6±0.6** |
| CoDA | Limited Labels (10%) | 70.6±0.8 |
| CoDA | Semi-Supervised (10% + Pseudo) | **76.0±0.8** |

## A.16 Transferring CoDA to Flow-Matching Models SD3

To further demonstrate the transferability of our framework, we extended the CoDA alignment stage to the Stable Diffusion 3 (SD3) architecture, which employs a Rectified Flow matching objective. The results on ImageIDC, ImageNette, and ImageWoof are presented in Table 24.

We draw several key observations from these results:

1. Similar to SDXL, SD3 is a general-purpose model and thus is unable to independently perform the DD task.

2. CoDA demonstrates remarkable transferability, seamlessly adapting to SD3's flow-matching architecture. The integration of CoDA (SD3+CoDA) leads to a substantial performance improvement over the base SD3 model.

3. Furthermore, across all configurations on ImageIDC and ImageNette, the SD3+CoDA combination matches or even surpasses the previous SOTA method (DiT+MGD[3]).

Overall, these results further validate CoDA's high adaptability and its effectiveness in bridging the core distribution gap across different generative frameworks, including those based on flow matching.

Table 24: Comparison of transferring CoDA to the SD3 architecture against other baselines. All results are evaluated using ResNetAP-10. Best results are in **bold**, and second best are in **bold**.

| Dataset | IPC | SD3 | DiT | DiT+MGD[3] | SD3+CoDA | DiT+CoDA | SDXL+CoDA |
|---|---|---|---|---|---|---|---|
| ImageIDC | 10 | 40.8±1.3 | 54.1±0.4 | 55.9±2.1 | 56.2±0.6 | **59.4±1.5** | **58.5±0.9** |
| | 20 | 42.4±0.7 | 58.9±0.2 | 61.9±0.9 | **65.4±1.5** | **65.8±1.2** | 64.6±0.5 |
| | 50 | 50.4±1.6 | 64.3±0.6 | 72.1±0.8 | **76.0±1.0** | 74.6±1.6 | **77.6±0.6** |
| ImageNette | 10 | 53.4±1.1 | 59.1±0.7 | 66.4±2.4 | 66.6±0.5 | 65.2±1.5 | **68.8±0.1** |
| | 20 | 59.6±0.9 | 64.8±1.2 | 71.2±0.5 | 72.2±0.9 | **73.0±1.2** | **72.1±0.2** |
| | 50 | 69.4±1.4 | 73.3±0.9 | 79.5±1.3 | 80.1±1.3 | **81.4±1.3** | **83.1±0.3** |
| ImageWoof | 10 | 32.2±0.8 | 34.7±0.5 | **40.4±1.9** | 38.2±0.8 | **39.2±1.3** | **39.2±0.7** |
| | 20 | 37.6±1.2 | 41.1±0.8 | **43.6±1.6** | 43.4±1.6 | **44.2±0.9** | 42.5±0.6 |
| | 50 | 41.8±1.7 | 49.3±0.2 | 56.5±1.9 | 55.4±1.1 | **59.2±0.7** | **59.4±1.0** |

## A.17 VISUALIZATION OF FINAL RESULTS

This section showcases some of the state-of-the-art (SOTA) images generated by our method.

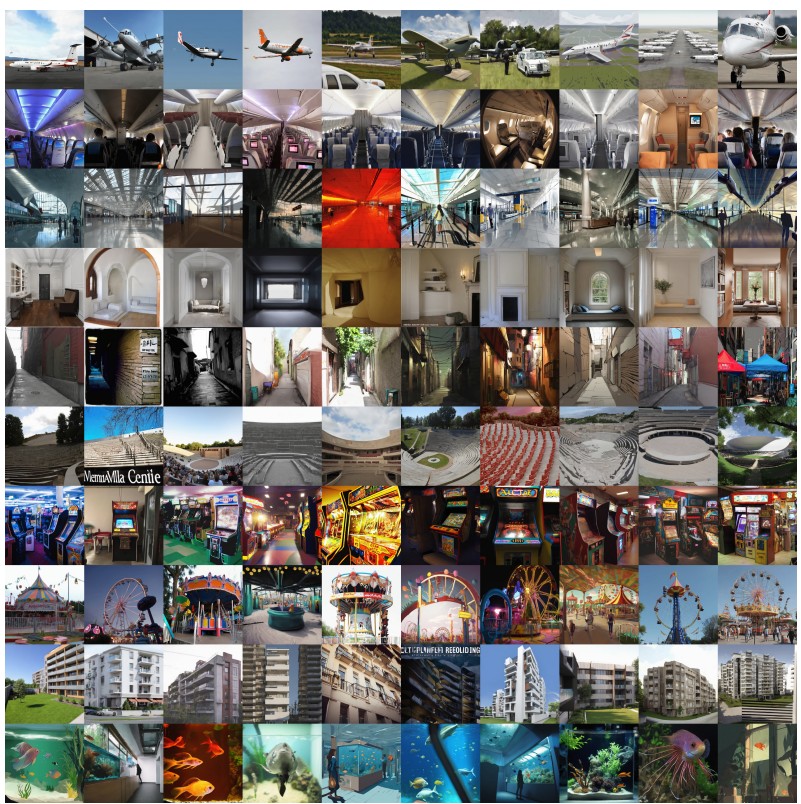

Figure 17: The complete generated dataset for the first 10 classes of Places365 at IPC=10. The rows, from top to bottom, correspond to the classes: 'airfield', 'airplane_cabin', 'airport_terminal', 'alcove', 'alley', 'amphitheater', 'amusement_arcade', 'amusement_park', 'apartment_building-outdoor', and 'aquarium'.

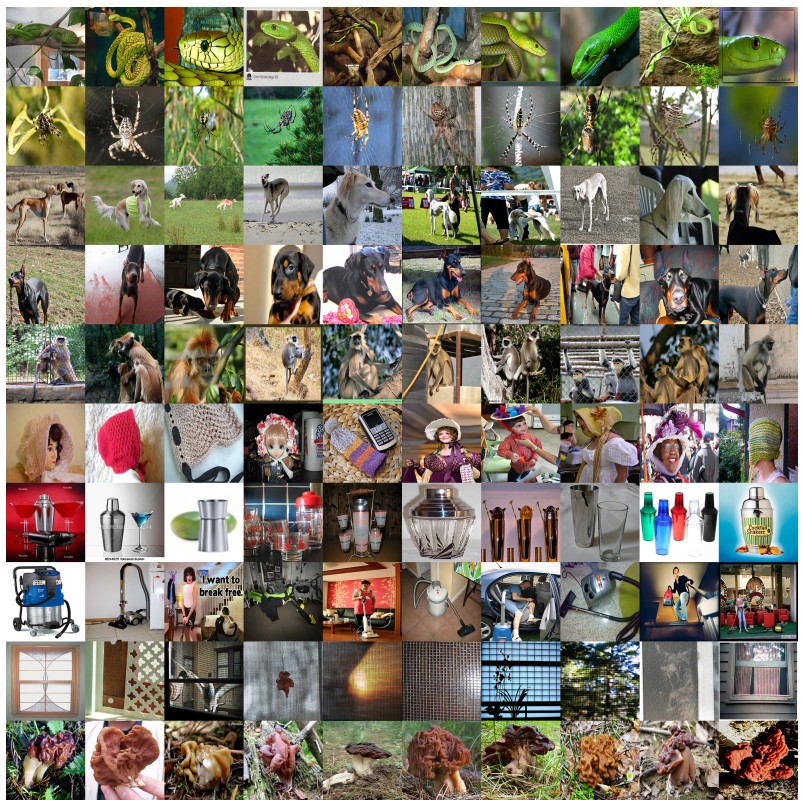

Figure 18: The complete generated dataset for ImageIDC at IPC=10.

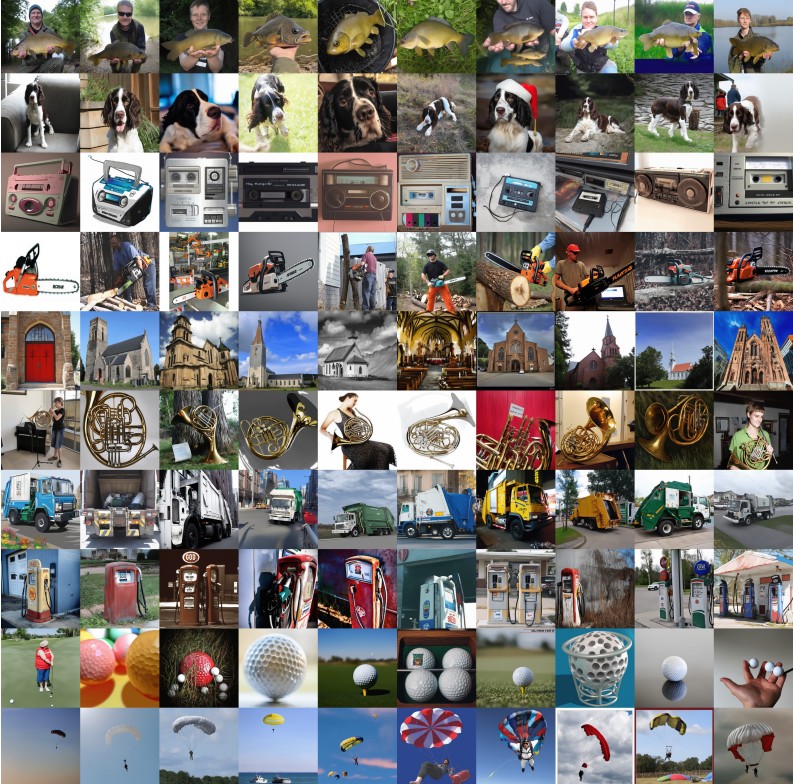

Figure 19: The complete generated dataset for ImageNette at IPC=10.

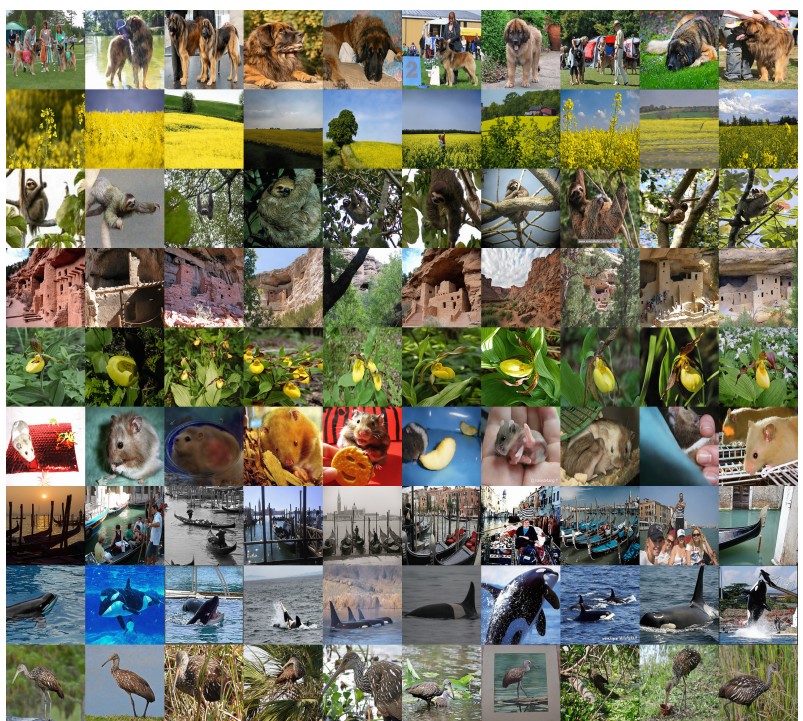

Figure 20: The complete generated dataset for ImageNet-A at IPC=10.

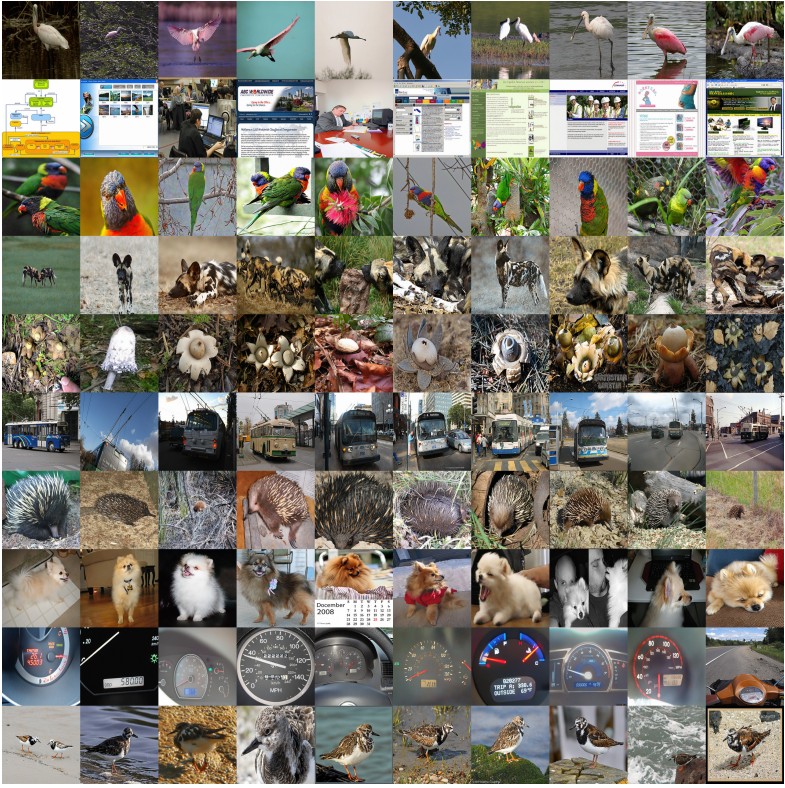

Figure 21: The complete generated dataset for ImageNet-B at IPC=10.

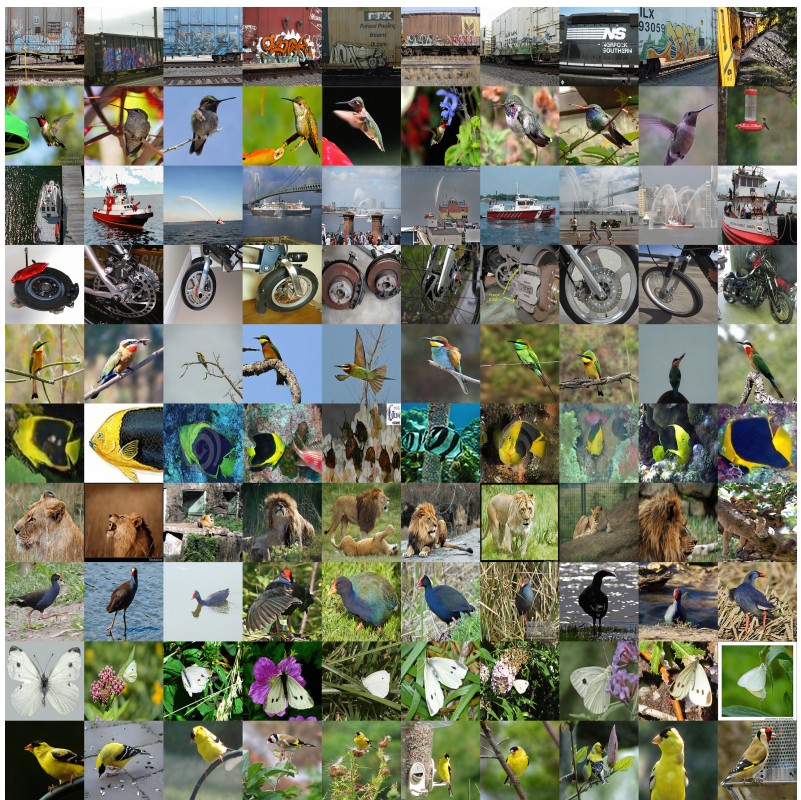

Figure 22: The complete generated dataset for ImageNet-C at IPC=10.

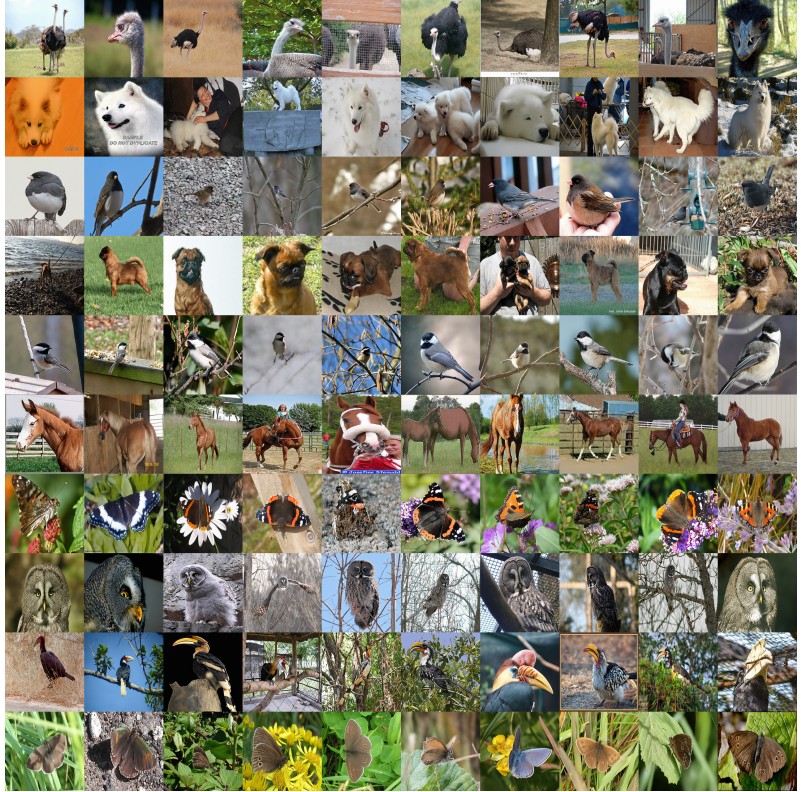

Figure 23: The complete generated dataset for ImageNet-D at IPC=10.

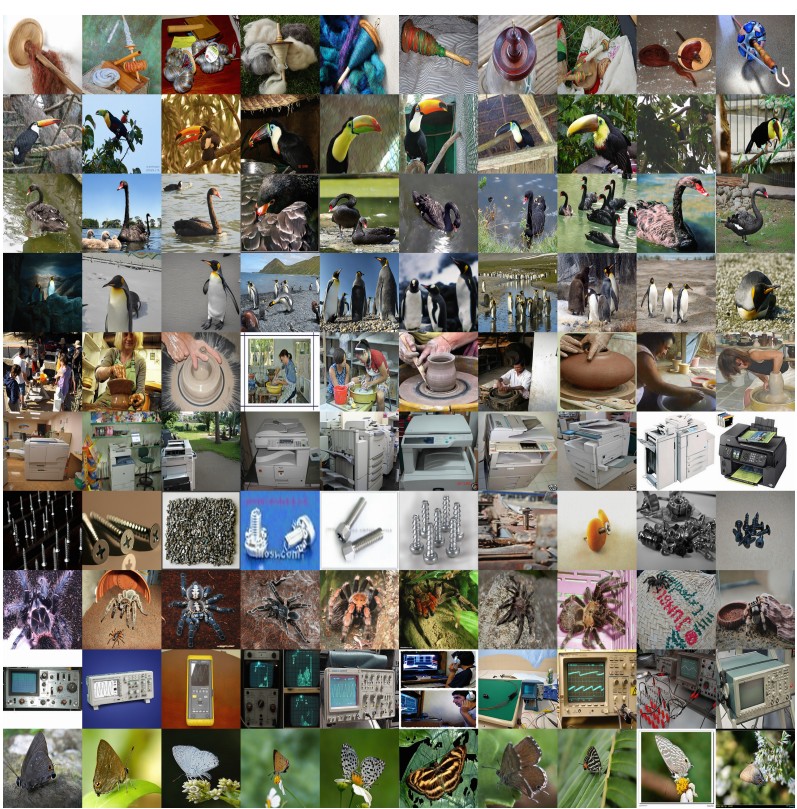

Figure 24: The complete generated dataset for ImageNet-E at IPC=10.

