# OpenReview forum: "CoDA: From Text-to-Image Diffusion Models to Training-Free Dataset Distillation"
_ICLR.cc/2026/Conference — ICLR 2026 Poster_

### Official Review · Reviewer_cWUx · 2025-10-24

**Soundness:** 2
**Presentation:** 3
**Contribution:** 3
**Rating:** 4
**Confidence:** 4

**Summary:**

This paper proposes a two-stage approach. First, the Distribution Discovery stage employs HDBSCAN (Hierarchical Density-Based Spatial Clustering of Applications with Noise) to identify all valid high-density clusters and subsequently select a set of representative samples. Second, the Distribution Alignment stage then utilizes the guidance of these representative samples to generate synthetic images, aligning the synthetic data distribution with the real data distribution.

**Strengths:**

1. The authors discover that the VAE latent space features exhibit poor separation due to the mixing of inter-class features, which severely degrades inter-class separability and intra-class compactness. To mitigate this issue, they leverage UMAP to preprocess the latent embeddings from the VAE.
2. The proposed HDBSCAN method, combined with several split strategies, successfully identifies and selects well-distributed, representative samples from the original ImageNet dataset.
3. The proposed approach demonstrates consistent performance improvements across the ImageNet-1k dataset and its 7 subsets.

**Weaknesses:**

1. Insufficient visual validation of UMAP: While the paper employs UMAP to preprocess VAE latent embeddings, the effectiveness of this dimensionality reduction for improving feature separation is not visually demonstrated.
2. Ambiguity in distribution alignment and scope violation: The Distribution Alignment stage involves a mechanism that appears to directly compensate for the difference between generated images and real samples. The strong dependence on a lambda hyperparameter suggests this step might be manually compensating for the difference, or potentially regressing to the exact real samples if lambda=1. This approach contradicts the principle of dataset distillation, which requires the synthesis of a small, artificial dataset, as its reliance on near-real samples aligns more closely with coreset selection rather than true distillation, making the paper's claimed scope questionable.
3. The claimed achievement of a new state-of-the-art accuracy of 60.4% in the 50-images-per-class (IPC) setup on ImageNet-1K in the abstract or contribution section is only a 0.1% improvement over the baseline of 60.3%. This minimal gain is highly likely to be within the margin of experimental randomness.

**Questions:**

1. In Table 5, why does the performance achieved by the "Ours (R)" configuration (utilizing real samples selected by HDBSCAN) yield lower accuracy compared to the "Ours (G)" configuration (which are synthetic images)?

---

> ### Author Response · Authors · 2025-11-24
>
> We sincerely thank Reviewer cWUx for the detailed review and valuable feedback. We greatly appreciate the reviewer's recognition of our core insight regarding the "poor separation" of VAE latent space features and the reviewer's endorsement of our solution leveraging UMAP to mitigate this issue. We are also encouraged by the reviewer's assessment that our proposed HDBSCAN-based pipeline successfully identifies well-distributed representative samples and demonstrates "consistent performance improvements" across ImageNet benchmarks. We have carefully addressed the reviewer's concerns regarding the visual validation of UMAP, the theoretical interpretation of the Distribution Alignment stage (specifically the role of $\gamma$), and the performance difference between Ours(R) and Ours(G) with new experiments and visualizations detailed below.
>
> ### **Weakness 1 (Visual validation of UMAP)**
>
> We appreciate the reviewer's suggestion. We have added Appendix A.11 to the revised paper to provide a comprehensive validation of UMAP's necessity and effectiveness from both qualitative and quantitative perspectives.
>
> **Qualitative Validation**
> Visually, as shown in Figures 8 and 9 of the revised paper, both the original VAE space and the linear PCA space fail to separate different classes, resulting in completely overlapping clusters. This confirms our critique of the raw VAE space in the paper. In contrast, our UMAP preprocessing successfully reshapes the data into clear "islands," perfectly achieving both high inter-class separation and intra-class compactness.
>
> **Quantitative Validation**
> Quantitatively, the difference is even more significant. We reported the Mean $\sigma$ (the mean of kernel bandwidths from t-SNE) as shown in the table below.
>
> **Table 1: Manifold compactness metrics for different features spaces.**
>
> | Space | Dimension | Mean $\sigma$ |
> | :---: | :---: | :---: |
> | Original VAE | 3136 | 12.57 |
> | Original CLIP | 768 | 8.06 |
> | VAE (PCA) | 768 | 8.06 |
> | CLIP (PCA) | 50 | 6.33 |
> | CLIP (UMAP, "min_dist=0.0") | 50 | **0.29** |
>
> Our UMAP preprocessing dramatically reduces the Mean $\sigma$ from 12.57 (original space) or 8.06 (PCA space) to 0.29, indicating a vastly more compact and separable manifold structure.
>
> **Performance Impact**
> This superior manifold structure directly translates into final performance. The experimental results are presented in the table below.
>
> **Table 2: Ablation study on feature spaces and clustering methods. All results are for ImageIDC (IPC=10) and are the mean of three runs on ResNetAP10.**
>
> | Dim | Pre | Space | Kmeans | CoDA |
> | :---: | :---: | :---: | :---: | :---: |
> | 3136 | None | VAE | 46.0$\pm$1.1 | 45.8$\pm$0.9 |
> | 768 | None | CLIP | 51.8$\pm$1.2 | 52.4$\pm$0.7 |
> | 768 | PCA | VAE | 46.6$\pm$0.6 | 53.0$\pm$1.0 |
> | 50 | PCA | CLIP | 49.8$\pm$0.5 | 53.6$\pm$0.9 |
> | 50 | PCA | VAE | 51.7$\pm$0.8 | 54.6$\pm$0.6 |
> | 50 | UMAP | CLIP | 54.4$\pm$0.7 | 56.0$\pm$1.2 |
> | 50 | UMAP | VAE | 54.8$\pm$0.7 | **58.5$\pm$0.9** |
>
> As shown in the table above, when executing CoDA on these different feature spaces, the UMAP-based approach achieves an optimal performance of 58.5%. This significantly outperforms the PCA-reduced VAE space (54.6%). Moreover, running CoDA on the original VAE space is nearly ineffective, yielding only 45.8%, which is similar to the poor results obtained by direct KMeans clustering (46.0%).
>
> These results collectively demonstrate the necessity and effectiveness of UMAP preprocessing.

---

> ### Author Response · Authors · 2025-11-24
>
> ### **Question 1 & Weakness 2 (Relationship between R and G, and the $\gamma$ parameter)**
>
> We thank the reviewer for this insightful comment regarding the theoretical nature of our framework and the specific role of the guidance parameter. It is undeniable that looking solely at the real dataset R obtained in the discovery stage, it can indeed be viewed as a form of coreset selection. However, CoDA includes a critical second stage: the Distribution Alignment stage. These two stages are mutually reinforcing.
>
> ### **Why Ours(G) outperforms Ours(R)**
>
> The real dataset R guides the generation to frame the macro structure of the final dataset. This ensures the generated images strictly adhere to the style of the target dataset. To further demonstrate the limitations of general generative models for DD tasks and the necessity of using the real dataset R for guidance, we have expanded Appendix A.12 (please also refer to our response to Reviewer ptsN on Question 1).
>
> By the late stages of generation, the guidance from R has already established the correct framework in the SDXL latent space. We attribute this to our strategy of first using a sufficiently large $\gamma$ and adequate guided generation steps to pull the latents deep into the core manifold of the target distribution. Subsequently, releasing the guidance during the Prior Injection Steps (PIS) allows SDXL to fully leverage its capabilities. Constrained within the target manifold, the model utilizes its priors to optimize, repair, and fine-tune the image, a process analogous to image inpainting. As shown in Figure 13 of the revised paper, this prior injection manifests as improved image contrast, amplification of representative features, and protection of core attributes, effectively performing an "intelligent refinement".
>
> As Reviewer ugRR similarly noted in CoDA's strengths, they agreed that the generative model SDXL can serve as "a powerful tool that can be harnessed to denoise and improve real world data."
>
> Therefore, the generated dataset (G) outperforms the real dataset (R) because it achieves an optimal balance between real world distribution and generative optimization.
>
> ### **Addressing the concern about $\gamma=1$ and regression to real data**
>
> In fact, setting $\lambda=1$ (denoted as $\gamma$ in our paper) would **not** cause the method to degenerate into simply reusing the real dataset.
>
> **Theoretical Reason**
>
> We speculate that the origin of this question is the definition in Section 3.2:
>
> $
> \hat{z}_{0, \mathrm{new}} = \hat{z}_0(z_t) + \Delta \hat{z}_0 = (1-\gamma)\hat{z}_0(z_t) + \gamma s_j.
> $
>
> where $\gamma = 1$ indeed implies:
>
> $\hat{z}_{0, \mathrm{new}} = s_j.$
>
> However, this is merely an intermediate calculation step used to derive the noise correction term $\Delta\epsilon_\theta$ in Equation 7.
>
> Crucially, our method is built upon the standard Classifier-Free Guidance (CFG) framework. We specifically chose to modify only the conditional predicted noise, while intentionally preserving the unconditional predicted noise unmodified to ensure generative fidelity. Therefore, even in the extreme case of $\gamma=1$, the presence of the unmodified unconditional noise (the model's generative prior) guarantees that the model invariably injects its inherent features into the generation process. Consequently, the final image $G$ remains a result guided by $s_j$, rather than an identical replica of $s_j$.
>
> **Empirical Evidence**
>
> We have added further visual analysis in Appendix A.13 of the revised paper. In our experiments, we found that once $\gamma$ reaches 0.35 or higher, the generation process typically collapses. The overpowering conditional guidance imposes an excessive modification to the predicted noise, drastically disrupting the diffusion trajectory. This pushes the latents into out-of-distribution states where the U-Net fails to predict valid noise, leading to numerical instability and resulting in pure black images.
>
> This confirms that using an extreme value like $\gamma=1$ is practically infeasible. Furthermore, as observed in Figure 14, even at a high value like $\gamma=0.3$, the generated image remains significantly distinct from the real image and never degenerates into the real dataset.

---

> ### Author Response · Authors · 2025-11-24
>
> ### **Weakness 3 (Marginal gain on ImageNet-1K)**
>
> We thank the reviewer for this careful observation regarding our performance metrics. We acknowledge that on the specific setting of ImageNet-1K at IPC 50, CoDA's improvement over the SOTA is relatively narrow. However, we believe the significance of our results becomes clear when viewed through a broader and more contextual lens.
>
> **Comprehensive SOTA Performance**
>
> While the margin on ImageNet-1K IPC 50 appears modest in isolation, CoDA establishes a significant performance lead in all experimental configurations across eight datasets, only achieving second-best results in the low-data (IPC $\le$ 20) settings on ImageWoof. Most notably, on the five ImageNet-A to ImageNet-E subsets, which represent varying levels of fine-grained difficulty, we outperformed the Target-trained SOTA by 9.6%, 8.4%, 7.5%, 4.8%, and 6.6%, respectively. This demonstrates consistent superiority across diverse data distributions.
>
> **The Core Contribution (Bridging the Gap without Training)**
>
> CoDA's core contribution lies in demonstrating that a Training-Free generative framework can rival or exceed Target-Trained methods. We contribute a complete pipeline, from the first stage of Distribution Discovery to the second stage of Distribution Alignment. Through these efforts, SDXL can be directly applied to different fine-grained tasks or even different domains without fine-tuning. As demonstrated in Table 7 and Appendix A.9 of the revised paper, CoDA seamlessly adapts to the scene dataset Places365 and the medical imaging dataset DRIVE with zero cost and no manual intervention, a capability unattainable by target-trained methods.
>
> **The Fair Comparison**
>
> Therefore, the strictly fair comparison is between CoDA and the other training-free method, $D^4M$. Taking ImageNet-1K IPC 10 as an example, as shown in Table 4 of the revised paper, CoDA (44.3%) far surpasses its direct competitor $D^4M$ (27.9%) by 16.4%. At IPC 50, CoDA (60.4%) also outperforms $D^4M$ (55.2%), achieving a 5.2% performance gain.
>
> In summary, CoDA's true achievement is successfully bridging the performance gap, reaching or even exceeding Target-trained SOTA performance without incurring the prohibitive "target pre-training" costs.

---

> ### Author Response · Authors · 2025-11-28
>
> Dear Reviewer cWUx,
>
> Thank you for the time you dedicated to reviewing our paper. Your insights concerning the visual verification of UMAP and the interpretation of the guidance parameter were instrumental in refining our manuscript.
>
> We hope that the newly added visualizations and the clarification on the parameter mechanism have satisfactorily resolved your doubts. Please do not hesitate to let us know if there are any other aspects we can elaborate on.
>
> Best Regards,
>
> Authors of Submission 8388

---

### Official Review · Reviewer_ptsN · 2025-10-25

**Soundness:** 3
**Presentation:** 3
**Contribution:** 2
**Rating:** 4
**Confidence:** 5

**Summary:**

The paper proposes Core Distribution Alignment (CoDA) — a dataset distillation framework that enables large-scale dataset distillation, relying solely on an off-the-shelf text-to-image diffusion model (SDXL).

CoDA introduces two phases:
- Distribution Discovery: Identifies the core distribution via UMAP + HDBSCAN clustering in the VAE latent space, selecting representative samples.
- Distribution Alignment: Guides diffusion sampling using a core-distribution-based energy term, aligning generated images with selected samples.

Experiments on ImageNet-1K and its subsets show CoDA achieves SOTA performance. The method is also computationally efficient and generalizes to new domains like Places365.

**Strengths:**

- **Clear formulation**: the paper presents a clean DD framework that combines the clustering-based representative discovery (such as D$^4$M and MGD$^3$) and energy-based diffusion guidance (such as MGD$^3$ and IGD). Even though both components exist in prior works, CoDA articulates them under a consistent probabilistic formulation (Eq. 4–7), which makes it easy to follow and reproduce.

- **Strong empirical performance and generalization**: CoDA achieves SOTA accuracy on multiple benchmarks, and the results generalize across both architectures and datasets. The experiments are extensive and well-controlled.

- **Efficient implementation**: CoDA’s distribution discovery phase runs entirely on CPU using VAE features, and the guidance process uses a well pre-trained T2I SDXL with no further training or SFT.

**Weaknesses:**

- **Overstated “truely training-free DD” claim and misleading title**: The title “From Text-to-Image Diffusion Models to Truly Training-Free Dataset Distillation” is somewhat exaggerated and conceptually inconsistent. The authors argue that prior generative DD methods are paradoxical because they depend on diffusion models pretrained on the target dataset like imagenet-1k, whereas CoDA avoids this by using a general text-to-image diffusion model (SDXL). However, this reasoning is misleading: using a pretrained generative model regardless of its training data is a standard and acceptable practice in DD research, not a paradox. In fact, CoDA still relies on a large diffusion model that was pretrained on an even larger and more expensive dataset (LAION) than ImageNet. Thus, the method is not “training-free” in any absolute sense. It merely shifts reliance from a domain-specific model to a more general pretrained one. The paper’s framing unfairly overemphasizes this distinction with a whole para and risks confusing readers about what “training-free” actually means in the context of DD. I do agree with your distributional mismatch claim, but not the paradoxical pretrained model claim that you described in both abstract and introduction.

- **Incremental method**: The authors a) replace the K-means algorithm with HDBSCAN and design a IPC Matching algorithm to obtain sufficient samples (similar to the prototype conception in D4M). Then, they b) use these samples to guide the denosing process (similar to the guidance mechanism in MGD3). Although the authors claim that the proposed method is distinct from D4M and MGD3,  it is still somewhat incremental based on the prior works.

**Questions:**

- **distribution mismatch**: Figure 1a is confusing. The authors claim that the general T2I LDM “generates samples confined to a limited region of the data distribution,” implying an intrinsic coverage limitation. Which algorithm do you use to generate the visualization? T-SNE or UMAP? Do you have quantitative results to support your observation? Since the pre-trained DMs have better IS-FID score among generative models, they generate high fidelity and diversity samples. I know the dataset domain is different, but another reason for raising such distribuition mismatch may be the hyper parameters you used in your visualization algorithm. Are the results come from the same settings?
- **Cross-architecture results**: What's the settings of table4? The results are come from a single eval network or the avg results of several eval networks? Are the eval networks be the same?
- **Lack of experiments**: How about apply coda to the DiT or other DMs pre-trained on the target (such as Imagenet) dataset? Will it augment the SOTA results? It is also important to explore the ultimate capacity of coda.
- **Lack of datasets**: Since the authors claim that the intrinsic core distribution is important in DD, I think the authors should provide the results on ImageWoof, which is a common dataset used in DD with more inter-class similarity.

---

> ### Author Response · Authors · 2025-11-24
>
> We sincerely thank Reviewer ptsN for the thorough evaluation and constructive criticism. We appreciate the reviewer's recognition of CoDA's "clear formulation," "strong empirical performance," and "efficient implementation." We are particularly grateful for the reviewer's incisive analysis regarding the definition of "training-free." The reviewer's observation that our method "shifts reliance from a domain-specific model to a more general pretrained one" is astute and perfectly captures the essence of our approach. This insight helped us clarify our core contribution: CoDA eliminates the prohibitive marginal cost of re-training for each new target domain. We have also carefully addressed the reviewer's questions regarding experimental settings, model transferability (to DiT), and dataset diversity (ImageWoof) by conducting comprehensive new experiments, as detailed below.
>
> ### **Question 1 (Clarification on Figure 1a and the limitations of General LDMs)**
>
> We appreciate the reviewer's scrutiny regarding the visualization and agree that validating the distribution mismatch is crucial. We provide the detailed implementation steps and further evidence below.
>
> **Visualization Implementation Details (Figure 1a)**
>
> To ensure a fair comparison, we strictly controlled the experimental setup:
>
> 1.  **Data Collection**: We constructed a combined dataset of 1,330 images comprising four distinct groups:
>
>     (a) Original Data: 1,300 real images from the ImageNette "tench" class.
>
>     (b) General LDM: 10 images generated by SDXL using only the text prompt "tench".
>
>     (c) ImageNet LDM: 10 images generated by the ImageNet-Trained LDM (using the official CompVis/latent-diffusion repository: https://colab.research.google.com/github/CompVis/latent-diffusion/blob/main/scripts/latent_imagenet_diffusion.ipynb).
>
>     (d) Ours General LDM: 10 images generated by our CoDA method.
>
> 2.  **Preprocessing & Encoding**: All 1,330 images underwent identical preprocessing (Resize to $256 \times 256$, ToTensor, Normalize) and were encoded into the latent space using the same SDXL VAE to ensure a unified feature space.
>
> 3.  **Dimensionality Reduction**: We flattened the latent vectors and performed t-SNE on the combined dataset with fixed hyperparameters (n_components=2, perplexity=30, random_state=42) to ensure reproducibility. The resulting projections were then separated by group for visualization.
>
> **Clarification on "Distribution Mismatch" vs. Image Quality**
>
> We concur that pre-trained General LDMs (like SDXL) demonstrate excellent perceptual quality (high IS/FID) in a general sense. However, the objective of Dataset Distillation is not merely to generate realistic images, but to accurately reconstruct the specific manifold of the target dataset.
>
> The "mismatch" we refer to is the discrepancy between the model's web-scale prior and the specific target domain. While general T2I models effectively grasp the semantic meaning of a category, their visual representation often diverges significantly from the specific form of expression characteristic of the target dataset. To further illustrate this issue, we have added Appendix A.12 (Limitations of General-Purpose Models in DD Tasks) to the revised paper:
>
> a. **Qualitatively**, as shown in the new Figure 11, images generated by the General LDM (b) exhibit significant stylistic divergence from the Original Data (a) and the Target-Trained LDM (c), whereas CoDA (d) successfully bridges this gap.
>
> b. **Quantitatively**, the FID scores confirm that a General LDM alone is incapable of effectively performing the DD task. Due to the severe distribution bias between its prior-generated dataset and the target dataset, its performance is significantly inferior to simple Random Selection of real data.
>
> ### **Question 2 (Experimental setup)**
>
> We thank the reviewer for pointing out this ambiguity. We have updated the caption of the corresponding table (Table 5 in the revised paper) to explicitly state the experimental conditions and reporting standards.
>
> To clarify the specific settings:
> a. **Protocol:** We utilized hard labels and strictly adhered to the training hyperparameters and evaluation protocol established in Minimax (Gu et al., 2024).
> b. **Evaluation:** We evaluated 5 distinct downstream architectures (ResNet10-AP, ResNet18, ResNet50, ResNet101, and ViT) across 7 ImageNet subsets.
>
> Crucially, the results presented in this table do not represent an average performance across multiple frameworks. Instead, to ensure transparency and rigorous baselining, we report the performance of our distilled data evaluated specifically on the distinct network architecture listed. Each entry represents the mean and standard deviation derived from three independent runs of the specific architecture.

---

> ### Author Response · Authors · 2025-11-24
>
> ### **Question 3 (Transferring CoDA to ImageNet-Trained DiT)**
>
> We thank the reviewer for this valuable suggestion. To explore the ultimate capacity of CoDA, we conducted extensive experiments transferring our framework to both (1) the target-trained DiT and (2) the Stable Diffusion 3 (SD3) architecture to test broader adaptability.
>
> Specifically, we encoded the representative samples $\mathcal{R}$ discovered by our SDXL-based pipeline using the ImageNet-Trained DiT's VAE and SD3's VAE, and used them to guide the generation process. The experimental results are presented in the table below.
>
> **Table 1: Adaptability of CoDA across different generator architectures (DiT & SD3). All results are the mean of three runs on ResNetAP10.**
>
> | Dataset | IPC | SD3 | DiT | DiT+MGD3 | SD3+CoDA | DiT+CoDA | SDXL+CoDA |
> | :---: | :---: | :---: | :---: | :---: | :---: | :---: | :---: |
> | ImageIDC | 10 | 40.8$\pm$1.3 | 54.1$\pm$0.4 | 55.9$\pm$2.1 | 56.2$\pm$0.6 | **59.4$\pm$1.5** | 58.5$\pm$0.9 |
> | ImageIDC | 20 | 42.4$\pm$0.7 | 58.9$\pm$0.2 | 61.9$\pm$0.9 | 65.4$\pm$1.5 | **65.8$\pm$1.2** | 64.6$\pm$0.5 |
> | ImageIDC | 50 | 50.4$\pm$1.6 | 64.3$\pm$0.6 | 72.1$\pm$0.8 | 76.0$\pm$1.0 | 74.6$\pm$1.6 | **77.6$\pm$0.6** |
> | ImageNette | 10 | 53.4$\pm$1.1 | 59.1$\pm$0.7 | 66.4$\pm$2.4 | 66.6$\pm$0.5 | 65.2$\pm$1.5 | **68.8$\pm$0.1** |
> | ImageNette | 20 | 59.6$\pm$0.9 | 64.8$\pm$1.2 | 71.2$\pm$0.5 | 72.2$\pm$0.9 | **73.0$\pm$1.2** | 72.1$\pm$0.2 |
> | ImageNette | 50 | 69.4$\pm$1.4 | 73.3$\pm$0.9 | 79.5$\pm$1.3 | 80.1$\pm$1.3 | 81.4$\pm$1.3 | **83.1$\pm$0.3** |
> | ImageWoof | 10 | 32.2$\pm$0.8 | 34.7$\pm$0.5 | **40.4$\pm$1.9** | 38.2$\pm$0.8 | 39.2$\pm$1.3 | 39.2$\pm$0.7 |
> | ImageWoof | 20 | 37.6$\pm$1.2 | 41.1$\pm$0.8 | 43.6$\pm$1.6 | 43.4$\pm$1.6 | **44.2$\pm$0.9** | 42.5$\pm$0.6 |
> | ImageWoof | 50 | 41.8$\pm$1.7 | 49.3$\pm$0.2 | 56.5$\pm$1.9 | 55.4$\pm$1.1 | 59.2$\pm$0.7 | **59.4$\pm$1.0** |
>
> **Application to Target-Trained DiT:**
>
> a. **Significant Enhancement**: Transferring CoDA to DiT significantly enhances DiT's baseline capability. At the IPC=50 setting, DiT+CoDA surpasses the original DiT by 10.3% on ImageIDC, 8.1% on ImageNette, and 9.9% on ImageWoof.
>
> b. **SOTA Comparison**: The DiT+CoDA combination outperforms the previous SOTA (DiT+MGD3) in 7 out of 9 configurations. It also surpasses our own "SDXL+CoDA" in specific low-data settings, proving that CoDA can seamlessly adapt to and boost various generative architectures.
>
> c. **Hyperparameter Insight (Knowledge Saturation)**:
> We observed a distinct shift in optimal hyperparameters depending on the IPC setting:
>
> i. Low IPC ($\le$ 20): The optimal configuration is $\gamma=1.0, \text{PIS}=15$. This setting allows more "generative freedom," suggesting that for small datasets, DiT's inherent ImageNet prior is sufficient and beneficial.
>
> ii. High IPC (=50): Using the low-IPC settings resulted in suboptimal performance (e.g., 70.4% on ImageIDC). However, tightening the guidance to $\gamma=0.8, \text{PIS}=8$ significantly boosted performance (e.g., to 74.6%).
> This suggests that while DiT's internal knowledge is high-quality, it is limited in diversity. When required to produce a larger, more diverse dataset (IPC=50), its internal prior "saturates." In this regime, it requires stronger guidance from the real dataset ($S_r$) (i.e., stronger $\gamma$, lower PIS) to achieve peak performance.
>
> **Application to Flow-Matching SD3:**
>
> a. **Limitations of General Model**: Similar to SDXL, SD3 is a general-purpose model and thus is unable to independently perform the DD task efficiently. This further corroborates our response to **Question 1** regarding the limitations of general-purpose models.
>
> b. **Significant Enhancement**: Integrating CoDA leads to a substantial performance improvement over the base SD3 model across all tested configurations.
>
> c. **SOTA Comparison**: On ImageIDC and ImageNette, the SD3+CoDA combination matches or even surpasses the previous SOTA method (DiT+MGD3), validating the framework's effectiveness on Flow Matching architectures.
>
> These experiments collectively demonstrate the portability and efficient adaptability of the CoDA framework. While both SD3+CoDA and DiT+CoDA prove to be powerful, we maintain our preference for SDXL+CoDA as our primary method. SDXL+CoDA offers comparable (and often superior) performance while retaining the crucial advantage of zero-shot generalization to completely new domains.
>
> We have added a new section, "Transferring CoDA to Target-Trained Models," in the ablation study and Appendix A.16 to elaborate on these findings. Additionally, we have provided visual analysis of this experiment in Appendix A.14.

---

> ### Author Response · Authors · 2025-11-24
>
> ### **Question 4 (ImageWoof Dataset)**
>
> We thank the reviewer for this valuable suggestion. We conduct a comprehensive evaluation on the ImageWoof subset shown below.
>
> **Table 2: Performance comparison with pre-trained diffusion models and other SOTA methods on ImageWoof.** All the results are reproduced by us for the 256x256 resolution. The missing results are due to out-of-memory. Results shown for the previous works are from MGD3. The arrow annotations indicate the performance difference between CoDA and the previous SOTA.
>
> | IPC | Test Model | Random | Herding | DiT | DM | IDC-1 | GLaD | Minimax | MGD3 | CoDA(Ours) | Full |
> | :---: | :---: | :---: | :---: | :---: | :---: | :---: | :---: | :---: | :---: | :---: | :---: |
> | 10 (0.8%) | ConvNet-6 | 24.3$\pm$1.1 | 26.7$\pm$0.5 | 34.2$\pm$1.1 | 26.9$\pm$1.2 | 33.3$\pm$1.1 | 33.8$\pm$0.9 | **37.0$\pm$1.0** | 34.7$\pm$1.1 | 35.7$\pm$1.4 ($\downarrow$1.3) | 86.4$\pm$0.2 |
> | 10 (0.8%) | ResNetAP-10 | 29.4$\pm$0.8 | 32.0$\pm$0.3 | 34.7$\pm$0.5 | 30.3$\pm$1.2 | 39.1$\pm$0.5 | 32.9$\pm$0.9 | 39.2$\pm$1.3 | **40.4$\pm$1.9** | 39.2$\pm$0.7 ($\downarrow$1.2) | 87.5$\pm$0.5 |
> | 10 (0.8%) | ResNet-18 | 27.7$\pm$0.9 | 30.2$\pm$1.2 | 34.7$\pm$0.4 | 33.4$\pm$0.7 | 37.3$\pm$0.2 | 31.7$\pm$0.8 | 37.6$\pm$0.9 | 38.5$\pm$2.5 | **38.8$\pm$1.3** ($\uparrow$0.3) | 89.3$\pm$1.2 |
> | 20 (1.6%) | ConvNet-6 | 29.1$\pm$0.7 | 29.5$\pm$0.3 | 36.1$\pm$0.8 | 29.9$\pm$1.0 | 35.5$\pm$0.8 | - | 37.6$\pm$0.9 | **39.0$\pm$3.5** | 38.1$\pm$1.3 ($\downarrow$0.9) | 86.4$\pm$0.2 |
> | 20 (1.6%) | ResNetAP-10 | 32.7$\pm$0.4 | 34.9$\pm$0.1 | 41.1$\pm$0.8 | 35.2$\pm$0.6 | 43.4$\pm$0.3 | - | **45.8$\pm$0.5** | 43.0$\pm$1.6 | 42.5$\pm$0.6 ($\downarrow$3.3) | 87.5$\pm$0.5 |
> | 20 (1.6%) | ResNet-18 | 29.7$\pm$0.5 | 32.2$\pm$0.6 | 40.5$\pm$0.5 | 29.8$\pm$1.7 | 38.6$\pm$0.2 | - | **42.5$\pm$0.6** | 41.9$\pm$2.1 | 42.0$\pm$0.8 ($\downarrow$0.5) | 89.3$\pm$1.2 |
> | 50 (3.8%) | ConvNet-6 | 41.3$\pm$0.6 | 40.3$\pm$0.7 | 46.5$\pm$0.8 | 44.4$\pm$1.0 | 43.9$\pm$1.2 | - | 53.9$\pm$0.6 | 54.5$\pm$1.6 | **56.8$\pm$0.9** ($\uparrow$2.3) | 86.4$\pm$0.2 |
> | 50 (3.8%) | ResNetAP-10 | 47.2$\pm$1.3 | 49.3$\pm$0.7 | 49.3$\pm$0.2 | 47.1$\pm$1.1 | 48.3$\pm$1.0 | - | 56.3$\pm$1.0 | 56.5$\pm$1.9 | **59.4$\pm$1.0** ($\uparrow$2.9) | 87.5$\pm$0.5 |
> | 50 (3.8%) | ResNet-18 | 47.9$\pm$1.8 | 48.3$\pm$1.2 | 50.1$\pm$0.5 | 46.2$\pm$0.6 | 48.3$\pm$0.8 | - | 57.1$\pm$0.6 | 58.3$\pm$1.4 | **61.2$\pm$0.9** ($\uparrow$2.9) | 89.3$\pm$1.2 |
> | 70 (5.4%) | ConvNet-6 | 46.3$\pm$0.6 | 46.2$\pm$0.6 | 50.1$\pm$1.2 | 47.5$\pm$0.8 | 48.9$\pm$0.7 | - | 55.7$\pm$0.9 | 55.1$\pm$2.5 | **56.4$\pm$1.2** ($\uparrow$1.3) | 86.4$\pm$0.2 |
> | 70 (5.4%) | ResNetAP-10 | 50.8$\pm$0.6 | 53.4$\pm$1.4 | 54.3$\pm$0.9 | 51.7$\pm$0.8 | 52.8$\pm$1.8 | - | 58.3$\pm$0.2 | 60.2$\pm$2.4 | **61.2$\pm$0.7** ($\uparrow$1.0) | 87.5$\pm$0.5 |
> | 70 (5.4%) | ResNet-18 | 52.1$\pm$1.0 | 49.7$\pm$0.8 | 51.5$\pm$1.0 | 51.9$\pm$0.8 | 51.1$\pm$1.7 | - | 58.8$\pm$0.7 | 59.7$\pm$2.7 | **62.4$\pm$1.4** ($\uparrow$2.7) | 89.3$\pm$1.2 |
> | 100 (7.7%) | ConvNet-6 | 52.2$\pm$0.4 | 54.4$\pm$1.1 | 53.4$\pm$0.3 | 55.0$\pm$1.3 | 53.2$\pm$0.9 | - | 61.1$\pm$0.7 | 60.1$\pm$1.2 | **62.4$\pm$0.7** ($\uparrow$1.3) | 86.4$\pm$0.2 |
> | 100 (7.7%) | ResNetAP-10 | 59.4$\pm$1.0 | 61.7$\pm$0.9 | 58.3$\pm$0.8 | 56.4$\pm$0.8 | 56.1$\pm$0.9 | - | 64.5$\pm$0.2 | 66.5$\pm$1.0 | **67.6$\pm$0.8** ($\uparrow$1.1) | 87.5$\pm$0.5 |
> | 100 (7.7%) | ResNet-18 | 61.5$\pm$1.3 | 59.3$\pm$0.7 | 58.9$\pm$1.3 | 60.2$\pm$1.0 | 58.3$\pm$1.2 | - | 65.7$\pm$0.4 | 68.8$\pm$0.7 | **71.4$\pm$1.4** ($\uparrow$2.6) | 89.3$\pm$1.2 |
>
> At high IPCs (50, 70, 100), CoDA significantly outperforms the SOTA across all three downstream architectures. At low IPCs (10, 20), while the performance on ConvNet-6 and ResNetAP-10 is marginally below the top SOTA, CoDA consistently achieves a strong second-best result.
>
> We further investigate the reason for this success in Appendix A.11 (Visualization and Analysis of Feature Spaces and the Role of UMAP). Specifically, Figure 9 in the revised paper visually demonstrates that our UMAP preprocessing can automatically adapt to tasks of varying granularity. Even for the fine-grained ImageWoof dataset, it successfully transforms the latent space into a structure characterized by high inter-class separation and intra-class compactness. This lays a critical foundation for CoDA's subsequent clustering and the second stage of generation.
>
> We have added the ImageWoof experiments to Section 4.2 (Comparison With State-of-the-Art Methods) of the revised paper.

---

> ### Author Response · Authors · 2025-11-24
>
> ### **Weakness 1 (The Practical Meaning of "Training-Free")**
>
> We thank the reviewer for the insightful summary. We fully agree with the characterization that CoDA "merely shifts reliance from a domain-specific model to a more general pretrained one." In fact, we argue that this shift is precisely the core advantage of our approach. In the context of CoDA, the term "training-free" strictly refers to the elimination of marginal training costs when adapting to a new target domain. In addition, we removed the word "Truly" from the title in our revision to avoid ambiguity.
>
> As the reviewer noted, using target-trained generative models for DD is a common practice. However, using a model like DiT trained on ImageNet itself indeed confines the scope of the method to the specific domain. To switch to a different domain like Places365, one must either find a new large-scale generative model pre-trained specifically on Places365, which involves resolving framework migration and parameter tuning issues, or pay the prohibitive cost of thousands of TPU-days to retrain a generative model on the new dataset before starting the DD task.
>
> In contrast, CoDA decouples the distillation process from domain-specific pre-training. It enables arbitrary and zero-cost switching of the working domain without any fine-tuning or manual intervention. We empirically validate this capability in the revised paper by successfully adapting to the scene dataset Places365 (Table 7 of the revised paper) and the medical imaging dataset DRIVE (Appendix A.9 of the revised paper).
>
> CoDA provides a framework that minimizes dependency on target-specific data. By successfully resolving the associated distribution mismatch, we enable general-purpose models to achieve or even surpass the performance of target-trained methods, thereby significantly lowering the barrier to entry for high-performance Dataset Distillation.
>
> ### **Weakness 2 (Incremental Method Concern)**
>
> We thank the reviewer for their careful reading. While CoDA shares the high-level "Discovery + Alignment" structure with prior works like D4M and MGD3, both stages of our method diverge fundamentally in their conceptual approach.
>
> ### **Discovery Stage**
>
> Conceptually, previous methods rely on global statistical clustering based on an "Idealized Space Assumption," whereas CoDA introduces a "Manifold-First Paradigm" to resolve intrinsic topological complexities.
>
> Specifically, these approaches underestimated the intrinsic complexity of the VAE space, assuming it is naturally convex, isotropic, and ready for direct Euclidean clustering. Consequently, their innovation is limited to swapping clustering algorithms. For example, MGD3 simply substituted D4M's mini-batch K-Means with standard K-Means and experimented with alternatives like Spectral Clustering or GMM.
>
> We identified that such "simple replacement" strategies are insufficient. Whether in the raw VAE space or a PCA-reduced space, direct clustering remains ineffective due to the diffuse and intermixed nature of the data distributions.
>
> To address this, we utilize UMAP preprocessing to fundamentally optimize the latent layout, transforming the feature space into a structure characterized by high inter-class separation and intra-class compactness. This transformation lays a solid foundation for the subsequent density-based discovery pipeline. As demonstrated in Appendix A.10, this pipeline achieves efficiency comparable to single-step K-Means while delivering significantly superior performance.
>
> We have added Appendix A.11 (Visualization and Analysis of Feature Spaces and the Role of UMAP) to the revised paper. This section provides further evidence that native VAE or other feature spaces like CLIP are unsuitable for direct clustering. It also demonstrates that UMAP preprocessing is superior to simple PCA, that standard K-Means is ineffective in this context, and that CoDA's discovery pipeline significantly outperforms simple one-step clustering algorithms.

---

> ### Author Response · Authors · 2025-11-24
>
> ### **Alignment Stage**
>
> While CoDA and MGD3 both modify the noise prediction, they represent fundamentally different derivation logics. MGD3 employs an empirical scaling factor primarily to adapt to noise intensity, whereas CoDA derives an exact mathematical inversion to rigorously bridge the data and noise manifolds.
>
> **The Empirical Heuristic in MGD3:**
> MGD3 calculates the guidance signal in Data Space, measuring the difference between the target mode $m_i$ and the predicted clean latent $\hat{x}_0^t$ (as shown in **their** Eq. 5):
>
> $$
> \mathbf{g}_t = (m_i - \hat{x}_0^t),
> $$
>
> However, MGD3 applies this Data Space signal directly to the Noise Space prediction ($\epsilon_\theta$), overlooking the fundamental spatial mismatch. Without deriving a mathematical bridge, they simply introduce $\sigma_t$ as a weighting factor (as shown in **their** Eq. 6):
>
> $$
> \epsilon_\theta(x_t, t, c) = \epsilon_\theta(x_t, t, c) + \lambda \cdot \mathbf{g}_t \cdot \sigma_t,
> $$
>
> Crucially, this use of $\sigma_t$ is merely an empirical choice intended to adaptively scale the guidance with noise intensity, rather than a theoretical solution to the spatial inconsistency.
>
> **CoDA's Rigorous Derivation:**
> In contrast, CoDA starts with the explicit objective of mapping the spatial correction into the noise space. Through Equations 4–7, we mathematically solve the inverse problem: how to translate a linear interpolation in data space into an equivalent correction in noise space. This derivation yields the exact noise correction term ($\Delta\epsilon_\theta$) with a precise, mandatory coefficient:
>
> $$
> \Delta \epsilon_\theta = \gamma \cdot g(z_t) \cdot \left( - \frac{\sqrt{\bar{\alpha}_t}}{\sqrt{1-\bar{\alpha}_t}} \right)
> $$
>
> Unlike MGD3's empirical $\sigma_t$, our coefficient is strictly derived from the diffusion forward process. It ensures that our guidance remains physically consistent with the diffusion trajectory at every timestep, enabling the U-Net to maximize its predictive capability while the scheduler naturally handles the sample update.
>
> This theoretical consistency translates directly into operational robustness. While MGD3 suffers from catastrophic performance collapse when the parameter shifts (as explicitly shown in **their** Figure 10), CoDA maintains superior performance across an exceptionally broad range of $\gamma$ values (as demonstrated in Tables 12 and 13). This validates that our rigorous derivation yields a significantly more stable and effective guidance mechanism.

---

> ### Author Response · Authors · 2025-11-28
>
> Dear Reviewer ptsN,
>
> We appreciate the time you took to evaluate our work. Based on your valuable suggestions, we have refined the scope of the 'training-free' concept and extensively validated the transferability of CoDA to target-trained models.
>
> We hope that these revisions and our response meet your expectations. Please let us know if you have any further questions; we are eager to engage in further discussion.
>
> Best Regards,
>
> Authors of Submission 8388

---

### Official Review · Reviewer_ugRR · 2025-10-28

**Soundness:** 3
**Presentation:** 3
**Contribution:** 3
**Rating:** 4
**Confidence:** 4

**Summary:**

This paper proposes Core Distribution Alignment (CoDA), a "training-free" framework for Dataset Distillation (DD) that solves a key problem: prior methods either paradoxically require models pre-trained on the target dataset or fail due to a "distributional mismatch" when using general models. CoDA works in two stages: first, a Distribution Discovery pipeline uses VAE, UMAP, and HDBSCAN to find the "intrinsic core distribution" of the target data. Second, a Distribution Alignment stage guides an off-the-shelf text-to-image model (like SDXL) to generate samples matching this core distribution. This approach achieves state-of-the-art results, surpassing even target-trained methods on benchmarks like ImageNet-1K.

**Strengths:**

1.⁠ ⁠The paper tackles a fundamental, conceptual flaw in a popular research area and provides a complete, working solution. This "truly training-free" framework is what generative DD should have been from the start.

2.⁠ ⁠The 2-stage design is a key strength. The paper identifies that both discovering the right distribution and aligning to it are essential. The Distribution Discovery pipeline is a major contribution on its own.

3.⁠ ⁠⁠The discovery that the generated set Ours (G) can outperform the real set Ours (R)  is a highly significant finding. It suggests that the generative prior is not just a source of bias to be overcome, but a powerful tool that can be harnessed to "denoise" and "improve" real-world data for a specific task.

**Weaknesses:**

1.⁠ ⁠The authors disclose that the key hyperparameters for the Distribution Discovery stage (n_neighbors for UMAP and min_cluster_size for HDBSCAN) exhibit "drift" across different datasets. While the method remains robustly above the baseline, achieving peak performance requires a new, dataset-specific grid search. This somewhat undermines the "plug-and-play" nature of the "truly training-free" framework.

2.⁠ ⁠This entire chain is extremely brittle. A small, insignificant change in the initial UMAP embedding could cause a cluster to be "un-splittable" by Strategy 1, triggering a completely different logic path (e.g., Strategy 3 vs. Strategy 2), leading to a radically different set of representative samples. The paper does not analyze the stability or variance of the Distribution Discovery stage itself, which is a significant methodological gap given its complexity.

**Questions:**

1.⁠ ⁠The "intelligent refinement" shown in Figure 9 is a key finding. But does this always work? You use the polysemous word 'bonnet' as an example of T2I failure in Figure 7. What happens in a high-disagreement scenario where the guiding sample s_j is a hat, but the SDXL prior (activated during PIS) is strongly biased towards cars? Does your guidance successfully correct the model, or does the "refinement" step cause the image to revert to a car, making the generated dataset worse than the real one?

2.⁠ ⁠Why did you choose to build this highly complex, heuristic-driven pipeline on a feature space, rather than using a different off-the-shelf feature extractor that is also "training-free," such as a CLIP encoder, which is known to provide a much more semantically separate latent space? Did you experiment with this?

3.⁠ ⁠What analysis did you perform to measure the stability of this Distribution Discovery stage? How do we know this cascade of heuristics reliably converges to a high-quality set of samples, rather than just being a fragile mechanism that happened to work for your tested datasets?

---

> ### Author Response · Authors · 2025-11-24
>
> We sincerely thank Reviewer ugRR for the thorough evaluation and constructive criticism. We appreciate the reviewer's recognition of CoDA's "clear formulation," "strong empirical performance," and "efficient implementation." We are particularly grateful for the reviewer's incisive analysis regarding the definition of "training-free." The reviewer's observation that our method "shifts reliance from a domain-specific model to a more general pretrained one" is astute and perfectly captures the essence of our approach. This insight helped us clarify our core contribution: CoDA eliminates the prohibitive marginal cost of re-training for each new target domain. We have also carefully addressed the reviewer's questions regarding experimental settings, model transferability (to DiT), and dataset diversity (ImageWoof) by conducting comprehensive new experiments, as detailed below.
>
> ### **Question 1 (Analysis of Prior Injection)**
>
> We thank the reviewer for this insightful question regarding the mechanism of prior injection and refinement.
>
> To clarify, the SDXL prior operates via the standard Classifier-Free Guidance (CFG) mechanism, predicting noise conditioned on the current latent state.
>
> In the specific scenario where the target "bonnet" corresponds to a "hat" (meaning $s_j$ is a "hat"), the guidance mechanism plays a decisive role. During the initial generation steps (prior to PIS), $s_j$ acts as a strong geometric constraint, forcefully pulling the latent trajectory into the manifold of the "hat" class. Consequently, when the process enters the Prior Injection Steps (PIS), the latent is already structurally established within the target distribution. At this stage, the model's denoising prediction will naturally adhere to the existing visual structure (the hat), effectively overriding the initial textual ambiguity.
>
> This mechanism is empirically supported by Tables 8 and 22 in the revised paper. These results demonstrate the necessity of using a sufficiently strong $\gamma$ combined with an adequate number of guided steps (Total Steps - PIS). This configuration ensures that the latent is fully anchored in the target manifold before the prior takes over. This allows the model to provide its "denoising" and "refining" effects to enhance quality, without injecting its inherent bias. This synergy is precisely what enables the generated dataset (G) to outperform the representative real dataset (R).

---

> ### Author Response · Authors · 2025-11-24
>
> ### **Question 2 (Using CLIP feature space for the discovery stage)**
>
> We thank the reviewer for this profound question that touches upon the core principles of our work. We have adopted the reviewer's suggestion and conducted a complete and comprehensive ablation study to systematically compare and analyze the VAE space versus the CLIP space.
>
> The experimental results are presented in the table below.
>
> **Table 1: Ablation study on feature spaces and clustering methods. All results are for ImageIDC (IPC=10) and are the mean of three runs on ResNetAP10.**
>
> | Dim | Pre | Space | Kmeans | CoDA |
> | :---: | :---: | :---: | :---: | :---: |
> | 3136 | None | VAE | 46.0$\pm$1.1 | 45.8$\pm$0.9 |
> | 768 | None | CLIP | 51.8$\pm$1.2 | 52.4$\pm$0.7 |
> | 768 | PCA | VAE | 46.6$\pm$0.6 | 53.0$\pm$1.0 |
> | 50 | PCA | CLIP | 49.8$\pm$0.5 | 53.6$\pm$0.9 |
> | 50 | PCA | VAE | 51.7$\pm$0.8 | 54.6$\pm$0.6 |
> | 50 | UMAP | CLIP | 54.4$\pm$0.7 | 56.0$\pm$1.2 |
> | 50 | UMAP | VAE | 54.8$\pm$0.7 | **58.5$\pm$0.9** |
>
> The results demonstrate that the CoDA framework effectively adapts to completely different feature spaces, and its optimal result in the CLIP space (56.0%) is also on par with the previous SOTA MGD3 (55.9%). However, this peak performance is still surpassed by our result from the VAE space (58.5%).
>
> We further conducted qualitative and quantitative analyses on CLIP and VAE spaces, adding Appendix A.11 to the revised paper. The quantitative Mean $\sigma$ (the mean of kernel bandwidths from t-SNE) data is shown below:
>
> **Table 2: Manifold compactness metrics for different features spaces.**
>
> | Space | Dimension | Mean $\sigma$ |
> | :---: | :---: | :---: |
> | Original VAE | 3136 | 12.57 |
> | Original CLIP | 768 | 8.06 |
> | VAE (PCA) | 768 | 8.06 |
> | CLIP (PCA) | 50 | 6.33 |
> | CLIP (UMAP, "min_dist=0.0") | 50 | **0.29** |
>
> We believe there are three primary reasons why CLIP's peak performance does not match VAE's in our context:
>
> 1.  **Intrinsic Separation**: Contrary to common intuition, the original CLIP space does not exhibit stronger class separation than the VAE space when dimensionality is controlled. While CLIP may seem more separable, this is largely because its original dimension (768d) is much lower than the VAE's (3136d). When compared at the same 768 dimensions (via PCA for VAE), both visualization and Mean $\sigma$ metrics are nearly identical for both spaces.
> 2.  **Domain Gap**: The CLIP feature space ($latents_{clip}$) is structured for text-visual alignment. In contrast, our downstream models are purely visual and prefer prototypes selected in a space that prioritizes the visual manifold structure, which the VAE space better represents.
> 3.  **Encode Gap**: To guide SDXL generation, it is necessary to map $latents_{clip}$ back to images, and then re-encode them with the SDXL VAE to get $latents_{vae}$ ($s_j$). This "Encode Gap" from the indirect encoding degrades the peak performance.
>
> Furthermore, we reiterate our core design principle: maintaining maximal simplicity by relying solely on a single off-the-shelf T2I diffusion model (SDXL), using its native VAE and U-Net without introducing any additional feature extractors or architectural components. The strong SOTA performance achieved by CoDA demonstrates the robustness of this Discovery pipeline. Even within the VAE space, which presents challenging clustering properties, our pipeline successfully identifies an excellent set of representative samples that are perfectly compatible with the subsequent guided generation stage.

---

> ### Author Response · Authors · 2025-11-24
>
> ### **Weakness 1 (Robust analysis)**
>
> We thank the reviewer for this critical observation regarding robustness. While we acknowledge that the requirement for hyperparameter selection may impact the "plug-and-play" experience, our extensive analysis reveals that the hyperparameter drift in the Discovery Stage is minimal and incurs negligible computational cost.
>
> First, at IPC=10, the exact same parameters (n=85, s=55) were optimal for all five ImageNet-A to E subsets, where n is the n_neighbors of UMAP and s is the min_cluster_size of HDBSCAN.
>
> Second, as can be seen from the results of the small-scale parameter search on ImageNet-1k in Table 3, the maximum fluctuation in results across different parameter configurations is only 0.742% (42.5 - 41.758). And n=85, s=55 is still optimal. We can call this the "universal optimal parameter."
>
> **Table 3: Small-scale parameter search on ImageNet-1k.**
>
> | Parameter Configuration | Result |
> | :---: | :---: |
> | n_neighbours=85 & size_min=50 | 42.226 |
> | n_neighbours=85 & size_min=55 | 42.523 |
> | n_neighbours=85 & size_min=60 | 41.954 |
> | n_neighbours=90 & size_min=55 | 41.866 |
> | n_neighbours=90 & size_min=60 | 41.758 |
> | n_neighbours=90 & size_min=65 | 42.016 |
> | n_neighbours=90 & size_min=70 | 42.142 |
>
> Further small-scale parameter searches on ImageIDC and ImageNette show that ImageIDC achieves optimal results at n=90, s=55, and ImageNette achieves optimal at n=85, s=60. It can be seen that the parameter drift is not large.
>
> More importantly, in the practice of domain change to Places365, we searched a 4x4 configuration matrix in the universal optimal parameter range as in Table 4. The optimal Ours(R) result was achieved at n=90, s=60. This optimal configuration is still within the small window of the universal optimal parameters.
>
> **Table 4: Parameter search when adapting to Places365.**
>
> | | size_min=55 | size_min=60 | size_min=65 | size_min=70 |
> | :---: | :---: | :---: | :---: | :---: |
> | **n_neighbours=75** | 43.33 | 42.33 | 43.13 | 43.13 |
> | **n_neighbours=80** | 44.27 | 43.07 | 43.87 | 44.07 |
> | **n_neighbours=85** | 41.20 | 41.60 | 42.73 | 43.60 |
> | **n_neighbours=90** | 43.20 | 44.37 | 43.87 | 44.23 |
>
> At the same time, we want to reiterate that the fine-tuning of CoDA's discovery stage runs entirely on CPU. Therefore, being able to complete a completely different domain transfer through this small-scale parameter search on CPU is reasonable and worthwhile.
>
> **Future Work**
>
> That said, automating this process is a key objective for our future work. We aim to develop a lightweight method—such as a small neural network—to map the target dataset's latent feature distribution directly to the optimal clustering parameters, thereby eliminating the parameter search effort entirely.
>
> ### **Weakness 2 & Question 3 (Stability Analysis of the Discovery Pipeline)**
>
> We thank the reviewer for this critical question regarding the stability of our pipeline. We agree that small perturbations, especially from the UMAP random seed, can influence the subsequent process. However, we respectfully disagree with the characterization of the pipeline as "brittle." The multi-strategy design (Split, ForcedSplit, KMeans) was intentionally designed not as a fragile serial chain, but as a set of complementary and adaptive compensation mechanisms. This design ensures that minor initial perturbations are self-corrected and do not cascade or amplify to cause significant differences in the final result.
>
> ### **Correcting the Logic of Strategy Switching**
>
> The reviewer's concern that a small perturbation could cause Strategy 1 to fail and lead to an uncertain takeover by Strategy 2 or 3 stems from a misunderstanding of the triggering conditions.
>
> The switch between Strategy 2 (KMeans on outliers) and Strategy 3 (ForcedSplit) is governed by a precise condition: the number of points identified as outliers by HDBSCAN.
>
> Strategy 3 (ForcedSplit) is a fallback mechanism, triggered only when the number of outliers is negligible. Specifically, the threshold is defined as:
> $$ \mathrm{num\_outliers} < 5 \times (\mathrm{samples\_needed} - \mathrm{samples\_acquired}) $$
>
> In the vast majority of cases, the number of actual outliers exceeds this threshold by two orders of magnitude (approx. $100\times$). Consequently, the process will stably default to Strategy 2. Strategy 3 is reserved solely for the extreme edge case where Strategy 1 fails to split a large cluster and almost no outliers remain.
>
> Therefore, a minor perturbation implies zero risk of triggering an unintended mode switch, underpinning the robustness of our pipeline.

---

> ### Author Response · Authors · 2025-11-24
>
> ### **Experimental Proof of Stability**
>
> We conducted a controlled experiment using varying UMAP random seeds to simulate the minor perturbations encountered during the discovery stage. We measured the final performance of the real dataset (R) and tracked the source of the samples (Init, Split, KMeans, ForcedSplit). The results are as follows:
>
> **Table 5: Stability analysis with UMAP seed. 'Overall Result' refers to the mean and standard deviation of results across five seeds.**
>
> | Seed | Init | Split | Kmeans | Forced Split | Individual Result | Overall Result |
> | :---: | :---: | :---: | :---: | :---: | :---: | :---: |
> | 1 | 34 | 9 | 52 | 5 | 50.8 | 50.16$\pm$0.43 |
> | 2 | 32 | 13 | 50 | 5 | 50.4 | 50.16$\pm$0.43 |
> | 3 | 34 | 11 | 50 | 5 | 50.2 | 50.16$\pm$0.43 |
> | 4 | 31 | 16 | 49 | 4 | 49.6 | 50.16$\pm$0.43 |
> | 5 | 30 | 15 | 50 | 5 | 49.8 | 50.16$\pm$0.43 |
>
> **Table 6: Source of the samples (Init / Split / KMeans / ForcedSplit).**
>
> | Class | seed 1 | seed 2 | seed 3 | seed 4 | seed 5 |
> | :---: | :---: | :---: | :---: | :---: | :---: |
> | 0 | 3 / 0 / 7 / 0 | 3 / 0 / 7 / 0 | 4 / 0 / 6 / 0 | 3 / 0 / 7 / 0 | 3 / 0 / 7 / 0 |
> | 1 | 6 / 0 / 4 / 0 | 6 / 0 / 4 / 0 | 3 / 2 / 5 / 0 | 7 / 0 / 3 / 0 | 2 / 4 / 4 / 0 |
> | 2 | 2 / 4 / 4 / 0 | 2 / 4 / 4 / 0 | 3 / 3 / 4 / 0 | 2 / 4 / 4 / 0 | 3 / 3 / 4 / 0 |
> | 3 | 4 / 0 / 6 / 0 | 4 / 1 / 5 / 0 | 4 / 0 / 6 / 0 | 4 / 0 / 6 / 0 | 2 / 2 / 6 / 0 |
> | 4 | 3 / 0 / 7 / 0 | 3 / 0 / 7 / 0 | 2 / 3 / 5 / 0 | 2 / 3 / 5 / 0 | 5 / 0 / 5 / 0 |
> | 5 | 4 / 0 / 6 / 0 | 3 / 2 / 5 / 0 | 4 / 0 / 6 / 0 | 4 / 0 / 6 / 0 | 4 / 0 / 6 / 0 |
> | 6 | 4 / 0 / 6 / 0 | 3 / 1 / 6 / 0 | 5 / 0 / 5 / 0 | 3 / 1 / 6 / 0 | 3 / 1 / 6 / 0 |
> | 7 | 2 / 3 / 0 / 5 | 2 / 3 / 0 / 5 | 2 / 3 / 0 / 5 | 2 / 4 / 0 / 4 | 2 / 3 / 0 / 5 |
> | 8 | 4 / 0 / 6 / 0 | 4 / 0 / 6 / 0 | 4 / 0 / 6 / 0 | 2 / 2 / 6 / 0 | 4 / 0 / 6 / 0 |
> | 9 | 2 / 2 / 6 / 0 | 2 / 2 / 6 / 0 | 3 / 0 / 7 / 0 | 2 / 2 / 6 / 0 | 2 / 2 / 6 / 0 |
>
> This new analysis demonstrates:
> 1.  **Highly Stable Performance**: The final accuracy is remarkably stable, at 50.16 $\pm$ 0.43.
> 2.  **Adaptive Compensation**: The impact of the UMAP seed is minimal. The variations are typically controlled to within a 1-sample difference per class. For instance, a seed might result in one fewer 'Init' sample, but this is immediately compensated by the 'Split' strategy identifying an additional one.
> 3.  **Consistent Logic Path**: Most importantly, the feared chaotic jump between Strategy 2 and Strategy 3 did not occur. As shown in the class-wise breakdown, the strategy selection for each specific class remains invariant across seeds. For example, specific classes (like Class 7 in our logs) consistently utilize Strategy 3 due to their intrinsic density properties, while others consistently utilize Strategy 2. This confirms that the logic path is dictated by robust data properties, not random noise.
>
> This stability is further supported by Figure 4 in the main paper. Even when significant perturbations occur, for example, the min_cluster_size parameter is varied dramatically, causing the source of the finally obtained samples to be significantly different. The final performance remains robustly above the SOTA baseline across a vast range. We also note that the optimal performance often occurs precisely when these strategies work in synergy.
>
> ### **Conclusion**
>
> We believe this robustness stems from each strategy effectively handling its inputs: Strategies 1 & 3 (density-based HDBSCAN) safely refine large features, while Strategy 2 (KMeans) provides a perfect complement by extracting details from outliers that HDBSCAN intentionally ignores.
>
> While we acknowledge that we rely on downstream accuracy rather than an explicit mathematical tool to measure parameter "goodness," the evidence overwhelmingly refutes the concern that CoDA is a "fragile mechanism."
>
> To directly address the reviewer's concern that our method might merely "happen to work for the tested datasets," we highlight our expanded evaluation. CoDA establishes a new SOTA in all experimental configurations across eight distinct datasets and enables zero-cost transfer to completely different domains, such as the scene dataset Places365 (Table 7 in revised paper) and even the specialized medical imaging dataset DRIVE (Appendix A.9 in revised paper). This consistent performance across such diverse domains proves that CoDA is a robust, generalizable framework, not a fragile heuristic overfitted to a single dataset.

---

> ### Author Response · Authors · 2025-11-28
>
> Dear Reviewer ugRR,
>
> We sincerely thank you for the time and effort dedicated to reviewing our work. We found your feedback, particularly concerning the stability of the discovery pipeline and the feature space selection, to be extremely valuable in strengthening our submission.
>
> We are writing to respectfully inquire if our response and the new experimental results have resolved your concerns. We remain fully available for further discussion should you have any additional questions.
>
> Best Regards,
>
> Authors of Submission 8388

---

### Official Review · Reviewer_NgTi · 2025-10-30

**Soundness:** 3
**Presentation:** 3
**Contribution:** 3
**Rating:** 6
**Confidence:** 4

**Summary:**

This paper presents CoDA, a training-free dataset distillation framework that leverages an off-the-shelf text-to-image diffusion model to generate distilled data that is better aligned with the target dataset. The key idea is to first discover an intrinsic, high-density “core” distribution in the latent space of the target data and then guide the diffusion process so that the synthesized samples follow this core distribution. Experiments on ImageNet-1K and its subsets indicate that this distribution-aware generation can match or outperform several existing, diffusion-based dataset distillation methods that require training or fine-tuning on the target data.

**Strengths:**

1. The paper clearly identifies a practical gap in current diffusion-based dataset distillation: many approaches rely on a diffusion model trained on the very dataset they aim to compress. CoDA directly addresses this by using only an off-the-shelf model.

2. The two-stage pipeline—(i) discovering a core distribution in latent space and (ii) aligning the diffusion generation to that distribution—matches the problem formulation and makes the contribution transparent.

3. The use of latent embedding, dimensionality reduction, and density-based clustering allows the method to pick representative, high-density samples instead of relying on naïve per-class sampling.

4. The method is tested on multiple ImageNet variants, different IPC regimes, and different evaluation settings, and it consistently shows advantages over other training-free or general-model baselines.

5. The ablation studies demonstrate that most of the gains come from closing the distribution gap rather than from task-specific training, which supports the main claim of the paper.

**Weaknesses:**

1. Although the approach does not train a diffusion model, the discovery stage (encoding, dimensionality reduction, clustering) plus guided sampling still incurs noticeable computational cost. A more explicit comparison of wall-clock time and memory with key baselines would make the “training-free” claim more convincing.

2. The quality of the discovered core distribution may depend on settings for the dimensionality reduction and clustering steps. The paper suggests that different datasets may require slightly different configurations, which could limit robustness.

3. The method is closely tied to the latent geometry of the chosen text-to-image model. It is not yet clear how portable the approach is to other diffusion architectures or to future models with different encoders.

**Questions:**

1. Can you report end-to-end runtime (discovery + generation) per dataset and per IPC, and compare it with both target-trained diffusion distillation methods and recent training-free approaches?

2. How robust is the distribution discovery stage when the off-the-shelf model’s latent space is not well aligned with the target domain (e.g., domain shift, fine-grained categories)?

3. Is it possible to reuse or amortize the discovered core distribution across related datasets or class subsets to reduce the discovery cost?

4. How does the method behave when labels are noisy or incomplete during the discovery stage?

5. For higher-resolution generation, do you require per-class retuning of guidance strength, or is there a single configuration that works across classes?

---

> ### Author Response · Authors · 2025-11-24
>
> We sincerely thank Reviewer NgTi for the comprehensive review and accurate summary of our work. We are particularly encouraged by the reviewer's recognition that CoDA effectively addresses the limitations of prior methods by leveraging only an off-the-shelf model. We also deeply appreciate the positive assessment of our high-density clustering mechanism and reviewer's observation that our approach consistently demonstrates advantages over other training-free baselines. Most importantly, we are grateful that the reviewer highlighted the core insight of our research: that the performance gains derive from closing the distribution gap rather than from task-specific training. We agree that the questions raised regarding computational cost, robustness analysis, sample reuse, handling noisy or incomplete labels, and portability are critical for establishing the practicality of our method. We have addressed these points comprehensively in the following response.
>
> ### **Question 1 & Weakness 1 (Computational Cost)**
>
> We thank the reviewer for highlighting this critical aspect. We agree that a transparent efficiency comparison is vital to substantiate our claims. Our conclusion is that CoDA maintains high efficiency comparable to existing training-free SOTAs while completely eliminating the prohibitive pre-training costs that target-trained methods incur when adapting to new domains.
>
> To quantitatively support this, we have conducted a comprehensive efficiency analysis, including end-to-end runtime, resource utilization, and a comparison against both training-free D4M and target-trained MGD3 SOTA methods.
>
> **Table 1: End-to-end efficiency and resource comparison (ImageNet-1K, IPC=50). "Change Domain" is the cost to adapt to a new dataset (e.g., Places365).**
>
> | Type | Step | Method | Encode (VRAM) | Discovery (RAM) | Result ($\mathcal{R}$) | Generate (VRAM) | End-to-End Time | Result ($\mathcal{G}$) | Change Domain |
> | :--- | :---: | :--- | :---: | :---: | :---: | :---: | :---: | :---: | :---: |
> | General-model (SDXL) | 50 | D4M | 1.5h (1553MiB) | 0.42h (1996 MiB) | - | 2.7s/img (12323MiB) | 39.4h | 55.2$\pm$0.1 | 0 |
> | General-model (SDXL) | 50 | MGD3 | 1.5h (1553MiB) | 0.42h (1996 MiB) | - | 2.7s/img (12323MiB) | 39.4h | 57.0$\pm$0.2 | 0 |
> | General-model (SDXL) | 50 | CoDA(PCA) | 1.5h (1553MiB) | 0.42h (2019 MiB) | 55.4$\pm$0.2 | 2.7s/img (12323MiB) | 39.4h | 57.8$\pm$0.2 | 0 |
> | General-model (SDXL) | 50 | CoDA(UMAP) | 1.5h (1553MiB) | 2.17h (2144 MiB) | 58.2$\pm$0.2 | 2.7s/img (12323MiB) | 41.2h | 60.4$\pm$0.2 | 0 |
> | General-model (SDXL) | 25 | CoDA(UMAP) | 1.5h (1553MiB) | 2.17h (2144 MiB) | 58.2$\pm$0.2 | 1.3s/img (12323MiB) | 21.7h | 60.1$\pm$0.2 | 0 |
> | Target-Trained (DiT) | 50 | MGD3 | 1.5h (1553MiB) | 0.42h (1996 MiB) | - | 1.2s/img (2796MiB) | 18.6h | 60.2$\pm$0.2 | >1000 TPU-days |
>
> **Table 2: Breakdown of the Distribution Discovery stage (224x224, on ImageIDC IPC=50).**
>
> | Type | Preprocess-time | Initial HDBSCAN | Strategy 1: SplitCluster | Strategy 2: Clustering Outliers | Strategy 3: ForcedSplit |
> | :--- | :---: | :---: | :---: | :---: | :---: |
> | UMAP | 72.273 s | 0.969 s | 1.156 s | 0.083 s | 0.200 s |
> | PCA | 0.740 s | 1.592 s | 0.002 s | 0.157 s | 0.000 s |
>
> As detailed in the tables above and our analysis:
>
> **a. Encode Stage:** Practical tests show no significant difference in runtime efficiency or VRAM usage between the different VAE models used by CoDA and baselines.
>
> **b. Discovery Stage:** While UMAP preprocessing constitutes the primary computational cost, it is indispensable for performance (proven in revised paper Appendix A.11). Furthermore, when UMAP is replaced by PCA, the efficiency becomes indistinguishable from the baselines. This indicates that our subsequent density-based mechanism, although logically complex, runs as efficiently as a direct single-step K-Means when operating in low-dimensional space.
>
> **c. Generate Stage:** While the DiT model used by MGD3 is faster, CoDA with 25 inference steps approaches this efficiency without performance degradation.
>
> **d. Domain Switching Cost:** Most importantly, target-trained methods incur a "hidden debt" of thousands of TPU-days whenever the domain changes. In contrast, CoDA's cost to switch domains is zero (proven in revised paper Appendix A.9).
>
> We have added further analysis to Appendix A.10 (Further Computational Efficiency Analysis) of the revised paper.

---

> ### Author Response · Authors · 2025-11-24
>
> ### **Question 2 & Weakness 2 (Robust analysis)**
>
> We thank the reviewer for this critical question regarding the robustness of our method. We agree that a comprehensive analysis summarizing performance across different granularities, domains, and hyperparameter settings is essential for establishing the practicality of CoDA. Through the following studies, we conclude that the discovery stage is capable of adaptively maintaining robustness and efficiently handling diverse datasets.
>
> ### **Robustness to Fine-Grained Categories**
>
> Fine-grained difficulty can be measured by the challenge of correct classification. The ImageNet-A through ImageNet-E datasets, introduced in GLaD (Cazenavette et al., 2023; Generalizing dataset distillation via deep generative prior), were collected based on varying levels of classification difficulty. Furthermore, ImageWoof has highest inter-class similarity, consisting of 10 classes: Australian terrier, Border terrier, Samoyed, Beagle, Shih-Tzu, English foxhound, Rhodesian ridgeback, Dingo, Golden retriever, and Old English sheepdog.
>
> We have added a new, comprehensive evaluation on the ImageWoof subset, as shown in the Table 3. Our experiments now cover 8 ImageNet subsets, which inherently represent different levels of fine-grained difficulty.
>
> **Table 3: Performance comparison with pre-trained diffusion models and other SOTA methods on ImageWoof.** All the results are reproduced by us for the 256x256 resolution. The missing results are due to out-of-memory. Results shown for the previous works are from MGD3. **The best result is marked in bold.** The arrow annotations indicate the performance difference between CoDA and the previous SOTA.
>
> | IPC | Test Model | Random | Herding | DiT | DM | IDC-1 | GLaD | Minimax | MGD3 | CoDA(Ours) | Full |
> | :--- | :--- | :---: | :---: | :---: | :---: | :---: | :---: | :---: | :---: | :---: | :---: |
> | 10 (0.8%) | ConvNet-6 | 24.3$\pm$1.1 | 26.7$\pm$0.5 | 34.2$\pm$1.1 | 26.9$\pm$1.2 | 33.3$\pm$1.1 | 33.8$\pm$0.9 | **37.0$\pm$1.0** | 34.7$\pm$1.1 | 35.7$\pm$1.4 ($\downarrow$1.3) | 86.4$\pm$0.2 |
> | 10 (0.8%) | ResNetAP-10 | 29.4$\pm$0.8 | 32.0$\pm$0.3 | 34.7$\pm$0.5 | 30.3$\pm$1.2 | 39.1$\pm$0.5 | 32.9$\pm$0.9 | 39.2$\pm$1.3 | **40.4$\pm$1.9** | 39.2$\pm$0.7 ($\downarrow$1.2) | 87.5$\pm$0.5 |
> | 10 (0.8%) | ResNet-18 | 27.7$\pm$0.9 | 30.2$\pm$1.2 | 34.7$\pm$0.4 | 33.4$\pm$0.7 | 37.3$\pm$0.2 | 31.7$\pm$0.8 | 37.6$\pm$0.9 | 38.5$\pm$2.5 | **38.8$\pm$1.3** ($\uparrow$0.3) | 89.3$\pm$1.2 |
> | 20 (1.6%) | ConvNet-6 | 29.1$\pm$0.7 | 29.5$\pm$0.3 | 36.1$\pm$0.8 | 29.9$\pm$1.0 | 35.5$\pm$0.8 | - | 37.6$\pm$0.9 | **39.0$\pm$3.5** | 38.1$\pm$1.3 ($\downarrow$0.9) | 86.4$\pm$0.2 |
> | 20 (1.6%) | ResNetAP-10 | 32.7$\pm$0.4 | 34.9$\pm$0.1 | 41.1$\pm$0.8 | 35.2$\pm$0.6 | 43.4$\pm$0.3 | - | **45.8$\pm$0.5** | 43.0$\pm$1.6 | 42.5$\pm$0.6 ($\downarrow$3.3) | 87.5$\pm$0.5 |
> | 20 (1.6%) | ResNet-18 | 29.7$\pm$0.5 | 32.2$\pm$0.6 | 40.5$\pm$0.5 | 29.8$\pm$1.7 | 38.6$\pm$0.2 | - | **42.5$\pm$0.6** | 41.9$\pm$2.1 | 42.0$\pm$0.8 ($\downarrow$0.5) | 89.3$\pm$1.2 |
> | 50 (3.8%) | ConvNet-6 | 41.3$\pm$0.6 | 40.3$\pm$0.7 | 46.5$\pm$0.8 | 44.4$\pm$1.0 | 43.9$\pm$1.2 | - | 53.9$\pm$0.6 | 54.5$\pm$1.6 | **56.8$\pm$0.9** ($\uparrow$2.3) | 86.4$\pm$0.2 |
> | 50 (3.8%) | ResNetAP-10 | 47.2$\pm$1.3 | 49.3$\pm$0.7 | 49.3$\pm$0.2 | 47.1$\pm$1.1 | 48.3$\pm$1.0 | - | 56.3$\pm$1.0 | 56.5$\pm$1.9 | **59.4$\pm$1.0** ($\uparrow$2.9) | 87.5$\pm$0.5 |
> | 50 (3.8%) | ResNet-18 | 47.9$\pm$1.8 | 48.3$\pm$1.2 | 50.1$\pm$0.5 | 46.2$\pm$0.6 | 48.3$\pm$0.8 | - | 57.1$\pm$0.6 | 58.3$\pm$1.4 | **61.2$\pm$0.9** ($\uparrow$2.9) | 89.3$\pm$1.2 |
> | 70 (5.4%) | ConvNet-6 | 46.3$\pm$0.6 | 46.2$\pm$0.6 | 50.1$\pm$1.2 | 47.5$\pm$0.8 | 48.9$\pm$0.7 | - | 55.7$\pm$0.9 | 55.1$\pm$2.5 | **56.4$\pm$1.2** ($\uparrow$1.3) | 86.4$\pm$0.2 |
> | 70 (5.4%) | ResNetAP-10 | 50.8$\pm$0.6 | 53.4$\pm$1.4 | 54.3$\pm$0.9 | 51.7$\pm$0.8 | 52.8$\pm$1.8 | - | 58.3$\pm$0.2 | 60.2$\pm$2.4 | **61.2$\pm$0.7** ($\uparrow$1.0) | 87.5$\pm$0.5 |
> | 70 (5.4%) | ResNet-18 | 52.1$\pm$1.0 | 49.7$\pm$0.8 | 51.5$\pm$1.0 | 51.9$\pm$0.8 | 51.1$\pm$1.7 | - | 58.8$\pm$0.7 | 59.7$\pm$2.7 | **62.4$\pm$1.4** ($\uparrow$2.7) | 89.3$\pm$1.2 |
> | 100 (7.7%) | ConvNet-6 | 52.2$\pm$0.4 | 54.4$\pm$1.1 | 53.4$\pm$0.3 | 55.0$\pm$1.3 | 53.2$\pm$0.9 | - | 61.1$\pm$0.7 | 60.1$\pm$1.2 | **62.4$\pm$0.7** ($\uparrow$1.3) | 86.4$\pm$0.2 |
> | 100 (7.7%) | ResNetAP-10 | 59.4$\pm$1.0 | 61.7$\pm$0.9 | 58.3$\pm$0.8 | 56.4$\pm$0.8 | 56.1$\pm$0.9 | - | 64.5$\pm$0.2 | 66.5$\pm$1.0 | **67.6$\pm$0.8** ($\uparrow$1.1) | 87.5$\pm$0.5 |
> | 100 (7.7%) | ResNet-18 | 61.5$\pm$1.3 | 59.3$\pm$0.7 | 58.9$\pm$1.3 | 60.2$\pm$1.0 | 58.3$\pm$1.2 | - | 65.7$\pm$0.4 | 68.8$\pm$0.7 | **71.4$\pm$1.4** ($\uparrow$2.6) | 89.3$\pm$1.2 |

---

> ### Author Response · Authors · 2025-11-24
>
> CoDA establishes a new SOTA in all experimental configurations across these eight datasets, only achieving second-best results in the low-data (IPC$\le$20) settings on ImageWoof. This demonstrates CoDA's robustness to datasets with varying degrees of fine-grained detail.
>
> The ImageWoof experiments have been added to Section 4.2 of the revised paper.
>
> ### **Robustness to Domain Shift**
>
> Our original paper already showed CoDA's ability to generalize from ImageNet (objects) to Places365 (scenes) without modification (Table 7 in revised paper).
>
> To further validate this, we have added a new case in Appendix A.9 (Case Study on Zero-Cost Domain Change: Drive) in the revised paper. This experiment demonstrates that CoDA can be seamlessly applied to a completely different and highly specialized medical imaging domain (retinal vessel segmentation), without any re-training or manual intervention, a capability that is unattainable for target-trained methods.
>
> ### **Robustness of Hyperparameters**
>
> **Discovery Stage**
>
> We found the discovery parameters to be highly stable. For instance, at IPC=10, the exact same parameters (n=85, s=55) were optimal for all five ImageNet-A to E subsets, where n is the n_neighbors of UMAP and s is the min_cluster_size of HDBSCAN.
>
> As can be seen from the results of the small-scale parameter search on ImageNet-1k in Table 4, the maximum fluctuation in results across different parameter configurations is only 0.742% (42.5 - 41.758). And n=85, s=55 is still optimal. We can call this the "universal optimal parameter."
>
> **Table 4: Small-scale parameter search on ImageNet-1k.**
>
> | Parameter Configuration | Result |
> | :--- | :--- |
> | n_neighbours=85 & size_min=50 | 42.226 |
> | n_neighbours=85 & size_min=55 | 42.523 |
> | n_neighbours=85 & size_min=60 | 41.954 |
> | n_neighbours=90 & size_min=55 | 41.866 |
> | n_neighbours=90 & size_min=60 | 41.758 |
> | n_neighbours=90 & size_min=65 | 42.016 |
> | n_neighbours=90 & size_min=70 | 42.142 |
>
> Further small-scale parameter searches on ImageIDC and ImageNette show that ImageIDC achieves optimal results at n=90, s=55, and ImageNette achieves optimal at n=85, s=60. It can be seen that the parameter drift is not large.
>
> Furthermore, in the practice of domain change to Places365, we searched a 4x4 configuration matrix in the universal optimal parameter range as in Table 5. The optimal Ours(R) result was achieved at n=90, s=60. This optimal configuration is still within the small window of the universal optimal parameters.
>
> **Table 5: Parameter search when adapting to Places365.**
>
> | | size_min=55 | size_min=60 | size_min=65 | size_min=70 |
> | :--- | :---: | :---: | :---: | :---: |
> | **n_neighbours=75** | 43.33 | 42.33 | 43.13 | 43.13 |
> | **n_neighbours=80** | 44.27 | 43.07 | 43.87 | 44.07 |
> | **n_neighbours=85** | 41.20 | 41.60 | 42.73 | 43.60 |
> | **n_neighbours=90** | 43.20 | 44.37 | 43.87 | 44.23 |
>
> At the same time, we want to reiterate that the fine-tuning of CoDA's discovery stage runs entirely on CPU. Therefore, being able to complete a completely different domain transfer through this small-scale parameter search on CPU is reasonable and worthwhile.
>
> **Alignment stage**
>
> We found that the model's prior knowledge for each class varies, so in a detailed parameter search, we did find that the optimal configuration changes for each dataset.
>
> However, the final performance remains highly robust. As Appendix A.7 and Tables 8 & 9 in the revised main paper demonstrate, the final performance exceeds the SOTA across a very large range of parameters.
>
> Therefore, for our experiments, we unified the generation parameters to $\gamma=0.10$, PIS=5 for 50 inference steps (and $\gamma=0.05$, PIS=2 for 25 inference steps), as this unified configuration generally yields competitive performance.

---

> ### Author Response · Authors · 2025-11-24
>
> ### **Question 3 (Reuse Samples)**
>
> We thank the reviewer for this insightful question. The answer is affirmative. Our discovered core distribution can be fully reused and amortized.
>
> As explicitly stated in Section 3.1 (subsection "Our density-based pipeline"), our discovery process is applied independently to each class.
>
> This means that once we run the discovery process for the full ImageNet-1K (1000 classes) and save the results, identifying core samples for any subset (such as ImageIDC or ImageNet-A) requires no recomputation. We simply retrieve the corresponding core samples for those classes from the storage. The results obtained in this manner are identical to running the discovery process on the subset alone. Consequently, the discovery-stage cost for all included subsets is effectively reduced to zero.
>
> ### **Question 4 (Noisy/incomplete label)**
>
> We thank the reviewer for raising these important practical considerations regarding data quality.
>
> ### **Noisy Labels**
>
> CoDA is inherently designed to handle noisy labels. This was a primary motivation for building our pipeline based on HDBSCAN rather than directly using K-Means.
>
> K-Means lacks intrinsic outlier detection capabilities and requires additional, external outlier removal steps, such as LOF. In contrast, HDBSCAN is fundamentally density-based, making it inherently robust to noise. For instance, a 'dog' image erroneously labeled as 'cat' will typically reside in an extremely low-density region within the 'cat' VAE feature space. HDBSCAN will automatically identify this point as noise and will not select it as part of the core sample set.
>
> To validate this, we conducted a visualization experiment on the class "n02102040 English springer", where we introduced noise by mixing in 10 randomly sampled images from the class "n02123045 Tabby cat".
>
> As shown in Figure 16 of the revised paper, none of the noisy samples (Tabby cats, marked in red) are located in absolute high-density regions. Since our selection mechanism strictly targets the highest-density cores to form $S_r$, these noisy data points are naturally bypassed without requiring manual intervention.
>
> ### **Incomplete Labels**
>
> The robustness of CoDA also enables it to handle incomplete labels efficiently, extending its applicability to semi-supervised dataset distillation.
>
> To illustrate this, we simulate a semi-supervised setting on the ImageIDC dataset and construct three configurations:
> 1.  **Full Supervision**: The standard setting using 100% of the labeled data.
> 2.  **Limited Labels (10%)**: We randomly select only 10% of images (130 images per class) as the labeled set, treating the remaining 90% as unseen.
> 3.  **Semi-Supervised (10% + Pseudo)**: We utilize the 10% labeled set to guide the utilization of the 90% unlabeled data via pseudo-labeling. Specifically, we calculate the class centers of the 10% labeled data in the CLIP feature space. We then perform a global similarity search for the unlabeled samples against these centers. An unlabeled sample is assigned a pseudo-label and added to the pool if its cosine similarity to a class center exceeds a confidence threshold of 0.85.
>
> The experimental results are presented in the table below.
>
> **Table 6: Performance comparison on ImageIDC (IPC=50) under different label availability settings. The test model is ResNetAP-10. Results are the mean and standard deviation of three independent runs.**
>
> | Method | Label Availability | Results |
> | :---: | :---: | :---: |
> | Random | Full Supervision | 68.1$\pm$0.7 |
> | D4M | Full Supervision | 63.8$\pm$0.4 |
> | MGD3 | Full Supervision | 72.1$\pm$0.8 |
> | CoDA | Full Supervision | **77.6$\pm$0.6** |
> | CoDA | Limited Labels (10%) | 70.6$\pm$0.8 |
> | CoDA | Semi-Supervised (10% + Pseudo) | 76.0$\pm$0.8 |
>
> While using only 10% of the labels leads to a performance drop (70.6%), employing our pseudo-labeling strategy significantly recovers performance to 76.0%. This not only surpasses the previous SOTA MGD3 by a large margin (3.9%), but also approaches the performance of CoDA under full supervision (77.6%). This indicates that CoDA can effectively discover the core distribution even with incomplete labels.
>
> The experiments and discussions related to noisy and incomplete labels have been added to Appendix A.15 of the revised paper.

---

> ### Author Response · Authors · 2025-11-24
>
> ### **Question 5 (Generation at different resolutions)**
>
> We thank the reviewer for this question regarding the scalability and configuration of our generation process.
>
> As detailed in our response to **Question 2 & Weakness 2 (Robustness analysis)**, we employed a unified parameter configuration for the alignment (generation) stage across all experiments, avoiding any per-class parameter tuning.
>
> This consistency extends to generation at both lower ($128 \times 128$) and higher ($1024 \times 1024$) resolutions. We applied the identical unified configuration of $\text{PIS}=5, \gamma=0.10$ across all classes, without any fine-grained, per-class adjustments.
>
> Consequently, we established the first $1024 \times 1024$ resolution baseline for Dataset Distillation using this exact unified configuration, as reported in Table 5 of the revised paper.
>
> ### **Weakness 3 (Portability of CoDA to different encoders or generator architectures)**
>
> We thank the reviewer for raising this valuable point regarding the generalizability of our framework. We argue that the CoDA framework is indeed portable to other diffusion architectures and encoders, and we have added two new experiments to the revised paper to demonstrate this.
>
> ### **Different Encoders**
>
> We replaced the VAE encoder with a CLIP encoder to test the adaptability of our discovery stage. The experimental results are presented in the table below.

---

> ### Author Response · Authors · 2025-11-24
>
> **Table 7: Ablation study on feature spaces and clustering methods. All results are for ImageIDC (IPC=10) and are the mean of three runs on ResNetAP10.**
>
> | Dim | Pre | Space | Kmeans | CoDA |
> | :---: | :---: | :---: | :---: | :---: |
> | 3136 | None | VAE | 46.0$\pm$1.1 | 45.8$\pm$0.9 |
> | 768 | None | CLIP | 51.8$\pm$1.2 | 52.4$\pm$0.7 |
> | 768 | PCA | VAE | 46.6$\pm$0.6 | 53.0$\pm$1.0 |
> | 50 | PCA | CLIP | 49.8$\pm$0.5 | 53.6$\pm$0.9 |
> | 50 | PCA | VAE | 51.7$\pm$0.8 | 54.6$\pm$0.6 |
> | 50 | UMAP | CLIP | 54.4$\pm$0.7 | 56.0$\pm$1.2 |
> | 50 | UMAP | VAE | 54.8$\pm$0.7 | **58.5$\pm$0.9** |
>
> The results demonstrate that the CoDA framework effectively adapts to completely different feature spaces, and its optimal result in the CLIP space (56.0%) is also on par with the previous SOTA MGD3 (55.9%). However, this peak performance is still surpassed by our result from the VAE space (58.5%).
>
> We believe the reason for this continued high efficiency is that using a different encoder simply changes the resulting latent feature distribution, while our discovery pipeline was designed to make almost no assumptions about it. It relies on UMAP to automatically optimize the latent manifold, followed by density-based methods to find representative samples. The multi-strategy synergy of the discovery pipeline ensures it can robustly sample representative points regardless of the initial latent distribution. Therefore, CoDA can work effectively in different feature spaces.
>
> We have added this analysis to the ablation study section "Transferring CoDA to Different Encoders" in the revised paper.
>
> ### **Different Generator Architectures**
>
> To extensively test the adaptability of our alignment stage, we replaced the base SDXL model with two distinct SOTA models:
> 1.  DiT-XL/2: A Transformer-based diffusion model pre-trained on ImageNet (Target-Trained).
> 2.  Stable Diffusion 3 (SD3): A general-purpose model based on the MMDiT architecture that employs a Rectified Flow matching objective (Training-Free).
>
> **Table 8: Adaptability of CoDA across different generator architectures (DiT & SD3). All results are the mean of three runs on ResNetAP10.**
>
> | Dataset | IPC | SD3 | DiT | DiT+MGD3 | SD3+CoDA | DiT+CoDA | SDXL+CoDA |
> | :---: | :---: | :---: | :---: | :---: | :---: | :---: | :---: |
> | ImageIDC | 10 | 40.8$\pm$1.3 | 54.1$\pm$0.4 | 55.9$\pm$2.1 | 56.2$\pm$0.6 | **59.4$\pm$1.5** | 58.5$\pm$0.9 |
> | ImageIDC | 20 | 42.4$\pm$0.7 | 58.9$\pm$0.2 | 61.9$\pm$0.9 | 65.4$\pm$1.5 | **65.8$\pm$1.2** | 64.6$\pm$0.5 |
> | ImageIDC | 50 | 50.4$\pm$1.6 | 64.3$\pm$0.6 | 72.1$\pm$0.8 | 76.0$\pm$1.0 | 74.6$\pm$1.6 | **77.6$\pm$0.6** |
> | ImageNette | 10 | 53.4$\pm$1.1 | 59.1$\pm$0.7 | 66.4$\pm$2.4 | 66.6$\pm$0.5 | 65.2$\pm$1.5 | **68.8$\pm$0.1** |
> | ImageNette | 20 | 59.6$\pm$0.9 | 64.8$\pm$1.2 | 71.2$\pm$0.5 | 72.2$\pm$0.9 | **73.0$\pm$1.2** | 72.1$\pm$0.2 |
> | ImageNette | 50 | 69.4$\pm$1.4 | 73.3$\pm$0.9 | 79.5$\pm$1.3 | 80.1$\pm$1.3 | 81.4$\pm$1.3 | **83.1$\pm$0.3** |
> | ImageWoof | 10 | 32.2$\pm$0.8 | 34.7$\pm$0.5 | **40.4$\pm$1.9** | 38.2$\pm$0.8 | 39.2$\pm$1.3 | 39.2$\pm$0.7 |
> | ImageWoof | 20 | 37.6$\pm$1.2 | 41.1$\pm$0.8 | 43.6$\pm$1.6 | 43.4$\pm$1.6 | **44.2$\pm$0.9** | 42.5$\pm$0.6 |
> | ImageWoof | 50 | 41.8$\pm$1.7 | 49.3$\pm$0.2 | 56.5$\pm$1.9 | 55.4$\pm$1.1 | 59.2$\pm$0.7 | **59.4$\pm$1.0** |
>
> **Transfer to DiT (DiT+CoDA):**
> 1.  As anticipated, the Target-Trained DiT is inherently aligned with the target dataset and can effectively perform the DD task on its own.
> 2.  Transferring CoDA to DiT significantly boosts its original capability. At the IPC=50 setting, it surpasses the original DiT by **10.3%**, **8.1%**, and **9.9%** on ImageIDC, ImageNette, and ImageWoof, respectively.
> 3.  The DiT+CoDA combination outperforms the previous SOTA (DiT+MGD3) in all but three configurations, and in several cases even exceeds our "SDXL+CoDA" configuration.
>
> **Transfer to SD3 (SD3+CoDA):**
> 1.  Similar to SDXL, SD3 is a general-purpose model and thus is unable to independently perform the DD task efficiently.
> 2.  Integrating CoDA leads to a substantial performance improvement over the base SD3 model across all tested configurations.
> 3.  On ImageIDC and ImageNette, the SD3+CoDA combination matches or even surpasses the previous SOTA method (DiT+MGD3), validating the framework's effectiveness on Flow Matching architectures.
>
> We speculate this is because CoDA's discovery and alignment stages are decoupled. Transferring to another diffusion architecture only changes the executor of the "guided generation." The injection of such an external condition can be implemented in any mainstream architecture, even if the specific form of injection differs from CoDA's.
>
> We have added this analysis to the ablation study section "Transferring CoDA to Target-Trained Models" and Appendix A.16 in the revised paper.

---

> ### Author Response · Authors · 2025-11-28
>
> Dear Reviewer NgTi,
>
> We would like to express our sincere appreciation for your insightful feedback and thorough review of our submission. Your constructive comments regarding computational cost and robustness have been instrumental in refining our work.
>
> We would like to inquire if our response and these new results effectively address your concerns. We are fully committed to addressing any remaining issues and eagerly anticipate your further feedback.
>
> Best Regards,
>
> Authors of Submission 8388

---

### Author Response · Authors · 2025-11-30
**Rebuttal Summary**

# Rebuttal Summary

**Dear Area Chair and Reviewers,**

We would like to express our sincere appreciation for your careful evaluation of our paper and for your valuable service. Below, we summarize the primary contributions of our submission and highlight the major points discussed in our rebuttal.

This paper proposes **CoDA**, a dataset distillation framework leveraging an off-the-shelf text-to-image model (SDXL). Instead of relying on diffusion models pre-trained on the target dataset (e.g., utilizing an ImageNet-trained DiT to distill ImageNet), we introduce "Distribution Discovery" and "Distribution Alignment" to bridge the distribution gap between general generative priors and specific domains. This achieves SOTA performance without the prohibitive cost of pre-training, establishing CoDA as a truly universal solution capable of performing dataset distillation tasks on any arbitrary dataset.

In the rebuttal, we address reviewers' concerns with extensive new experiments and analyses as follows:

**Verified Cross-Architecture & Cross-Space Generality (ptsN, NgTi):**

We demonstrated CoDA's modular adaptability by transferring the Discovery stage to the CLIP feature space and the Alignment stage to DiT and SD3 architectures. All three new configurations achieved performance comparable to or even surpassing previous SOTA baselines, compellingly proving CoDA's exceptional transferability and efficiency.

**Proven Robustness & Stability (NgTi, ugRR):**

1. **Fine-Grained Data:** We added the challenging **ImageWoof** benchmark, where CoDA established a new SOTA across all high-IPC settings.
2. **Pipeline Stability:** We conducted sensitivity analyses using random seeds as perturbations. The results show negligible variance, refuting concerns about the pipeline being "brittle."
3. **Parameter Configuration:** Further optimal parameter analysis confirms the existence of a **"universal parameter"** configuration that achieves optimal performance across various datasets, even those from different domains.
4. **Data Noise:** New experiments with **noisy labels** and **semi-supervised** settings confirm CoDA's inherent ability to filter outliers and function with incomplete data.

**Clarified "Training-Free" Value & Efficiency (ptsN, cWUx):**

We clarified that "Truly Training-Free" in our original submission refers to **zero transfer cost** for new domains, which has been further clarified in our revision. In addition, we removed the word "Truly" from the title in our revision to avoid ambiguity. We validated this by adapting CoDA to **Places365** and the specialized medical dataset **DRIVE** with **zero re-training or manual intervention**—a capability unattainable by target-trained methods. Detailed efficiency breakdowns confirm CoDA is computationally competitive while offering superior flexibility.

**Unveiled Generation Mechanism (cWUx, ugRR):**

We provided theoretical and visual analyses of the guidance parameter $\gamma$. We demonstrated that CoDA leverages the generative prior for the **intelligent refinement** of real images, explaining why our generated set ($G$) consistently outperforms the retrieved real set ($R$).

---

In summary, CoDA offers an efficient, robust, and generalizable paradigm for dataset distillation. We sincerely thank the area chair and all reviewers once more for the thoughtful feedback, and look forward to any further opinions.

---

### Meta-Review · Area_Chair_bZ8o · 2025-12-23

**Summary:**

The major concerns shared by reviewers include:
1. The method design is not actually "training-free," as even without targeted training, diffusion models are still trained by large amounts of data. The distribution discovery and alignment process also inherently embeds "training."
2. The hyperparameters need to be tuned for different datasets, which also downplays the claim of "training-free."

Some other concerns include:
1. Can the proposed method be adapted to other diffusion models?
2. The robustness of the proposed method against UMAP change.
3. Differences from previous methods.
4. Visual validation of UMAP.
5. Whether the guidance will lead to a real sample.

**Reviewer Concerns:**

Most of the concerns have been addressed, except for the following minor issues:
1. The exploration of hyperparameter tuning is limited. The authors are recommended to expand the hyperparameter groups to reveal the range within which the method is robust.
2. In the supplemented experiments of DiT models, why are the adopted $\gamma$ so different from SDXL?

The authors are encouraged to incorporate these two questions in revising their manuscript.

**Reviewer Scores:**

As most of the concerns have been addressed, I would expect that the reviewers can agree that this paper has reached the bar of acceptance.

---

### Decision · Program_Chairs · 2026-01-26

Accept (Poster)